# Toll signaling controls stem cell proliferation in intestinal regeneration and tumorigenesis

Guofan Peng [1,2,7], Shichao Yang[1,2,7], Yuexia Zhang[1,2,7], Yu Zhao[1,2], Xiaoyun Huang [3], Shengen Yi[4], Lei Gu [5], Ganqian Zhu[1,2], Kewei Zheng [1,2], Huijun Zhou [6✉], Kang Han [1,2✉] & Jun Zhou [1,2✉]

## Abstract

**The *Drosophila* Toll/NF-κB pathway has been extensively studied for its roles in innate immunity and embryonic development. Nevertheless, the regulatory mechanisms underlying Spz/Toll signaling in non-immune contexts remain poorly understood. Here, we demonstrate a critical role for *Toll* in regulating intestinal stem cell activity through direct transcriptional control of *PI3K* and *Akt* in an insulin-independent manner. Time-series transcriptomic analysis of intestinal damage and repair responses reveals that the stress-responsive factor *Jumu* regulates *Spz* expression to activate Toll signaling. Disruption of the Jumu/Spz/Toll cascade or PI3K/Akt signaling impairs intestinal regeneration and suppresses tumor growth, and epistasis analysis confirms that PI3K/Akt functions downstream of *Toll*. Our findings elucidate an autocrine Spz/Toll-mediated mechanism that drives stem cell function via the PI3K/Akt pathway during tissue homeostasis and uncover a critical non-immune role of Toll signaling in both physiological and pathological contexts.**

**Keywords** Toll/TLRs Signaling; Akt/PI3K Signaling; Intestinal Stem Cell; Tumorigenesis; *Drosophila*
**Subject Categories** Cancer; Signal Transduction; Stem Cells & Regenerative Medicine

## Introduction

The *Drosophila* Toll pathway was originally identified for its essential role in dorsal-ventral patterning during embryogenesis (reviewed in Nüsslein-Volhard, 2022). The pathway connects the Toll receptor—whose cytoplasmic domain is homologous to the interleukin-1 receptor—to gene expression through *dorsal*, the first *Drosophila* homolog of mammalian NF-κB (Steward et al, 1984). The mammalian homolog of Toll is Toll-like receptor (*TLR*). *TLRs* constitute an evolutionarily conserved family regulating innate immunity across species (Kawai et al, 2024; Ronald and Beutler, 2010; Valanne et al, 2022). In mammals, *TLRs* act as pattern recognition receptors, directly binding conserved microbial molecules such as peptidoglycan, lipopolysaccharides, flagellin, and viral RNA (Ronald and Beutler, 2010). In contrast, *Drosophila Toll* is activated not by microbial molecules but by the endogenous ligand Spätzle (*Spz*), a homolog of nerve growth factor (Hanson and Lemaitre, 2020; Kounatidis and Ligoxygakis, 2012; Leulier and Lemaitre, 2008). In *Drosophila*, pathogen recognition occurs upstream of *Toll* via peptidoglycan recognition proteins (PGRPs) and glucan-binding proteins (GNBPs), such as PGRP-SA and GNBP1. These proteins trigger a proteolytic cascade that culminates in the activation of the Spätzle-processing enzyme (SPE). SPE cleaves Spz, thereby enabling it to bind and activate Toll. Upon ligand binding, the adapter protein MyD88 promotes the formation of a MyD88/Tube/Pelle hetero-trimer. This complex induces the degradation of Cactus, releasing the transcription factors Dif and dorsal, which translocate to the nucleus to activate target genes, including antimicrobial peptides (AMPs) such as *drosomycin* (*drs*) (De Gregorio et al, 2002). *Toll* mutants exhibit increased susceptibility to fungal infections and fail to induce *drs* expression (Lemaitre et al, 1996).

Beyond its functions in immunity and embryogenesis, Toll signaling regulates diverse physiological processes across species. In *Drosophila*, larvae lacking *Toll-6* or *Toll-8* display reduced locomotion, defective neuromuscular junction growth, and fewer synapses (McLaughlin et al, 2016). Toll signaling is also required for the survival of dopaminergic neurons (Zhang et al, 2024a). Additionally, Toll signaling mediates epithelial cell competition, in which mutant cells with relative growth disadvantages are eliminated by apoptosis through confrontation with surrounding wild-type cells (Meyer et al, 2014). *Toll-6* has been shown to drive mechanical tension-dependent tumor cell competition in *Drosophila* (Kong et al, 2022). In mammals, although *TLRs* are primarily characterized as regulators of innate and adaptive immunity (Hamerman and Barton, 2024; Liu et al, 2010), accumulating evidence highlights their critical roles in embryogenesis and tissue homeostasis. For example, *TLRs* regulate the proliferation of neural progenitor cells (NPCs) during embryonic brain development, as

[1]The Affiliated XiangTan Central Hospital of Hunan University, School of Biomedical Sciences, Hunan University, Changsha 410082 Hunan Province, China. [2]Hunan Key Laboratory of Animal Models and Molecular Medicine, Hunan University, Changsha 410082 Hunan Province, China. [3]Intelliphecy Center for Systems Medicine, Intelliphecy, Shenzhen 518000, China. [4]Department of General Surgery, The Second Xiangya Hospital of Central South University, Changsha 410011 Hunan, China. [5]Epigenetics Laboratory, Max Planck Institute for Heart and Lung Research, 61231 Bad Nauheim, Germany. [6]Gastroenterology Department 1,The Affiliated Cancer Hospital of Xiangya School of Medicine, Central South University/Hunan Cancer Hospital, Changsha 410013 Hunan, China. [7]These authors contributed equally: Guofan Peng, Shichao Yang, Yuexia Zhang. ✉E-mail: zhouhuijun@hnca.org.cn; hankang1988@hnu.edu.cn; junzhou82@hnu.edu.cn

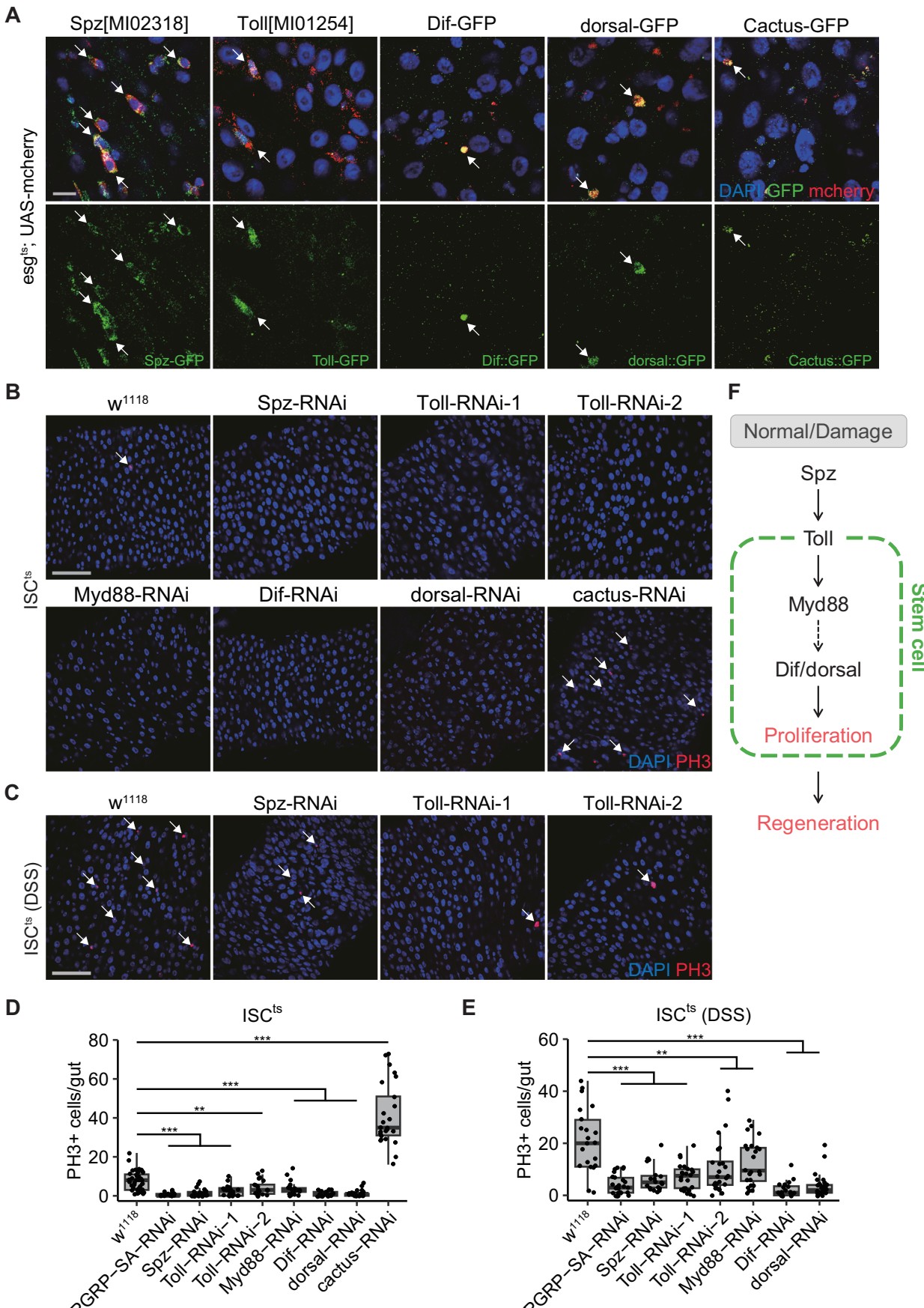

**Figure 1. Toll signaling controls intestinal stem cell proliferation.**

(A) Representative images of the midgut of flies with indicated genotypes expressing *Spz*, *Toll*, *Dif*, *dorsal* or *Cactus-GFP* (green) at 29 °C for 10 days. Nuclei (blue), *esg>mCherry* (red). The white arrows show GFP-positive ISCs. Scale bar: 10 µm. (B) Representative images of the posterior midgut of flies with indicated genotypes at 29 °C for 10 days. PH3 staining (red), Nuclei (blue). The white arrows show PH3-positive cells. Scale bar: 50 µm. (C) Representative images of the posterior midgut of flies with indicated genotypes with 3% DSS treatment at 29 °C for 1 day (after 10 days of normal food). PH3 staining (red), Nuclei (blue). The white arrows show PH3-positive cells. Scale bar: 50 µm. (D) Quantification of PH3-positive cells per adult midgut of the indicated genotypes at 29 °C for 10 days. $n = $ 37, 20, 25, 20, 22, 23, 18, 26, 25 (from left to right). (E) Quantification of PH3-positive cells per adult midgut of the indicated genotypes with 3% DSS treatment at 29 °C for 1 day (after 10 days of normal food). $n = $ 21, 26, 19, 24, 25, 24, 19, 27 (from left to right). (F) Model of Toll-induced intestinal regeneration. Data information: In (D, E), box plots indicate median (center line), 25th–75th percentiles (box) and minima/maxima within 1.5 × interquartile range (whiskers); outliers are shown as individual points. Statistical significance in (D, E) was determined using a two-tailed unpaired *t*-test (*$p < 0.05$; **$p < 0.01$; ***$p < 0.001$). Exact *p* values are provided in Table EV1. Source data are available online for this figure.

evidenced by impaired neurogenesis in *Tlr2*-deficient mice (Okun et al, 2011, 2010). TLR signaling in hypothalamic neurons also contributes to age-associated obesity (Shechter et al, 2013). In the intestine, *TLR4* not only promotes colitis-associated cancer in mice and humans (Burgueño et al, 2021), but also protects against dextran sulfate sodium (DSS)-induced acute colitis (Shi et al, 2023). Although these studies highlight emerging non-immune functions of *TLRs*, their roles in tissue homeostasis and disease—particularly beyond the nervous and immune systems—remain poorly understood.

The *Drosophila* midgut epithelium undergoes rapid renewal driven by highly proliferative intestinal stem cells (ISCs). These ISCs self-renew and give rise to two progenitor lineages: enteroblasts (EBs) that terminally differentiate into absorptive enterocytes (ECs), and pre-enteroendocrine cells that generate secretory enteroendocrine cells (EEs) (Micchelli and Perrimon, 2006; Ohlstein and Spradling, 2006; Zeng and Hou, 2015). Consequently, *Drosophila* serves as a powerful genetic model to dissect the molecular mechanisms underlying stem cell biology and intestinal tumorigenesis (Bahuguna et al, 2021; Bonfini et al, 2021; Buchon et al, 2009; Guo et al, 2013; Jiang et al, 2011; Patel et al, 2015; Zhai et al, 2015; Zhang et al, 2024b; Zhou et al, 2021, 2015; Zhou and Boutros, 2020). Here, we report a non-immune role of Toll signaling in the regulation of stem cell activity during both homeostasis and tumorigenesis. Mechanistically, we demonstrate that the Toll-dependent transcription factors Dif and dorsal directly regulate the expression of PI3K and Akt to promote stem cell proliferation. Inhibition of Toll/PI3K/Akt signaling suppresses stem cell mitosis during intestinal regeneration and tumorigenesis. Conversely, activation of Toll/PI3K/Akt signaling enhances stem cell proliferation and tumor growth. Furthermore, we identify the transcription factor Jumu as a direct regulator that binds to the *spz* promoter to drive its expression, thereby initiating the Toll/PI3K/Akt signaling cascade. Collectively, our study reveals a previously uncharacterized mechanism whereby Jumu-mediated Spz/Toll/PI3K/Akt signaling controls stem cell function during intestinal regeneration and tumorigenesis.

## Results

### Toll signaling controls intestinal stem cell proliferation

The canonical role of Toll signaling in inducing AMPs expression during systemic infection is well established (Leulier et al, 2003; Vaz et al, 2019, Fig. EV1A,G,H). However, flies with EC-specific *Dif* depletion exhibited only moderate susceptibility to oral infection

(Fig. EV1C,G). In contrast, ISC/EB-specific knockdown of *Dif* significantly compromised host survival following infection (Fig. EV1E,G). Simultaneously, we inhibited the IMD pathway via RNAi against *Relish*, the NF-κB-like transcription factor essential for this signaling cascade. Under these conditions, we observed no significant change in survival following either systemic or intestinal challenge with *Staphylococcus aureus* (*S.a*) (Fig. EV1B,D,F,G). To investigate the transcriptional response of the Toll pathway to infection, we analyzed publicly available RNA-seq datasets (Bou Sleiman et al, 2020). We observed that several Toll-related genes and their downstream targets, such as *Defensin* (*Def*) and *Metchnikowin* (*Mtk*), were significantly upregulated following *Pseudomonas entomophila* (*P.e*) infection (Fig. EV1I,K). qPCR analysis confirmed that *P.e* challenge markedly induced *Toll* transcription (Fig. EV1J). Taken together, these results indicate that the Toll pathway is robustly activated in response to intestinal infection. Although Toll activity in ISCs/EBs contributes to host defense, overall intestinal Toll signaling is not strictly required for survival following infection.

To characterize the distribution of Toll pathway components in the intestine, we examined the expression of *Spz*, *Toll*, *Cact*, *Dif*, and *dorsal* using transgenic reporters. These included MiMIC lines carrying a GFP cassette (*Spz[MI02318]* and *Toll[MI01254]*) and protein fusion lines (*Cact-GFP*, *Dif-GFP*, *dorsal-GFP*). These reporters were crossed to the *esg^{ts}>mCherry* line (*esg-Gal4, UAS-mCherry, tubGal80^{ts}*), which labels ISCs and EBs in red. Interestingly, we observed that *Spz* and *Toll* transcripts, as well as Cact, Dif, and dorsal fusion proteins, were primarily enriched in esg+ progenitor cells (ISCs and EBs) (Fig. 1A). In *ISC^{ts} > GFP* (*esg-Gal4, Su(H)-Gal80, UAS-GFP, Tub-Gal80^{ts}*) flies, the *Su(H)-Gal80* transgene suppresses Gal4 activity specifically in Su(H)-positive EBs, thereby restricting *esg-Gal4*-driven expression to ISCs. This genetic setup allows ISC-specific manipulation and visualization. Using *ISC^{ts} > GFP* flies, we further confirmed the protein expression of dorsal and Cactus in ISCs by antibody staining (Fig. EV2A). These results suggest that Toll pathway components are highly enriched in intestinal stem cells.

To determine whether Toll signaling regulates ISC proliferation, we used the ISCts driver to knock down core pathway components (*PGRP-SA*, *Spz*, *Toll*, *Myd88*, *Dif*, and *dorsal*) by RNAi and quantified mitotic activity using phospho-histone H3 (PH3) staining. Depletion of these Toll pathway components resulted in a marked reduction in PH3-positive cells (Figs. 1B,D and Fig. EV2C). In contrast, knockdown of *Cactus*, a negative regulator of Toll signaling, led to an increase in PH3-positive cells (Fig. 1B,D). Previous studies used the detergent DSS and the herbicide Paraquat to damage the intestinal epithelia and trigger ISC proliferation and

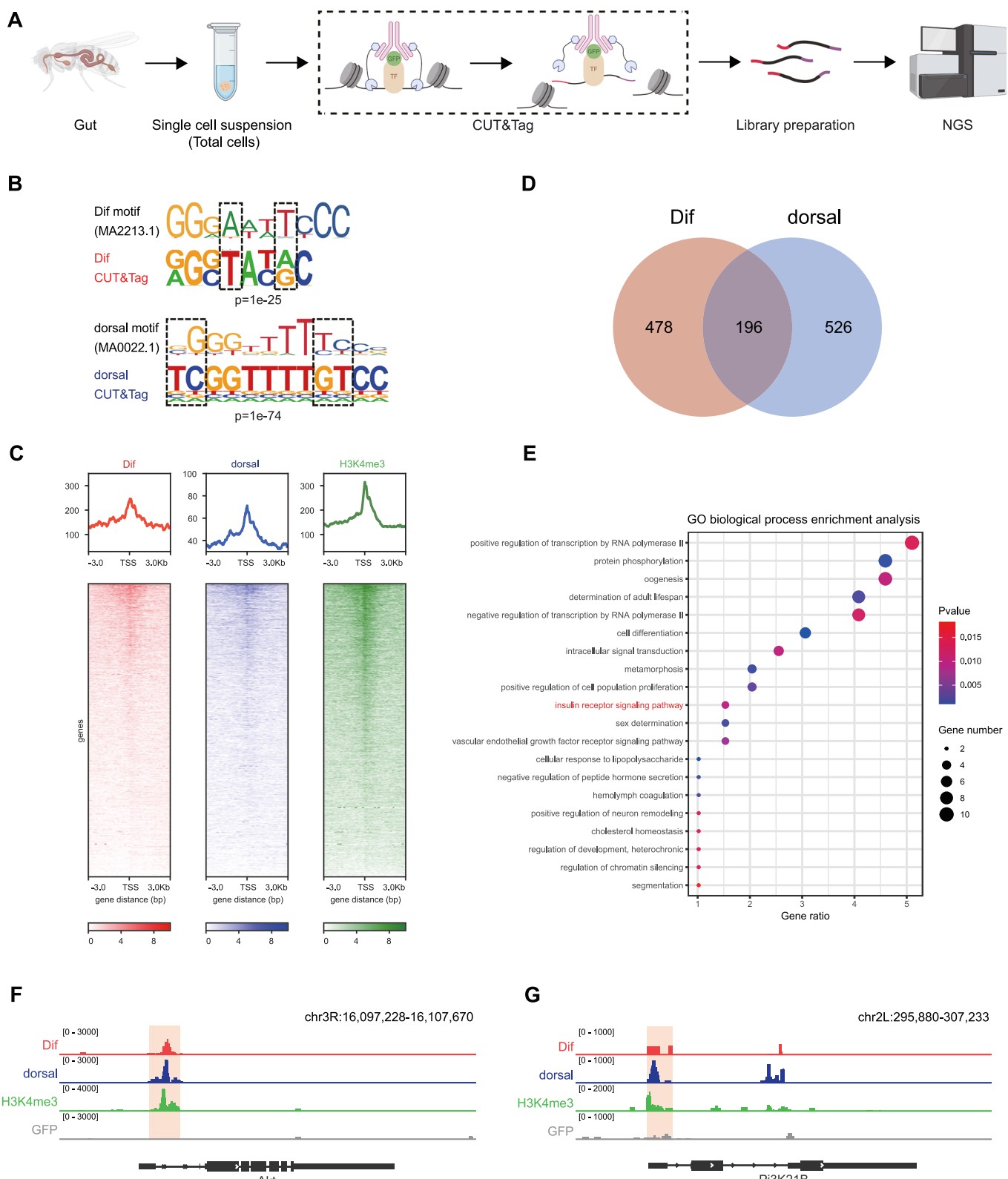

**Figure 2. Toll pathway regulates the expression of PI3K/Akt through a Dif/dorsal-mediated transcriptional mechanism.**

(A) A schematic view of the CUT&Tag assay in *Drosophila* intestinal cells. (B) Identified motifs from Dif and dorsal peaks compared with Dif (MA2213.1) and dorsal (MA0022.1) motifs reported in the JASPAR database. The corresponding *p* values are given. Black dashed boxes show different nucleotides identified from CUT&Tag in the intestinal cells of *Drosophila*. (C) Heatmaps showing the genome-wide CUT&Tag binding profiles of Dif (red), dorsal (blue) and H3K4me3 (green) at TSS regions in the intestinal cells. (D) Venn diagram showing overlap between Dif and dorsal peaks in the intestinal cells. (E) GO term enrichment for common targets of Dif and dorsal. (F, G) Integrative genomics viewer (IGV) snapshots showing the enrichment of Dif (red), dorsal (blue), and H3K4me3 (green) at the *Akt* or *Pi3K21B* gene loci. The high-confidence peaks of Dif, dorsal, and H3K4me3 are highlighted in orange (*q* value <0.01, TSS-proximal). Source data are available online for this figure.

regenerative responses (Jiang et al, 2011). We next investigated whether Toll signaling is required for this regenerative response. qPCR assays showed that DSS treatment induced the expression of Toll pathway components (Fig. EV2D). Accordingly, inhibition of Toll signaling dampens the DSS- and Paraquat-induced intestinal regenerative responses (Figs. 1C,E and EV2E–G). Similarly, ISC-specific knockdown of *Dif* significantly suppressed mitotic activity following *P.e* infection (Fig. EV2B,H). Furthermore, intestinal regeneration was severely compromised in *PGRP-SA^seml^* mutant flies (Fig. EV2I,J). These data together indicate that the Toll pathway plays an essential role in ISCs by controlling their proliferation during homeostasis and regeneration (Fig. 1F).

## Toll pathway regulates PI3K/Akt expression through a Dif/dorsal-mediated transcriptional mechanism

To elucidate the molecular mechanisms underlying Toll-mediated intestinal regeneration, we employed the cleavage under targets and tagmentation (CUT&Tag) assay to map the chromatin landscape (H3K4me3) and the binding of transcription factors Dif and dorsal in the *Drosophila* intestine under homeostatic conditions (Fig. 2A). We used an anti-GFP antibody to target Dif-GFP and Dorsal-GFP fusion proteins, followed by Tn5 transposase-mediated tagmentation, which allowed us to profile their genomic binding sites. The chromatin binding profiles of Dif and dorsal showed a strong correlation with each other (Fig. EV2M). Density and heatmap analyses revealed that the Dif, dorsal, and H3K4me3 binding peaks were enriched around transcription start sites (TSSs) (Figs. 2C and EV2N). Moreover, the chromatin binding peaks of Dif and dorsal were broadly distributed on chromosomes 1, 2, and 3 (Fig. EV2K,L). In total, we obtained 674 and 722 target genes for Dif and dorsal, respectively (Fig. 2D). The complete list of targets is provided in Dataset EV1. In addition, the detected peaks for both Dif and dorsal were significantly enriched for their respective transcription factor binding motifs reported in the JASPAR database (Fig. 2B). Gene Ontology (GO) analysis suggested that Dif and dorsal target genes were enriched in the insulin receptor signaling pathway (Fig. 2E). Detailed analysis revealed high chromatin accessibility (H3K4me3) and overlapping Dif and dorsal peaks in close proximity to the TSSs of *Akt* and *Pi3K21B* genes (Fig. 2F,G). Given that histone modification H3K4me3 is associated with active transcription, these results suggest that Toll-dependent transcription factors Dif and dorsal directly bind to the TSS regions of *Akt* and *PI3K*, thereby potentially activating their gene transcription.

## Toll receptor-mediated PI3K/Akt pathway controls stem cell activity

We next asked whether Toll pathway activation induces the transcription of *PI3K* and *Akt* in the intestine. To this end, we

overexpressed *Dif* in stem cells using *ISC^ts^* and monitored the expression of *PI3K* and *Akt* in the intestine by qPCR assays. As expected, we observed that Toll pathway activation induced the expression of *Akt* and *PI3K* (Fig. 3A). Consistently, intestinal damage induced by DSS treatment and *P.e* infection upregulated Toll pathway components and markedly increased *Akt* and *PI3K* expression (Fig. EV3A,B,I). Importantly, knockdown of *Dif* markedly suppressed *P.e* infection-induced upregulation of *Akt* and *PI3K*, indicating that this upregulation in response to *P.e* infection depends on Toll pathway activity within ISCs (Fig. EV3B). *Toll^10B^* is a constitutively active gain-of-function mutant of the Toll receptor generated by a cysteine-to-threonine substitution (G2.916 → A) in the extracellular domain, which disrupts ligand-independent autoinhibition and leads to sustained activation of downstream signaling pathways (Maxton-Küchenmeister et al, 1999; Schneider et al, 1991). To confirm that Toll activation enhances Akt activity, we stained for phospho-Akt (p-Akt) when *Toll^10B^* or *Dif* was overexpressed in ISCs (*ISCts > UAS-Toll^10B^* or *UAS-Dif*). We found that *Toll^10B^* or *Dif* overexpression increased both the number of p-Akt-positive stem cells and the intensity of p-Akt staining per cell, indicating enhanced p-Akt activity in stem cells (Figs. 3B,D and EV3E,F). Consistently, ISC-specific over-expression of *Pi3K21B* also elevated p-Akt levels (Figs. 3B,D and Fig. EV3E,F). As a negative control, *Akt* RNAi effectively abolished p-Akt staining, confirming the antibody specificity and signal authenticity (Figs. 3B,D and EV3E,F).

The insulin pathway is known to be involved in the regulation of intestinal stem cell activity in *Drosophila* (Foronda et al, 2014; Mattila et al, 2018; Biteau et al, 2010). Consistent with previous findings, *PI3K* and *Akt* inhibition in stem cells reduced their proliferative ability (Figs. 3C,E,G,H and EV3C,D). In contrast, overexpression of *PI3K* and *Akt* induced stem cell proliferation (Fig. 3C,E,F). Notably, overexpression of *Toll* or *Dif* strongly enhanced stem cell proliferative activity, whereas treatment with Akt and PI3K inhibitors significantly suppressed this hyperproli-ferative phenotype (Fig. EV3G,H). In addition, the overexpression of *Toll*, *Dif*, *Akt*, or *PI3K* all resulted in a significant shortening of the lifespan of flies, suggesting that sustained activation of proliferation adversely affects overall organismal health (Fig. EV3J,K). To further determine whether PI3K/Akt acts downstream of the Toll pathway, we conducted epistatic analysis. We concurrently inhibited the Toll pathway (*PGRP-SA* RNAi) while inducing PI3K/Akt (*UAS-Akt*) and monitored the stem cell mitotic activity. We found that the proliferation inhibition caused by *PGRP-SA* RNAi was rescued by *Akt* overexpression (Fig. 3E,F). Conversely, Akt knockdown completely abolished *Dif*-induced hyperproliferation (Fig. 3G,H). Hence, these results suggest that the Toll pathway controls the expression and activity of *Akt* to regulate stem cell proliferation in normal homeostasis and intestinal

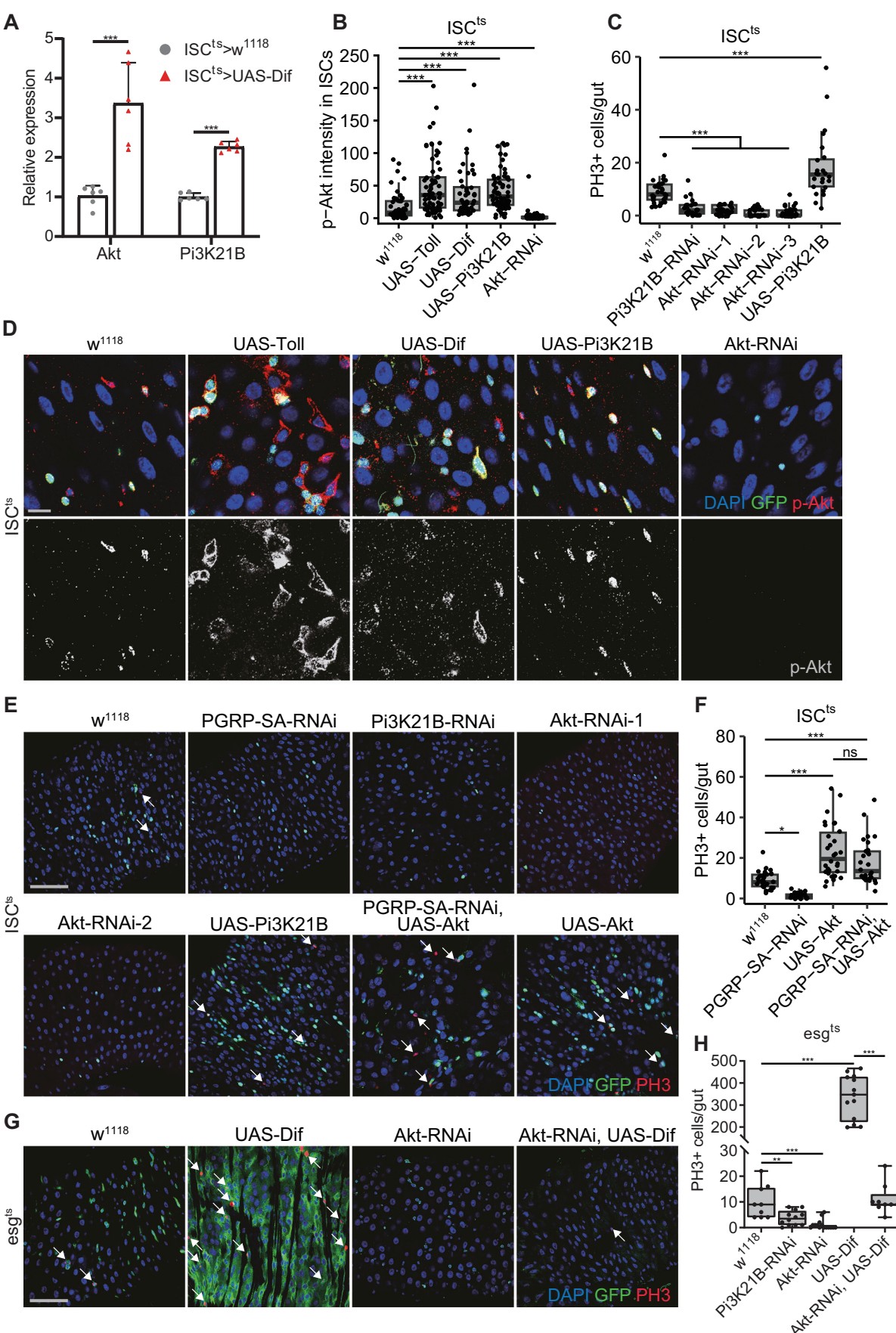

**Figure 3. Toll receptor signaling controls intestinal stem cell activity via the PI3K/Akt pathway.**

(A) RT-qPCR analysis of *Pi3K21B* and *Akt* in the midgut of *ISC^{ts}* > *w^{1118}* and *ISC^{ts}* > *UAS-Dif* flies at 29 °C for 10 days. n = 6 (three independent biological replicates with two technical replicates each). (B) Intensity statistics of p-Akt in the posterior midgut of flies with indicated genotypes at 29 °C for 8 days. n = 55, 69, 47, 65, 64 (from left to right). (C) Quantification of PH3-positive cells per adult midgut of the indicated genotypes at 29 °C for 10 days. n = 34, 24, 28, 25, 33, 24 (from left to right). (D) Representative images of the posterior midgut of flies with indicated genotypes at 29 °C for 8 days. p-Akt staining (red), Nuclei (blue), and ISC (green). The image shown is a magnified view of the boxed region in Fig. EV3E. Scale bar: 10 μm. (E) Representative images of the posterior midgut of flies with indicated genotypes at 29 °C for 10 days. PH3 staining (red), Nuclei (blue), and ISC (green). The white arrows show PH3-positive cells. Scale bar: 50 μm. (F) Quantification of PH3-positive cells per adult midgut of the indicated genotypes at 29 °C for 10 days. n = 34, 25, 30, 28 (from left to right). (G) Representative images of the posterior midgut of flies with indicated genotypes at 29 °C for 10 days. PH3 staining (red), Nuclei (blue), *esg* > *GFP* (green). The white arrows show PH3-positive cells. Scale bar: 50 μm. (H) Quantification of PH3-positive cells per adult midgut of the indicated genotypes at 29 °C for 10 days. n = 9, 15, 13, 12, 9 (from left to right). Data information: In (A), data are presented as mean ± SEM. In (B, C, F, H), box plots indicate median (center line), 25th–75th percentiles (box) and minima/maxima within 1.5 × interquartile range (whiskers); outliers are shown as individual points. Statistical significance in (A–C, F, H) was determined using a two-tailed unpaired *t*-test (ns, no significant difference; *p < 0.05; **p < 0.01; ***p < 0.001). Exact *p* values are provided in Table EV1. Source data are available online for this figure.

regeneration, and molecular evidence in Fig. 2 indicates that *PI3K* also functions downstream of the Toll pathway, potentially contributing to similar cellular responses.

## Toll receptor-mediated PI3K/Akt signaling is required for intestinal tumor growth

To explore potential relevance to human disease, we queried the Alliance of Genome Resources database for human diseases associated with orthologs of Dif and dorsal transcriptional targets (Alliance of Genome Resources Consortium, 2020; Hu et al, 2023). Intriguingly, we found that the Dif/dorsal transcriptional targets were enriched in human diseases like colorectal tumorigenesis, renal fibrosis and neurodegenerative diseases (Fig. 4A). We next re-analyzed published RNA-seq data from *Notch*-depleted *Drosophila* midguts, a well-established intestinal tumor model (Guo et al, 2019). Indeed, we observed a significant increase in the expression of genes involved in the Toll and PI3K/Akt pathways within the stem cells of the *Notch*-deficient tumor intestine (Figs. 4B and EV4A–C). This suggests that intestinal tumor growth transcriptionally activates the Toll and PI3K/Akt pathways in the stem cells.

To further explore whether the Toll receptor-mediated PI3K/Akt pathway is required for intestinal tumorigenesis, we activated or inhibited the Toll and PI3K/Akt pathway activity to monitor tumor growth. Importantly, activation of Toll signaling (*UAS-PGRP-SA*, *UAS-Spz*, *UAS-Toll^{10B}*, *UAS-Dif*) or Akt (*UAS-Akt*) enhanced *Notch-RNAi*-induced tumorigenesis (Figs. 4C,F and EV4D,E). Conversely, inhibition of Toll or PI3K/Akt signaling strongly suppressed intestinal tumor progression (Figs. 4C,D and EV4F). Notably, treatment with an *Akt* inhibitor effectively suppressed both baseline tumor growth and the excessive proliferation induced by *UAS-Spz* or *UAS-Dif* (Figs. 4C,F and EV4D,E). We further found that the Toll receptor-mediated PI3K/Akt pathway is required for the development of an independent intestinal tumor model generated by *APC-RNAi* and *Ras^{v12}* ectopic expression (Figs. 4E and EV5A,B). DCP1 (an apoptosis effector caspase marker) and TUNEL staining revealed that knockdown of *PGRP-SA*, *Toll*, or *Akt* significantly suppressed apoptosis in the intestinal tumor (Fig. EV5C–E). Moreover, *Toll* or *Dif* inhibition significantly decreased both the number of p-Akt-positive stem cells and the p-Akt levels per cell, indicating attenuated *Akt* activity in intestinal tumors (Figs. 4G,H and EV5F). Overall, the above results suggest that Toll

receptor-mediated PI3K/Akt pathway is enriched in tumor conditions, and that Toll/Akt/PI3K activities are critical for intestinal tumorigenesis.

## Toll/PI3K/Akt signaling drives intestinal tumor growth

To assess the pathological relevance of the Toll/Akt/PI3K pathway in intestinal tumor growth, we performed survival assays in tumor-bearing flies after manipulating either Toll or PI3K/Akt signaling. We found that inhibition of either *Toll* or *Akt* in tumor-bearing flies significantly extended the lifespan (Fig. 5A,B). Of note, *PI3K* RNAi and *Dif* RNAi significantly reduced intestinal tumor growth, but had a limited effect on the longevity of the tumor-bearing flies, which might be due to other signaling cross-talks that impaired intestinal stem cell functions and caused animal death. In contrast, activation of the Toll or PI3K/Akt pathway displayed an enhanced mortality of tumor-bearing flies (Fig. 5C,D). Next, we aimed to investigate whether PI3K/Akt functions downstream of the Toll pathway during intestinal tumorigenesis. To this end, we performed epistasis analysis to test the interaction between the Toll pathway and Akt activity in intestinal tumor growth. We found that *Akt* RNAi inhibited *PGRP-SA* overexpression-induced *Notch* tumor growth (Figs. 4C and 5E,F). Conversely, ectopic expression of *Akt* induced intestinal tumor growth even after *Toll* RNAi (Figs. 4C and 5E,G). These results indicated that the Toll receptor-mediated PI3K/Akt pathway has a proliferative role for tumor growth and leads to early death of the animal. Overall, stem cell-specific inhibition of the Toll and PI3K/Akt pathways delays tumor growth and is beneficial for the life expectancy of tumor-bearing flies.

## *Jumeaux* expression is correlated with *Spz* in the intestine

*Spz*, a known morphogen, activates the Toll signaling pathway by binding to Toll-like receptors (Lewis et al, 2013; DeLotto and DeLotto, 1998). To further investigate the upstream regulators of *Spz* during intestinal regeneration, we conducted a time-series transcriptome analysis during DSS-induced intestinal damage and repair. Using hierarchical clustering on the spline-smoothed expression profiles of genes, we identified a gene cluster that had a similar expression pattern to *Spz* (Fig. 6A–C). We then focused on transcription factors among the 102 genes upregulated over time during regeneration (Fig. 6C). The Pearson correlation coefficient (*r*) was calculated to evaluate the relationship between *Spz* and

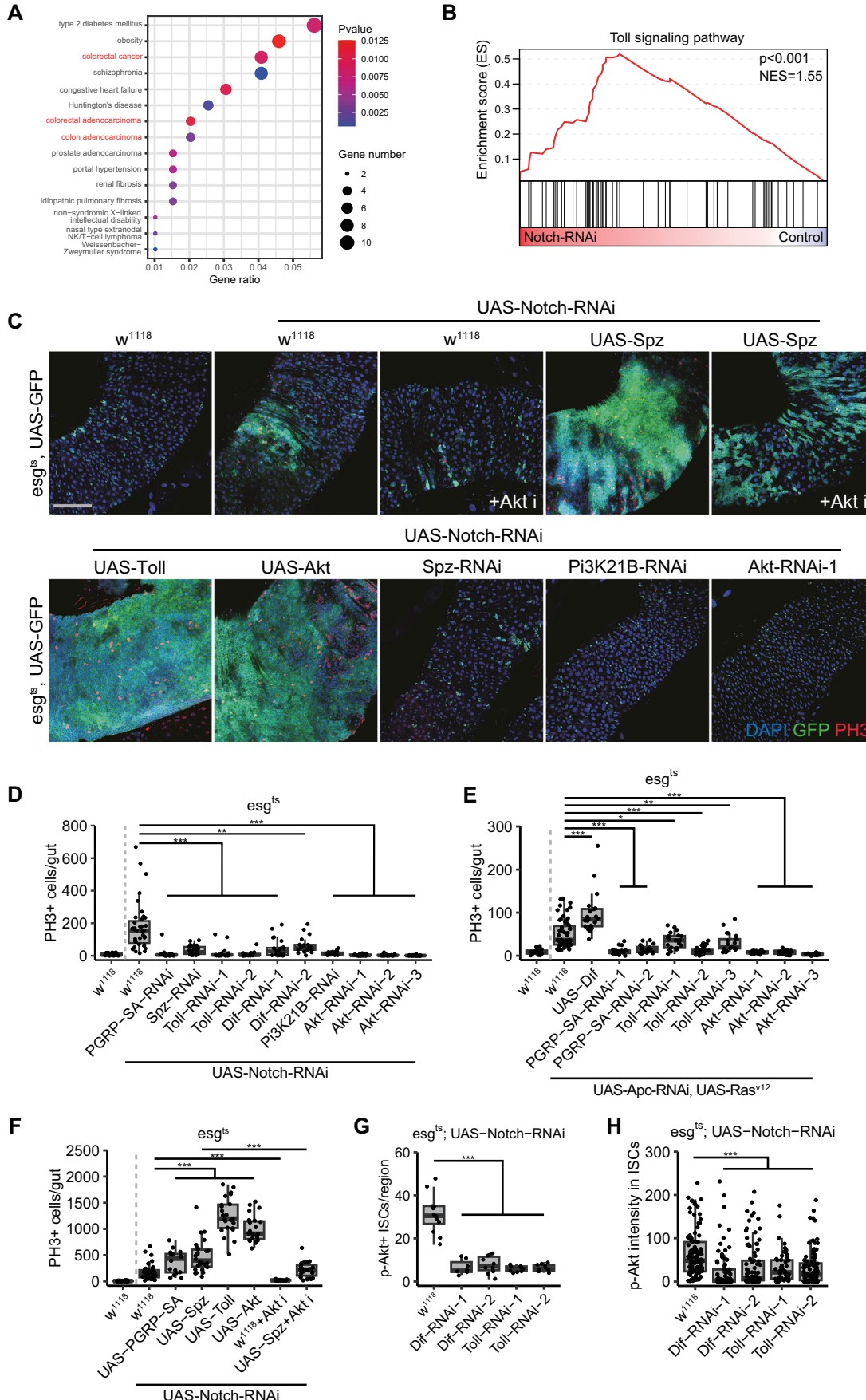

**Figure 4. The Toll receptor-mediated PI3K/Akt pathway is required for intestinal tumor growth.**

(A) Enrichment analysis of common targets of Dif and dorsal in the Alliance of Genome Resources (AGR) disease database. (B) GSEA showing highly significant enrichment for expression in the Toll signaling pathway in ISCs of the Notch-depleted tumor intestine (using dataset GSE112331). (C) Representative images of the posterior midgut of flies with indicated genotypes with or without 100 mM Akt inhibitor at 29 °C for 8 days. PH3 staining (red), Nuclei (blue), and esg > GFP (green). "Akt i" denotes Akt inhibitor. Scale bar: 100 μm. (D) Quantification of PH3-positive cells per adult midgut of the indicated genotypes at 29 °C for 8 days. $n = 29, 39, 23, 28, 20,$ 22, 23, 21, 24, 22, 25, 25 (from left to right). (E) Quantification of PH3-positive cells per adult midgut of the indicated genotypes at 29 °C for 16 days. $n = 17, 59, 23, 21, 22,$ 22, 34, 22, 23, 23, 18 (from left to right). (F) Quantification of PH3-positive cells per adult midgut of the indicated genotypes at 29 °C for 8 days. $n = 29, 39, 16, 26, 23, 23,$ 18, 23 (from left to right). (G) Quantification of p-Akt-positive ISCs in the posterior midgut of flies with the indicated genotypes after 8 days at 29 °C. $n = 12, 7, 10, 10, 11$ (from left to right). (H) Intensity statistics of p-Akt in the posterior midgut of flies with indicated genotypes at 29 °C for 8 days. $n = 108, 146, 120, 86, 112$ (from left to right). Data information: In (D–H), box plots indicate median (center line), 25th–75th percentiles (box) and minima/maxima within 1.5 × interquartile range (whiskers); outliers are shown as individual points. Statistical significance in (D–H) was determined using a two-tailed unpaired $t$-test ($^*p < 0.05$; $^{**}p < 0.01$; $^{***}p < 0.001$). Exact $p$ values are provided in Table EV1. Source data are available online for this figure.

transcription factor expression, with significance assessed using a two-tailed $t$-test ($p < 0.05$). Intriguingly, we found the transcription factor Jumeaux (Jumu) was upregulated during DSS-induced damage and repair. Jumu showed similar signatures as Toll ligand Spz and stem cell marker Delta and esg (Figs. 6D–F and EV6A). Like Spz, Jumu was specifically expressed in intestinal stem cells (Fig. 6G), which is consistent with previous reports (Zeng and Hou, 2015; Dutta et al, 2015; Doupé et al, 2018). In addition, Jumu expression was induced in response to P.e infection (Fig. EV6B). Previous studies have identified that Jumu regulates the proliferation and differentiation of hemocytes and contributes to Toll-dependent formation of melanotic nodules in Drosophila (Zhang et al, 2016). Drosophila Jumu gene is highly conserved with the vertebrate Forkhead Box N4 (FOXN4) (Fig. 6H). FOXN4 is associated with inflammatory bowel disease, which causes severe intestinal damage (Zhang et al, 2022). In addition, Jumu was found to bind the TSS region of the Spz genomic locus (Fig. 6I). Moreover, we found the expression of Jumu was highly enriched in the Notch-RNAi tumor cells, similar to Spz and Delta (Fig. 6J). Together, these results suggest that Spz is potentially regulated by Jumu to mediate Toll activation during intestinal regeneration and tumorigenesis.

### Jumeaux regulates Spz expression to control ISC proliferation

We next investigated whether Jumu is required for Spz transcription. To test this, we performed CUT&Tag-qPCR with two primer pairs targeting the Spz promoter, based on the Jumu binding peaks (Fig. 6I), and one pair targeting an intron as a negative control. Compared to controls, Jumu exhibited strong binding to the Spz promoter, and this binding was further enhanced by DSS treatment. This result confirms that Jumu regulates Spz expression through directly binding to its promoter (Fig. 7A). qPCR assays revealed that Jumu knockdown significantly reduced Spz expression in the intestine (Fig. 7B). Simultaneously, knockdown of Jumu, using either esg- or ISC-specific drivers, decreased both the number and the proliferative activity of ISCs (Figs. 7C,E and EV6C,D). Overexpression of Spz induced excessive ISC proliferation, and this effect was unaffected by simultaneous Jumu knockdown, indicating that Jumu acts genetically upstream of Spz (Fig. 7C,E). We further found that Jumu knockdown markedly attenuated both Spz-GFP and Toll-GFP signals compared to controls, indicating downregulation of Spz and Toll expression (Figs. 7J,K and EV6E,F). In addition, p-Akt staining showed that Jumu knockdown significantly reduced Akt activity and suppressed DSS-induced Akt activation (Fig. EV6G,H). Jumu RNAi also dampened DSS-induced intestinal

regeneration, as evidenced by reduced PH3+ cells (Fig. 7D,F). Since Spz/Toll signaling is required for intestinal tumorigenesis, we next tested whether Jumu is required for Toll-dependent tumor growth using RNAi. Interestingly, Jumu RNAi significantly suppressed the expression of Spz and its downstream targets PI3K and Akt (Fig. 7G), which eventually led to inhibition of tumor development (Fig. 7H,I). Hence, our investigations identify Jumu as a novel damage-response regulator that mediates Toll receptor-dependent stem cell proliferation in both homeostatic and disease conditions.

## Discussion

The Toll signaling pathway was initially characterized as a key regulator of innate immunity, primarily mediating the production of AMPs. However, emerging evidence from both Drosophila and mammalian studies highlights its non-immune roles in development and disease. In this study, we uncover an unexpected role of the Toll pathway in the regulation of stem cell activity, functioning as a critical signaling cascade that governs tissue plasticity during intestinal damage and oxidative stress. Our findings demonstrate that the Toll-dependent NF-κB-like transcription factors Dif and dorsal directly regulate PI3K and Akt expression to promote ISC proliferation and maintain tissue homeostasis. We further identify the stress-responsive transcription factor Jumu as an upstream regulator of Spz, mediating Toll-dependent stem cell function during intestinal regeneration. Furthermore, the Jumu/Spz/Toll-mediated PI3K/Akt signaling cascade is critical for the growth of intestinal tumors and the lifespan of tumor-bearing animals. Collectively, these results highlight a critical non-immune role for Toll signaling in both physiological and pathological contexts.

### Toll receptor signaling in stem cells

TLRs recognize both microbial and non-microbial molecules to regulate transcription networks in physiological and pathological conditions in mammals. Studies propose that TLRs respond to the released host molecules, known as danger-associated molecular patterns (DAMPs), and induce inflammatory responses during tissue damage and repair (Burgueño and Abreu, 2020; Matzinger, 2002). For instance, mammalian TLR4 binds the glycosaminoglycan Hyaluronic acid in the extracellular matrix, thereby promoting intestinal tumor growth. The binding of Hyaluronic acid to TLR4 in macrophages enables $PGE_2$ production and activates EGFR in Lgr5+ intestinal stem cells to promote proliferation (Riehl et al,

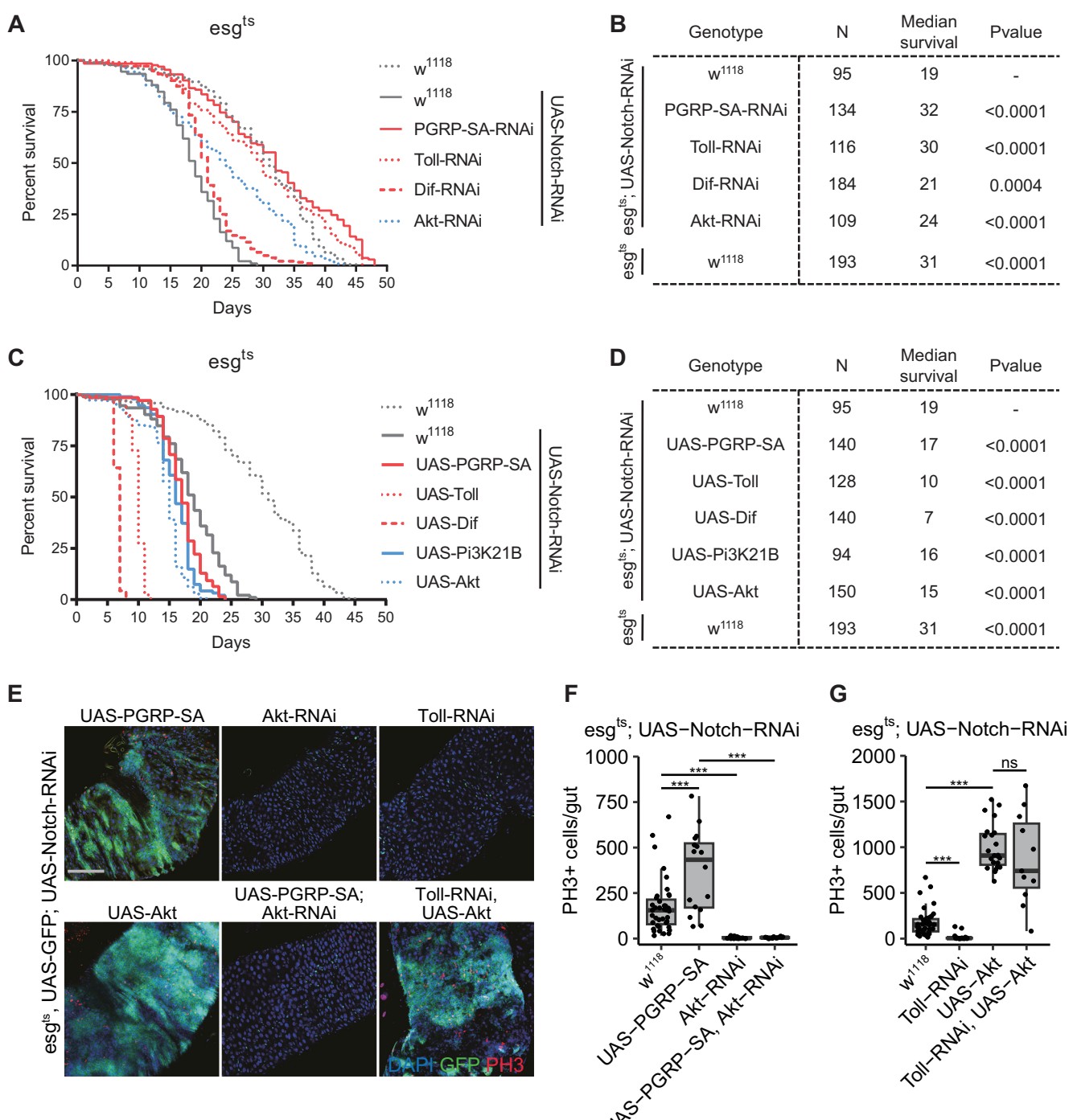

**Figure 5. The proactive role of the Toll/Akt/Pi3K pathway in intestinal tumor growth.**

(A) Lifespan curves of female flies of the indicated genotypes at 29 °C. The number of flies ranged from 95 to 193. (B) Summary of statistics of the lifespan shown in (A). (C) Lifespan curves of female flies of the indicated genotypes at 29 °C. The number of flies ranged from 94 to 193. (D) Summary of statistics of the lifespan shown in (C). (E) Representative images of the posterior midgut of flies with indicated genotypes at 29 °C for 8 days. PH3 staining (red), Nuclei (blue), esg > GFP (green). Scale bar: 100 μm. (F) Quantification of PH3-positive cells per adult midgut of the indicated genotypes at 29 °C for 8 days. $n$ = 39, 16, 22, 13 (from left to right). (G) Quantification of PH3-positive cells per adult midgut of the indicated genotypes at 29 °C for 8 days. $n$ = 39, 20, 23, 11 (from left to right). Data information: In (F, G), box plots indicate median (center line), 25th–75th percentiles (box) and minima/maxima within 1.5 × interquartile range (whiskers); outliers are shown as individual points. Statistical significance in (F, G) was determined using a two-tailed unpaired $t$-test (ns no significant difference; $^{*}p < 0.05$; $^{**}p < 0.01$; $^{***}p < 0.001$). Exact $p$ values are provided in Table EV1. Source data are available online for this figure.

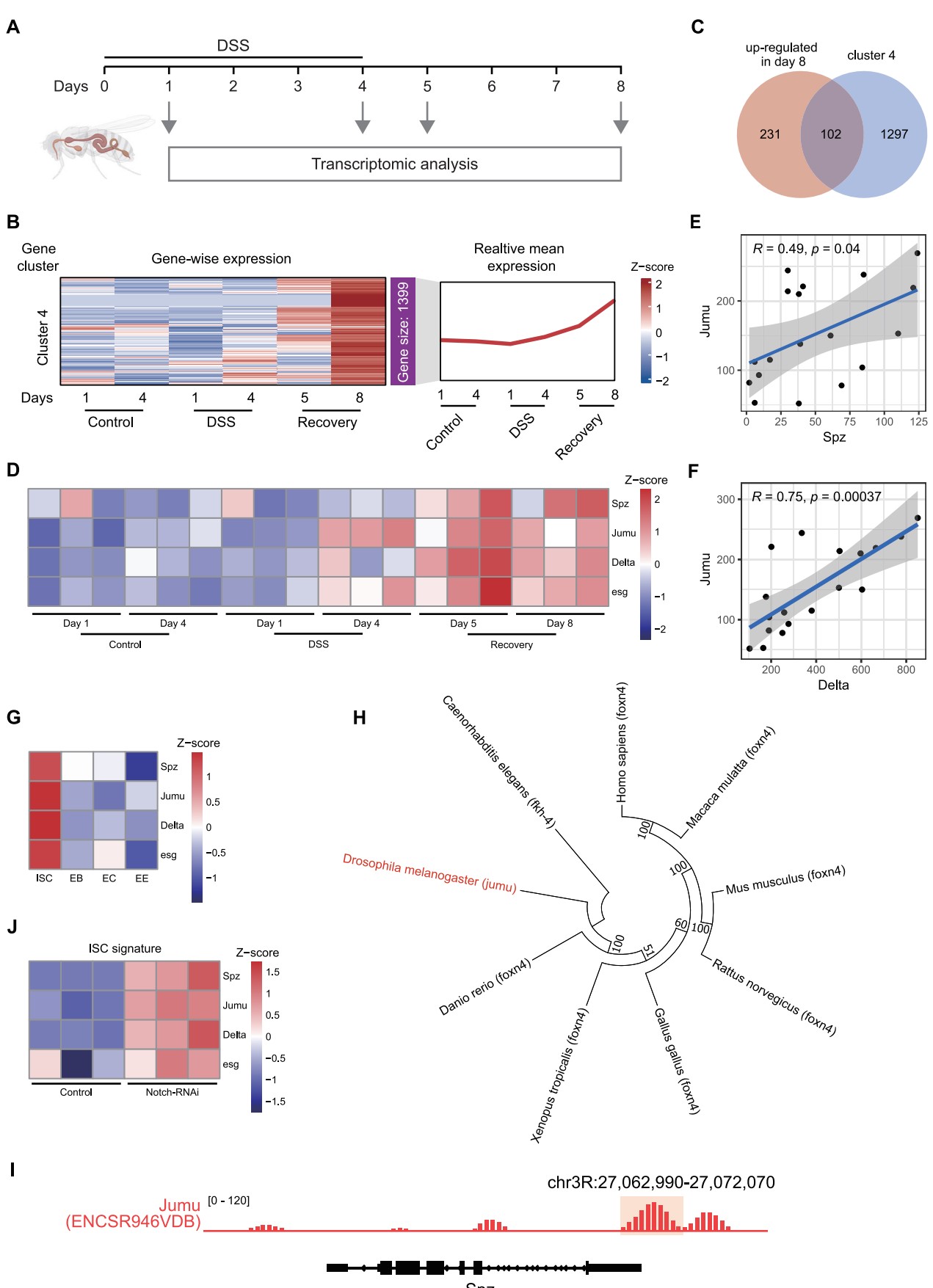

**Figure 6. *Jumu* regulates *Spz* expression by directly binding to the TSS.**

(A) Schematic of experimental design. Flies received 3% DSS in 5% sucrose for 4 days, after which the treatment was replaced with 5% sucrose. Samples were collected at the indicated time points. (B) Clustered heatmap of genes in cluster 4 (left). The mean expression of gene cluster 4 is shown (right). Heatmap displays log2(FPKM + 1) data scaled using a Z-score. (C) Venn diagram showing overlap between 333 upregulated genes in day 8 and 1399 genes in cluster 4. (D) Heatmap showing expression of *Spz, Jumu, Delta,* and *esg* in the midgut of $w^{1118}$ flies with or without DSS treatment. Heatmap displays log2(FPKM + 1) data scaled using a Z-score. (E) Scatter plot showing expression correlation between *Spz* and *Jumu*. *Spz* and *Jumu* were identified among the 102 overlapping upregulated genes from the Venn diagram in (C). Each point represents an individual sample from RNA-seq data, with *Spz* expression on the x-axis and *Jumu* expression on the y-axis. (F) Scatter plot showing expression correlation between *Delta* and *Jumu*. *Delta* and *Jumu* were identified among the 102 overlapping upregulated genes from the Venn diagram in (C). Each point represents an individual sample from RNA-seq data, with *Delta* expression on the x-axis and Jumu expression on the y-axis. (G) Heatmap showing expression of *Spz, Jumu, Delta,* and *esg* in various cell types (using dataset GSE61361). Heatmap displays log2(FPKM + 1) data scaled using a Z-score. (H) Phylogenetic analysis of Jumu and its homologs based on amino acid sequences. Proteins from flies are highlighted in red font. The bootstrap values obtained from 1000 bootstrap iterations are shown in the cladogram. (I) IGV snapshots showing the enrichment of *Jumu* (red, ENCSR946VDB) at the *Spz* gene locus in ChIP-seq data from the ENCODE database. High-confidence peak is highlighted in orange (q value <0.01, TSS-proximal). (J) Heatmap showing expression of *Spz, Jumu, Delta,* and *esg* in ISCs of the Notch-depleted tumor intestine (using dataset GSE112331). Heatmap displays DESeq2 normalized counts scaled using a Z-score. Source data are available online for this figure.

2020, 2015). *TLR4*-deficient neonatal mice exhibit significant loss of *Lgr5* + ISCs, while adults display shortened intestines and colons (Riehl et al, 2015). In addition, *TLR4* activates β-catenin signaling to drive intestinal proliferation (Santaolalla et al, 2013). The TLR pathway also regulates intestinal regeneration during DSS-induced tissue damage and repair. *Myd88* mutant mice show reduced proliferation, delayed repair and increased susceptibility to DSS-induced mortality (Rakoff-Nahoum et al, 2004; Shi et al, 2023). Collectively, these studies implicate TLR/Myd88 signaling in stem cell-mediated intestinal damage response, but the detailed mechanisms remain unclear. Our study reveals that *Drosophila* Toll pathway components, primarily expressed in ISCs, promoted proliferation by transcriptionally regulating PI3K/Akt. Inhibition of Toll/PI3K/Akt signaling reduced ISC populations in response to DSS treatment, suggesting a critical and conserved function of Toll/PI3K/Akt signaling in the regulation of stem cell activity during intestinal homeostasis.

*Pten* is a lipid phosphatase that antagonizes the PI3K/Akt pathway (Mukherjee et al, 2021). Both physiological and oncogenic activation of PI3K signaling elevate the expression of its negative regulator, *Pten* (Mukherjee et al, 2021). Consistent with this, our RNA-seq analysis revealed significant upregulation of *Pten* upon infection or in intestinal tumors, concomitant with activation of the Toll/PI3K/Akt pathway (Figs. EV3A and EV4C). This suggests a compensatory feedback mechanism to maintain pathway homeostasis. A key downstream effector, Foxo, is phosphorylated by Akt, leading to its cytoplasmic retention and functional inactivation (Mukherjee et al, 2021). Accordingly, *Foxo* transcript levels remained stable in intestinal tumors or were slightly reduced upon infection (Figs. EV3A and EV4C). This suggests that Akt primarily regulates Foxo through phosphorylation-mediated inactivation rather than by modulating its mRNA expression. Together, these findings highlight a finely balanced regulatory network in which Toll signaling modulates PI3K/Akt activity to control stem cell proliferation.

## Toll receptor signaling in intestinal tumorigenesis

Components of the Toll receptor signaling pathway are highly expressed in intestinal epithelial cells during colitis-associated cancer and colorectal cancer (Burgueño and Abreu, 2020). *TLR4* activation has been shown to trigger β-catenin and EGFR signaling —two key pathways that drive stem cell proliferation and

tumorigenesis in the intestine (Riehl et al, 2020; Santaolalla et al, 2013). Consistent with a pro-tumorigenic role, *TLR4*-deficient mice are protected against colitis-associated cancer (Santaolalla et al, 2013), and *TLR4* is highly expressed in colorectal cancer patients (Sussman et al, 2014). Constitutive *TLR4* activation promotes spontaneous intestinal tumor growth through cytokine receptor signaling pathways, such as IL-6 signaling (Shi et al, 2020). In line with this, treatment with *TLR4* antagonist TAK-242 reduced the number and size of intestinal tumors in $Apc^{Min/+}$ mice (Wu et al, 2018). Similarly, *Myd88*-deficient mice develop fewer tumors and exhibit reduced levels of inflammatory cytokines, including *IL-1β*, *IL-6* and insulin-like growth factor 1 (*IGF-1*), compared to $Apc^{Min/+}$ controls (Rakoff-Nahoum and Medzhitov, 2007). These findings strongly support the requirement of TLR4/Myd88 signaling in intestinal tumorigenesis. Our study further demonstrates a tumor-promoting role for Toll receptor signaling, providing molecular details showing that the NF-κB-like transcription factors directly regulate the expression and activity of PI3K/Akt to drive tumor progression.

*TLRs* regulate host-microbe interactions and tissue regeneration to safeguard immune homeostasis and intestinal barrier function. Our findings highlight that the TLR/PI3K/Akt pathway regulates stem cell proliferation and tumor growth, extending the role of *TLRs* in intestinal regeneration beyond innate immunity and suggesting the therapeutic potential of targeting this pathway in stem cells for cancer treatment.

Mammalian *TLRs* sense an invasion of bacteria by detecting microbial components such as peptidoglycans. In *Drosophila*, however, Toll signaling is initiated by pathogen recognition through peptidoglycan recognition proteins (e.g., PGRP-SA) and glucan-binding proteins (e.g., GNBP1), which trigger a serine protease cascade culminating in the cleavage of Spz by SPE or other proteases (e.g., Easter), enabling Spz to bind and activate Toll (Valanne et al, 2022). Although we established an essential role for Spz/Toll signaling in ISC proliferation and tumor growth, we did not examine whether canonical microbial sensors (PGRP-SA, GNBPs) or proteases (SPE, Easter) are expressed or active in the midgut under these conditions. While we hypothesize that *Spz* activation may occur independently of microbial triggers in these contexts, the molecular identity and distribution of the upstream factors initiating this cascade remain uncharacterized in our dataset. Future studies aimed at dissecting the proteolytic

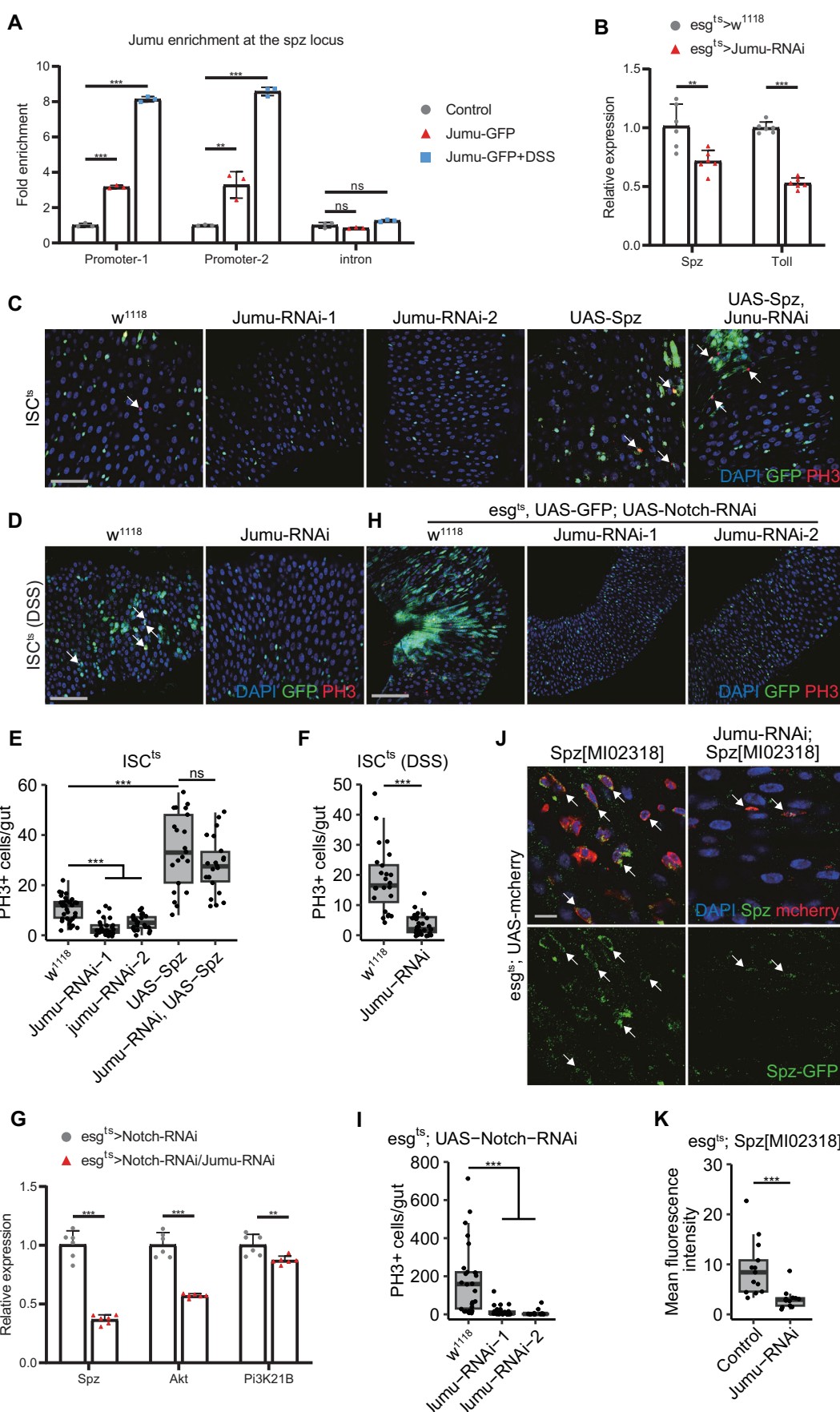

**Figure 7. *Jumu* regulates stem cell proliferation and tumor growth through the Toll pathway.**

(A) CUT&Tag-qPCR analysis at the *Spz* locus in the midgut of *Jumu-GFP* flies with or without 3% DSS, compared with *w[1118]* controls. (B) RT-qPCR analysis of *Spz* and *Toll* in the midgut of *esg[ts]* > *w[1118]* and *esg[ts]*>*Jumu-RNAi* flies at 29 °C for 10 days. $n = 6$ (three independent biological replicates with two technical replicates each). (C) Representative images of the posterior midgut of flies with indicated genotypes at 29 °C for 10 days. PH3 staining (red), Nuclei (blue), and ISC (green). The white arrows show PH3-positive cells. Scale bar: 50 µm. (D) Representative images of the posterior midgut of flies with indicated genotypes with 3% DSS treatment at 29 °C for 1 day (after 10 days of normal food). PH3 staining (red), Nuclei (blue), and ISC (green). The white arrows show PH3-positive cells. Scale bar: 50 µm. (E) Quantification of PH3-positive cells per adult midgut of the indicated genotypes at 29 °C for 10 days. $n = 32, 28, 24, 21, 22$ (from left to right). (F) Quantification of PH3-positive cells per adult midgut of the indicated genotypes with 3% DSS treatment at 29 °C for 1 day (after 10 days of normal food). *w[1118]*, $n = 24$; *Jumu-RNAi*, $n = 31$. (G) RT-qPCR analysis of *Spz*, *Akt*, and *Pi3K21B* in the midgut of *esg[ts]*>*Notch-RNAi* and *esg[ts]*>*Notch-RNAi/Jumu-RNAi* flies at 29 °C for 8 days. $n = 6$ (three independent biological replicates with two technical replicates each). (H) Representative images of the posterior midgut of flies with indicated genotypes at 29 °C for 8 days. PH3 staining (red), Nuclei (blue), and *esg* > *GFP* (green). Scale bar: 100 µm. (I) Quantification of PH3-positive cells per adult midgut of the indicated genotypes at 29 °C for 8 days. $n = 26, 34, 15$ (from left to right). (J) Representative images of the midgut of flies with indicated genotypes at 29 °C for 10 days. Nuclei (blue), *Spz-GFP* (green), and *esg>mCherry* (red). The white arrows show GFP-positive ISCs. Scale bar: 10 µm. (K) Quantification of *Spz-GFP* mean fluorescence intensity in the posterior midgut of flies with indicated genotypes at 29 °C for 10 days. $n = 13$ independent biological samples. Data information: In (A, B, G), data are presented as mean ± SEM. In (E, F, I, K), box plots indicate median (center line), 25th–75th percentiles (box) and minima/maxima within 1.5 × interquartile range (whiskers); outliers are shown as individual points. Statistical significance in (A, B, E–G, I, K) was determined using a two-tailed unpaired *t*-test (ns no significant difference; *$p < 0.05$; **$p < 0.01$; ***$p < 0.001$). Exact *p* values are provided in Table EV1. Source data are available online for this figure.

machinery and upstream recognition events in intestinal cells will be essential to fully understand how Spz/Toll signaling is initiated during regeneration and tumorigenesis.

Notably, we identify the transcription factor *Jumu* as a direct upstream activator of *Spz* in ISCs. Previous studies have established *Jumu* as a stem cell-specific transcription factor under homeostatic conditions in *Drosophila* (Doupé et al, 2018; Dutta et al, 2015). While Doupé et al. observed a modest increase in GFP+ ISCs in the posterior midgut upon *Jumu* RNAi, our whole-midgut analysis showed that *Jumu* knockdown reduced PH3+ ISCs, impairing proliferation during homeostasis, regeneration, and tumorigenesis. This discrepancy may stem from variations in the tissue regions examined (posterior hindgut vs. entire midgut) and the quantification methodologies employed (localized sectional counts vs. whole-midgut assessment). Importantly, we demonstrate that *Jumu* directly binds the *Spz* promoter to drive its expression, thereby activating the Toll/PI3K/Akt signaling cascade and regulating ISC proliferation and tumor growth. Thus, *Jumu* governs both homeostatic ISC maintenance and stress-induced regenerative proliferation. *Drosophila Jumu* is the ortholog of vertebrate *FOXN4*, which has been implicated in human inflammatory bowel disease. The Jumu/Spz/Toll signaling cascade that mediates stem cell function in intestinal regeneration remains to be fully elucidated in both *Drosophila* and mammalian models. Our studies uncover a non-immune role for Spz/Toll signaling in regulating stem cell activity through PI3K/Akt in *Drosophila*. Whether this novel FOXN4/TLR/PI3K/Akt signaling cascade is conserved in mammals requires further investigation.

## Methods

### Reagents and tools table

| Reagent/resource | Reference or source | Identifier or catalog number |
|---|---|---|
| **Experimental models** | | |
| Adh-Gal4, UAS-GFP | Telemann lab | N/A |
| MyoIA-GAL4, UAS-GFP, Tub-GAL80[ts] | Bruce Edgar | Jiang et al, 2009 |

| Reagent/resource | Reference or source | Identifier or catalog number |
|---|---|---|
| esg-Gal4, Su(H)-Gal80, UAS-GFP, Tub-Gal80[ts] | Heinrich Jasper | Rodriguez-Fernandez et al, 2019 |
| esg-GAL4; Tub-GAL80[ts], UAS-GFP | Bruce Edgar | Jiang et al, 2009 |
| esg[ts]; UAS-Apc-RNAi, UAS-Ras[v12] | Michael Boutros | N/A |
| UAS-PGRP-SA | This study | N/A |
| UAS-myr-Akt | Xinhua Lin | N/A |
| Spz[MI02318] | Bloomington *Drosophila* Stock Center | 34313 |
| Toll[MI01254] | Bloomington *Drosophila* Stock Center | 36134 |
| Dif-GFP | Bloomington *Drosophila* Stock Center | 42673 |
| dorsal-GFP | Bloomington *Drosophila* Stock Center | 42677 |
| PGRP-SA[seml] | Bloomington *Drosophila* Stock Center | 55716 |
| UAS-PGRP-SA RNAi-1 | Bloomington *Drosophila* Stock Center | 60037 |
| UAS-PGRP-SA RNAi-2 | Bloomington *Drosophila* Stock Center | 60037 |
| UAS-Spz RNAi | Bloomington *Drosophila* Stock Center | 34699 |
| UAS-Toll RNAi-1 | Bloomington *Drosophila* Stock Center | 31477 |
| UAS-Toll RNAi-2 | Bloomington *Drosophila* Stock Center | 35628 |
| UAS-Dif RNAi-1 | Bloomington *Drosophila* Stock Center | 29514 |
| UAS-Dif RNAi-2 | Bloomington *Drosophila* Stock Center | 30513 |
| UAS-Pi3K21B RNAi | Bloomington *Drosophila* Stock Center | 36810 |
| UAS-Akt RNAi-1 | Bloomington *Drosophila* Stock Center | 31701 |
| UAS-Akt RNAi-2 | Bloomington *Drosophila* Stock Center | 33615 |
| UAS-Jumu RNAi | Bloomington *Drosophila* Stock Center | 44117 |
| UAS-Toll[10b] | Bloomington *Drosophila* Stock Center | 58987 |
| UAS-Dif | Bloomington *Drosophila* Stock Center | 22201 |
| UAS-Pi3K21B | Bloomington *Drosophila* Stock Center | 25899 |
| Cactus-GFP | Vienna Drosophila RNAi Center | 318145 |
| UAS-Toll RNAi-1 | Vienna Drosophila RNAi Center | 100078 |

| Reagent/resource | Reference or source | Identifier or catalog number |
| --- | --- | --- |
| UAS-dorsal RNAi | TsingHua Fly Center | THU1126 |
| UAS-Myd88 RNAi | TsingHua Fly Center | THU3533 |
| UAS-Cact RNAi | TsingHua Fly Center | THU0621 |
| UAS-Akt RNAi | TsingHua Fly Center | THU4852 |
| UAS-Jumu RNAi | TsingHua Fly Center | THU5637 |
| **Antibodies** | | |
| Rabbit polyclonal anti-GFP | Proteintech | Cat#50430-2-AP |
| Rabbit polyclonal anti-Phospho-Histone H3 (Ser10) | Cell Signaling Technology | Cat#9701 L |
| Rabbit polyclonal anti-cleaved *Drosophila* Dcp1 (Asp215) | Cell Signaling Technology | Cat#9578S |
| Rabbit monoclonal anti-tri-methyl-histone H3 (Lys4) | Cell Signaling Technology | Cat#9751 |
| Rabbit monoclonal anti-phospho-Akt (Ser473) | Cell Signaling Technology | Cat#4060S |
| Mouse monoclonal anti-cactus | Developmental Studies Hybridoma Bank | Cat#3H12 |
| Mouse monoclonal anti-dorsal | Developmental Studies Hybridoma Bank | Cat#7A4 |
| Goat anti-rabbit IgG (H + L) secondary antibody Alexa 488 | Thermo Fisher Scientific | Cat#A11034 |
| Donkey anti-rabbit IgG (H + L) secondary antibody Alexa 555 | Thermo Fisher Scientific | Cat#A31572 |
| Goat anti-rabbit IgG (H + L) secondary antibody Alexa 546 | Thermo Fisher Scientific | Cat#A11035 |
| **Oligonucleotides and other sequence-based reagents** | | |
| qPCR Akt forward | CTTTGCGAGTATTAACTGGACAGA | |
| qPCR Akt reverse | GGATGTCACCTGAGGCTTG | |
| qPCR Def forward | CTTCGTTCTCGTGGCTATCG | |
| qPCR Def reverse | ATCCTCATGCACCAGGACAT | |
| qPCR Dif forward | CGGGCATCTACCAAAAGAAA | |
| qPCR Dif reverse | AGCTTCTTGGCGCACTGTAT | |
| qPCR Drs forward | TTGTTCGCCCTCTTCGCTGTCCT | |
| qPCR Drs reverse | GCATCCTTCGCACCAGCACTTCA | |
| qPCR PGRP-SA forward | ACTACCAAGTGCGTCCCATCC | |

| Reagent/resource | Reference or source | Identifier or catalog number |
| --- | --- | --- |
| qPCR PGRP-SA reverse | TAATATCGTTGAAGTCCAGCTCGTTC | |
| qPCR Pi3K21B forward | AGAGAAGGAATGCGGAGGAGATG | |
| qPCR Pi3K21B reverse | CAGGCAATGGACAGAGCATAGTG | |
| qPCR Rp49 forward | TACAGGCCCAAGATCGTGAAG | |
| qPCR Rp49 reverse | GACGCACTCTGTTGTCGATACC | |
| qPCR Spz forward | GACACCTGGCAGTTAATTGTCA | |
| qPCR Spz reverse | CGAAGTCACAGGGTTGATCCG | |
| qPCR Toll forward | TCCAGACCCAGATCAACTCC | |
| qPCR Toll reverse | TAGCCCAGCGAGCTAATGTT | |
| cut&tag-qPCR promoter-1 forward | CGACTCGACTCGTCTGCTC | |
| cut&tag-qPCR promoter-1 reverse | GCTCACACTCTCGCAAGCTA | |
| cut&tag-qPCR promoter-2 forward | AGCGGACCGCGATTTTATCA | |
| cut&tag-qPCR promoter-2 reverse | AAGTTTTTCGTGAAGCAGGCG | |
| cut&tag-qPCR intron forward | CTCGAGAGTTGCCTCTAGCTT | |
| cut&tag-qPCR intron reverse | AGGATCGGCTTGCCTTTAGA | |
| **Chemicals, enzymes and other reagents** | | |
| BrightRed Apoptosis Detection Kit | Millipore | Cat#A113 |
| CUT&Tag reagent kit | Ruoyu Biotech | Cat#CUT-01 |
| Dextran sulfate sodium salt | MP Biomedicals | Cat#02160110 |
| HiScript III All-in-one RT SuperMix Perfect for qPCR | Vazyme | Cat#R333-01 |
| LY294002 | Selleck | Cat#S1105 |
| Maxima SYBR Green/ROX qPCR Master Mix | Thermo Fisher Scientific | Cat#K0222 |
| MK-2206 2HCl | Selleck | Cat#S1078 |
| Paraquat | Sigma-Aldrich | Cat#856177 |
| RNA isolater Total RNA Extraction Reagent | Vazyme | Cat#R401-01 |
| Sucrose | Sigma-Aldrich | Cat#S7903 |
| Triton X-100 | Fisher Scientific | Cat#BP151-500 |
| Trypsin | Solarbio | Cat#T1302 |

| Reagent/resource | Reference or source | Identifier or catalog number |
|---|---|---|
| Vectashield antifade mounting medium with DAPI | Vector Labs | Cat#H-1200 |
| **Software** | | |
| Bowtie2 v2.5.3 | https://bowtie-bio.sourceforge.net/bowtie2/index.shtml | |
| ChIPseeker v1.44.0 | https://www.bioconductor.org/packages/release/bioc/html/ChIPseeker.html?utm_source=chatgpt.com/ | |
| ClusterGVis | https://github.com/junjunlab/ClusterGVis | |
| deepTools v3.5.5 | https://test-argparse-readoc.readthedocs.io/en/latest/index.html | |
| DEseq2 v1.48.2 | https://bioconductor.org/packages/release/bioc/html/DESeq2.html | |
| GraphPad Prism v8.0 | https://www.graphpad.com/ | |
| GSEA v4.3.3 | http://www.broadinstitute.org/gsea/index.jsp | |
| Hisat2 v2.2.1 | https://daehwankimlab.github.io/hisat2/ | |
| ImageJ | https://imagej.nih.gov/ij/download.html | |
| MACS2 v2.2.9.1 | https://github.com/macs3-project/MACS | |
| MEGA v11 | https://www.megasoftware.net/ | |
| PANGEA | https://www.flyrnai.org/tools/pangea/web/home/7227 | |
| R | https://www.r-project.org | |
| RStudio | https://www.rstudio.com/ | |
| Trimmomatic v0.39 | http://www.usadellab.org/cms/?page=trimmomatic | |

## *Drosophila* culture and media

Animals were reared at either 18 °C or 29 °C with a 12-h light/dark cycle with 60% humidity. Ten liters of standard fly medium contained: 615.4 ml sugar syrup, 692.3 g corn flour, 92.3 g soy flour, 323 g yeast, 17.5 g methyl-4-hydroxybenzoate, and 76.9 g agar. Animals in the control and experimental groups were transferred to fresh food every 2 days to prevent fungal infection.

## Fly stocks

$w^{1118}$ flies were utilized as the wild-type strain. The following *Drosophila melanogaster* stocks were used in this study: *Adh-Gal4, UAS-GFP* (Telemann lab), *MyoIA-GAL4, UAS-GFP; Tub-GAL80* (Bruce Edgar (Jiang et al, 2009)), *ISC^ts > GFP (esg-Gal4, Su(H)-Gal80, UAS-GFP, Tub-Gal80ts,* Heinrich Jasper (Rodriguez-Fernandez et al, 2019)), *esg-GAL4; Tub-GAL80, UAS-GFP* (Bruce Edgar (Jiang et al, 2009)), *esg^ts; UAS-Apc-RNAi, UAS-Ras^v12* (Michael Boutros), *UAS-PGRP-SA, UAS-myr-Akt* (Xinhua Lin).

The following strains were obtained from the Bloomington Stock Center: *Spz[MI02318]* (BLN34313), *Toll[MI01254]* (BLN36134), *Dif-GFP* (BLN42673), *dorsal-GFP* (BLN42677), *PGRP-SA^seml* (BLN55716), *UAS-PGRP-SA RNAi-1* (BLN60037), *UAS-PGRP-SA RNAi-2* (BLN60037), *UAS-Spz RNAi* (BLN34699), *UAS-Toll RNAi-1* (BLN31477), *UAS-Toll RNAi-2* (BLN35628), *UAS-Dif RNAi-1* (BLN29514), *UAS-Dif RNAi-2* (BLN30513), *UAS-Pi3K21B RNAi* (BLN36810), *UAS-Akt RNAi-1* (BLN31701), *UAS-Akt RNAi-2* (BLN33615), *UAS-Jumu RNAi* (BLN44117), *UAS-Toll^10b* (BLN58987), *UAS-Dif* (BLN22201), *UAS-Pi3K21B* (BLN25899). *Cactus-GFP* (v318145) and *UAS-Toll RNAi-1* (v100078) were obtained from the Vienna Drosophila RNAi Center. *UAS-dorsal-RNAi* (THU1126), *UAS-Myd88-RNAi* (THU3533), *UAS-Cact-RNAi* (THU0621) and *UAS-Akt RNAi* (THU4852) and *UAS-Jumu RNAi* (THU5637) were obtained from TsingHua Fly Center. For in vivo experiments, we utilized female Gal4 driver strains in crosses with male UAS-RNAi or overexpression lines. Female progenies were randomly selected for experimental use.

## Systemic and oral infection experiments

Fly crosses were maintained at 18 °C, and their progeny were collected 3 days after eclosion. For systemic infection experiments, flies were infected by intrathoracic pricking with a needle dipped in a suspension of *S.a* bacteria ($OD_{600} = 0.5$) under light $CO_2$ anesthesia. For oral infection experiments, flies were starved for 3 h and then transferred to cages containing filter paper disks immersed in 5% sucrose solution with either *P.e* or *S.a* bacteria ($OD_{600} = 100$). The *MyoIA^ts* flies were shifted to 29 °C for 10 days to induce the RNAi prior to oral infection. For survival curve analysis, adult flies were transferred every 2 days to freshly prepared cages containing either standard food or contaminated filter paper disks (Zhou and Boutros, 2020). Statistical significance ($p$ values) between two groups was assessed using log-rank (Mantel–Cox) tests in GraphPad Prism 8.

## RT-qPCR

Total RNA was extracted from two whole flies or ten midguts (from female flies) using the RNA isolator Total RNA Extraction Reagent (Vazyme, R401-01). cDNA was synthesized from 1 μg of RNA using the HiScript III All-in-one RT SuperMix Perfect for qPCR (Vazyme, R333-01). qPCR was performed using Maxima SYBR Green/ROX qPCR Master Mix (Thermo Fisher Scientific, K0222) in a QuantStudio 1 Real-Time PCR System. *Rp49* was used as an internal control, and the qPCR data were analyzed by the ΔΔCT method as previously described (Zhou et al, 2021). Primer sequences are listed in the Reagents and tools table.

## DSS and paraquat feeding experiments

All feeding experiments were performed at 29 °C. Empty cages were prepared with filter paper disks containing 200 μl 3% DSS (MP Biomedicals, 02160110-CF) or 5 mM paraquat (Sigma-Aldrich, 856177) diluted in 5% sucrose solution (Amcheslavsky et al, 2009; Mundorf et al, 2019). A 5% sucrose solution was used as the control feeding solution. Adult flies were maintained on standard food at 29 °C for 10 days to induce the RNAi prior to DSS or Paraquat feeding.

## PI3K and Akt inhibitor feeding experiments

All feeding experiments were conducted at 29 °C. Empty cages were prepared with filter paper disks containing 200 μl of either 50 μM of the *PI3K* inhibitor LY294002 (Selleck, S1105) or the *Akt* inhibitor MK-2206 2HCl (Selleck, S1078) diluted in 5% sucrose solution. A 5% sucrose solution served as the control. Adult flies were transferred to inhibitor-containing cages immediately after eclosion and maintained under these conditions until dissection.

## RNA-Seq data analysis

For analysis of the RNA-Seq data from publicly accessible datasets (GEO: GSE128489 and GSE112331), paired-end reads were mapped to dm6 from FlyBase (Larkin et al, 2021) as the reference genome using Hisat2 version 2.2.1 (Kim et al, 2019). Then, FeatureCounts (version 2.0.6) (Liao et al, 2014) was employed for counting reads. Significantly differentially expressed genes (DEGs) ($\log_2$ fold change >1 or <−1 and $p < 0.05$)) were identified using the R package DEseq2. GSEA was performed using the GSEA (version 4.3.3) software (http://www.broadinstitute.org/gsea/index.jsp), and Gene sets for the Toll signaling pathway were obtained from the FlyBase pathway database. Clustering and visualization were performed using ClusterGVis (https://github.com/junjunlab/ClusterGVis). The phylogenetic relationships among the selected species were reconstructed based on the sequences of *Jumu* using the neighbor-joining method in MEGA (version 11).

## CUT&Tag assay

CUT&Tag was performed using the CUT&Tag reagent kit (Ruoyu Biotech, CUT-01) following the manufacturer's instructions. Female flies were collected on the 5th day post-eclosion, and 30 midguts were dissected in ice-cold sterile PBS. The intestinal tissues were segmented into three to four pieces and transferred to low-adhesion microcentrifuge tubes, washed gently with 400 μl of cold PBS to remove debris. For tissue digestion, segments were treated with 500 μl of 0.5% trypsin (Solarbio, T1302) and incubated for 15 min at 29 °C with shaking at 1000 rpm. After brief settling on ice, the supernatant was collected and centrifuged at 500×*g* for 5 min; the pellet was resuspended in 50 μl of cold PBS. This digestion process was repeated two to three times until complete dissociation was achieved. The pooled cell suspensions were filtered sequentially through 100-μm cell strainers. Cell counting was performed using a hemocytometer with 0.4% Trypan blue staining. Next, 5 μl of pre-activated Concanavalin A-coated beads were added to the cells. Cells were resuspended in 100 μl of Dig-wash Buffer (containing digitonin, protease inhibitor and wash buffer) and incubated with primary antibodies against GFP (1:100, Proteintech, 50430-2-AP) and H3K4me3 (1:100, Cell Signaling Technology, 9751) and secondary antibodies in order. A 1:250 dilution of the pAG-Tn5 adapter complex was prepared in Dig-wash Buffer and added to the cells. After resuspension in Fragmentation Buffer (containing MgCl$_2$ and Dig-wash Buffer), the fragmentation reaction was terminated by adding 2.5 μl EDTA to the 50 μl bead mixture. DNA was then extracted using DNA purification magnetic beads. The extracted DNA was then amplified and purified for library construction. The DNA was also amplified for qPCR analysis, and the primer sequences for CUT&Tag-qPCR are listed in the Reagents and tools table. The final library was sequenced on an Illumina NovaSeq 150PE system.

## CUT&Tag data analysis

Raw data were preprocessed using Trimmomatic version 0.39 (Bolger et al, 2014) to remove low-quality reads and clean adapter sequences. The filtered reads were then aligned to the *Drosophila melanogaster* reference genome (dm6 from UCSC) using Bowtie2. Peak calling was performed using MACS2 version 2.2.9.1 with a *q* value threshold of <0.01 (Zhang et al, 2008) and peak annotation was conducted using the R package ChIPseeker (Yu et al, 2015). Enrichment analysis of target genes from both datasets was conducted using the PANGEA tool (Pathway, Network, and Gene set enrichment analysis) (Hu et al, 2023). For visualization of genomic occupancy around peaks (±3 kb flanking TSSs), BAM files were converted to BIGWIG format using the bamCoverage function in deepTools with the "--normalizeUsing RPKM" option, followed by analysis with the 'computeMatrix' and "plotHeatmap" functions.

## Immunofluorescence and TUNEL staining

Midguts were dissected and immersed in 1X phosphate-buffered saline (PBS). The tissues were then fixed in 4% paraformaldehyde (PFA) for 25 min at room temperature. Tissues were subsequently washed four times in 1X PBST (1X PBS containing 0.1% Triton X-100) and incubated for 30 min in blocking solution (1% bovine serum albumin in 1X PBST). The midguts were then incubated with primary antibodies at 4 °C overnight. The midguts were incubated for 2 h with secondary antibodies. Samples were finally washed four times in 1X PBST and mounted in a Vectashield mounting medium containing DAPI (Vector Labs). Primary antibodies used were: rabbit anti-GFP (1:3000, Proteintech, 50430-2-AP), rabbit anti-PH3 (1:800, Cell Signaling Technology, 9701 L), rabbit anti-p-Akt (1:800, Cell Signaling Technology, 4060S), mouse anti-cactus (1:100, Developmental Studies Hybridoma Bank, 3H12), rabbit anti-DCP1 (1:400, Cell Signaling Technology, 9578S), and mouse anti-dorsal (1:100, Developmental Studies Hybridoma Bank, 7A4). Secondary antibodies used: goat anti-rabbit-488 (1:3000, Thermo Fisher Scientific, A11034), donkey anti-rabbit-555 (1:3000, Thermo Fisher Scientific, A31572) and goat anti-mouse-546 (1:3000, Thermo Fisher Scientific, A11030). TUNEL staining was performed using the BrightRed Apoptosis Detection Kit (Millipore, A113) according to the manufacturer's instructions. All images were acquired using a Zeiss LSM 980 confocal laser microscope.

## Quantification of PH3+ cell counts per midgut

The mitotic indices for all specified genotypes (including time points and conditions) were determined by counting dividing cells marked with pH3 staining. For each midgut, all pH3-positive cells across the entire intestine were counted. For most experiments, more than 20 midguts were dissected per genotype/condition as biological replicates. The data are presented as the mean cell number with standard deviation (SD) for midguts from each genotype or condition, based on counts from all biological replicates.

## Quantification and statistical analysis

Statistical analyses were performed using GraphPad Prism (for imaging data; version 8.0) and RStudio (for CUT&Tag and RNA-seq data). For image analysis, data normality was assessed using the Shapiro–Wilk test. For datasets following a normal distribution, comparisons between two groups were performed using a two-tailed Student's *t*-test. For multiple-group comparisons, one-way analysis of variance (ANOVA) was applied, followed by Tukey's post hoc test for pairwise comparisons. The log-rank (Mantel–Cox) test was used to compare survival curves across genotypes. Differential expression of bulk RNA-seq data was evaluated in DESeq2 with $p$ values estimated by the Wald test. Pearson correlation coefficients and significance were calculated using the R function *cor.test*. Gene enrichment analysis was performed using a hypergeometric distribution algorithm to assess the significance of gene enrichment. The results are presented as mean ± SEM. Statistical significance was defined as $p < 0.05$, with significance levels indicated as $*p < 0.05$, $**p < 0.01$, and $***p < 0.001$. Experiments were not blinded.

p-Akt intensity and Spz-GFP mean fluorescence intensity were quantified using ImageJ (version: 1.54 f). For p-Akt, the mean fluorescence intensity of each ISC (GFP+ cell) was measured using the "Mean Gray Value" function, with background correction performed by subtracting the average fluorescence from three to five neighboring regions of interest (ROIs) containing no cells. For Spz-GFP, total fluorescence in the intestines was quantified and normalized to the total area of measurement, with background fluorescence per unit area subtracted. At least ten midguts were analyzed for each genotype. Quantification of p-Akt-positive, TUNEL-positive and Toll-GFP-positive cells in the posterior midgut was performed using confocal microscopy.

## Data availability

CUT&Tag data generated in this study are available in the NCBI GEO repository with the accession number: GSE279642(https://www.ncbi.nlm.nih.gov/geo/query/acc.cgi?acc=GSE279642). The RNA-seq data have been submitted to the Zenodo database (https://doi.org/10.5281/zenodo.17111755 and https://doi.org/10.5281/zenodo.17116751).

The source data of this paper are collected in the following database record: biostudies:S-SCDT-10_1038-S44319-026-00693-9.

## Peer review information

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

## Acknowledgements

We are grateful to XJ. Ma, Z.Guo, P. Ligoxygakis for comments on the manuscript. We would like to thank B. Edgar, A. Telemann, ZZ. Zhai, Developmental Studies Hybridoma Bank (DSHB), Bloomington Drosophila Stock Center, Kyoto stock center and Vienna Drosophila RNAi Center (VDRC) for fly stocks and antibodies. We thank Yansong Xiong, Yanan Hao, and the Analytical Instrumentation Center of Hunan University for assistance in confocal microscopy. The work is supported by National Natural Science Foundation of China (32270890, 82200198, and 82370177), the Department of Science and Technology of Hunan Province (2023JJ0007 and 2023JJ20023), Hunan Furong Program High-Level Health Talent Support Project (20241226039), the Key R&D Program of Hunan Province (2024JK2108), Wu Jieping Medical Foundation Research Special Fund (320.6750.2023-19-30), and Hunan Provincial NSF Key Project (2025JJ30008).

## Author contributions

**Guofan Peng**: Data curation; Formal analysis; Validation; Investigation; Visualization; Methodology; Writing—original draft; Writing—review and editing. **Shichao Yang**: Formal analysis; Validation; Investigation; Methodology. **Yuexia Zhang**: Investigation. **Yu Zhao**: Investigation. **Xiaoyun Huang**: Resources; Investigation. **Shengen Yi**: Resources; Funding acquisition; Investigation. **Lei Gu**: Resources; Investigation. **Ganqian Zhu**: Resources; Funding acquisition; Validation. **Kewei Zheng**: Resources; Investigation. **Huijun Zhou**: Data curation; Supervision; Funding acquisition; Project administration. **Kang Han**: Resources; Supervision; Investigation; Methodology; Writing—original draft; Project administration; Writing—review and editing. **Jun Zhou**: Resources; Formal analysis; Supervision; Funding acquisition; Writing—original draft; Project administration; Writing—review and editing.

Source data underlying figure panels in this paper may have individual authorship assigned. Where available, figure panel/source data authorship is listed in the following database record: biostudies:S-SCDT-10_1038-S44319-026-00693-9.

## Disclosure and competing interests statement

The authors declare no competing interests.

# Expanded View Figures

**Figure EV1.  Toll signaling is not essential for intestinal immunity against infection.**

(A) Lifespan curves of female flies of the indicated genotypes systemically infected with the bacteria *S.a* at 25 °C. The *Adh-Gal4* driver induces gene expression in the fat body. *w^1118^*, n = 108; *Dif-RNAi*, n = 103. (B) Lifespan curves of female flies of the indicated genotypes systemically infected with the bacteria *S.a* at 18 °C. The *Adh-Gal4* driver induces gene expression in the fat body. *w^1118^*, n = 108; *Relish-RNAi*, n = 126. (C) Lifespan curves of female flies of the indicated genotypes orally infected with the bacteria *S.a* at 29 °C. The *Myo1A-Gal4* driver induces gene expression in the ECs. *w^1118^*, n = 154; *Dif-RNAi*, n = 190. (D) Lifespan curves of female flies of the indicated genotypes orally infected with the bacteria *S.a* at 29 °C. The *Myo1A-Gal4* driver induces gene expression in the ECs. *w^1118^*, n = 154; *Relish-RNAi*, n = 211. (E) Lifespan curves of female flies of the indicated genotypes orally infected with the bacteria *S.a* at 29 °C. The *esg-Gal4* driver induces gene expression in ISCs/EBs. *w^1118^*, n = 126; *Dif-RNAi*, n = 133. (F) Lifespan curves of female flies of the indicated genotypes orally infected with the bacteria *S.a* at 29 °C. The *esg-Gal4* driver induces gene expression in ISCs/EBs. *w^1118^*, n = 126; *Relish-RNAi*, n = 128. (G) Summary of statistics of the lifespan shown in (A–F). (H) RT-qPCR analysis of *Dif* and its target *Drs* in the flies with indicated genotypes upon *S.a* infection at 25 °C for 24 h. n = 6 (three independent biological replicates with two technical replicates each). (I) Gene set enrichment analysis (GSEA) revealed an enrichment in the Toll signaling pathway in the intestine upon *P.e* infection at 4 and 16 h post-challenge. (J) RT-qPCR analysis of *Toll* in *ISC^ts^ > w^1118^* flies with or without *P.e* infection at 29 °C for 1 day (after 10 days of normal food). n = 6 (three independent biological replicates with two technical replicates each). (K) Heatmap showing expression of a subset of the genes involved in the Toll signaling pathway in the intestine upon *P.e* infection at 4 and 16 h post-challenge (using dataset GSE128489). Heatmap displays log2(FPKM + 1) data scaled using a *Z*-score. Data information: In (H, J), data are presented as mean ± SEM. Statistical significance in (H, J) was determined using a two-tailed unpaired *t*-test (*$p < 0.05$; **$p < 0.01$; ***$p < 0.001$). Exact *p* values are provided in Table EV1.

▶

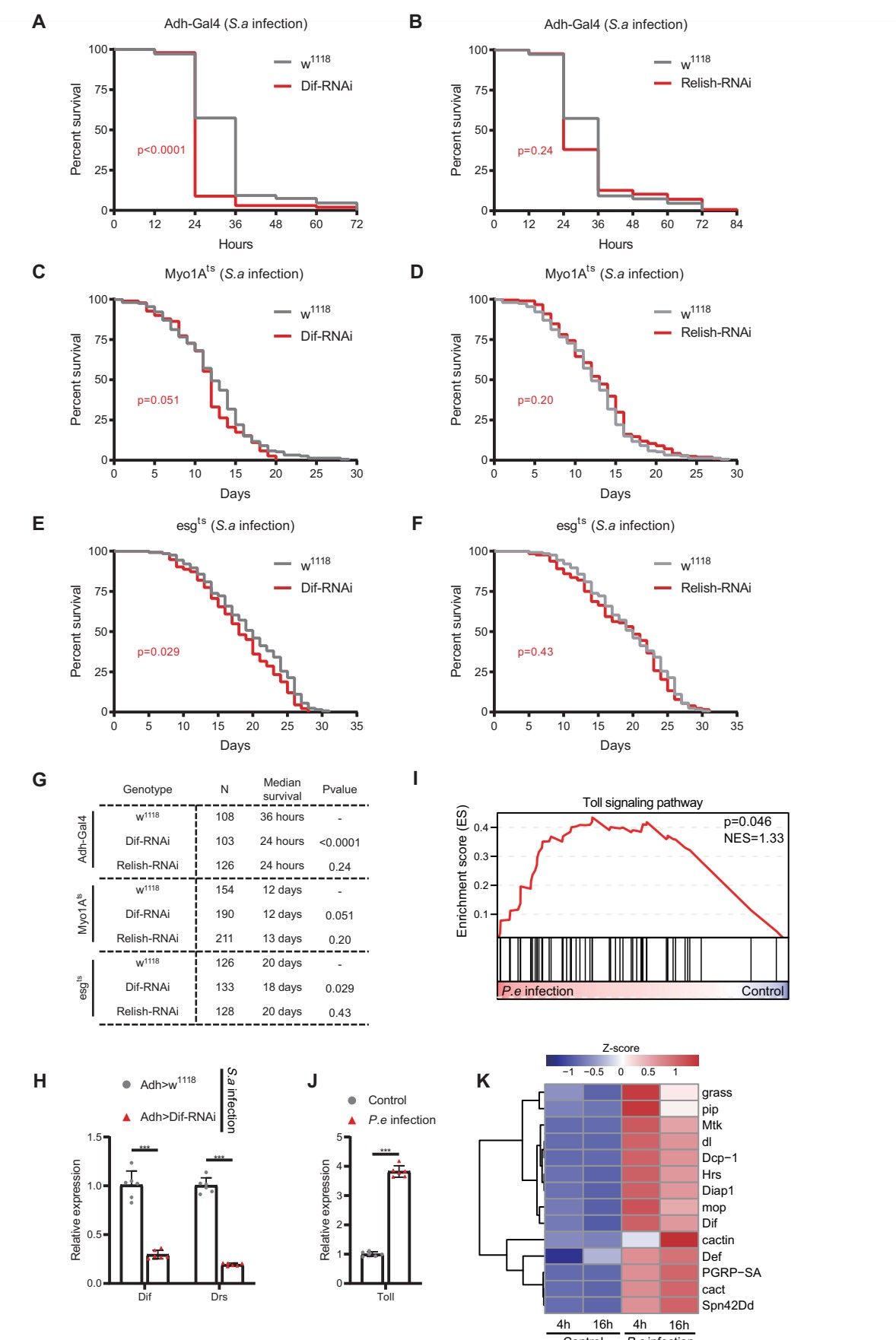

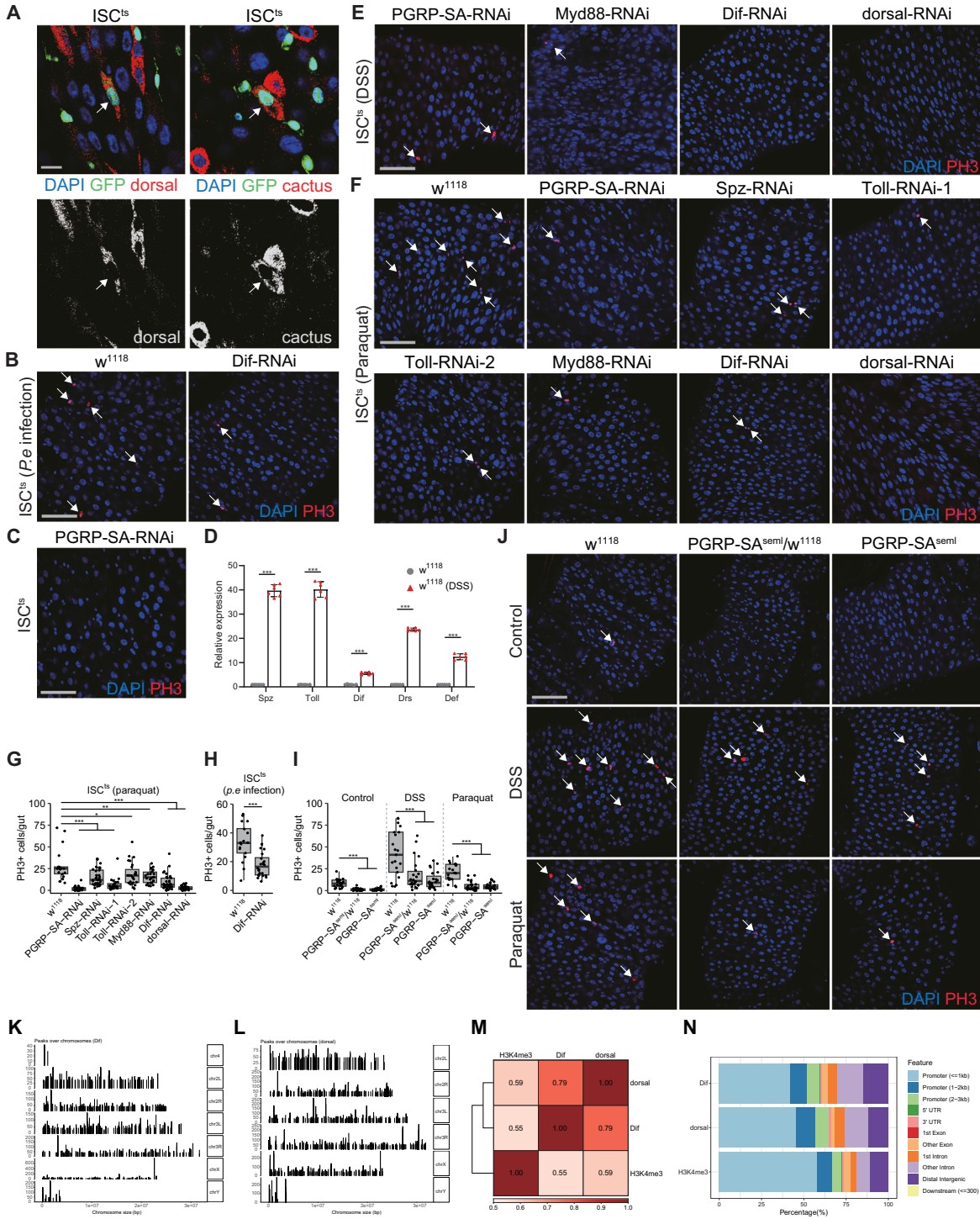

◀ **Figure EV2.  Intestinal regeneration requires the Toll signaling pathway.**

(A) Representative images of the posterior midgut of *ISC^ts* flies. dorsal or Cactus staining (red), Nuclei (blue), and ISC (green). The white arrows show dorsal-positive or Cactus-positive ISCs. Scale bar: 10 μm. (B) Representative images of the posterior midgut of flies with indicated genotypes upon *P.e* infection at 29 °C for 1 day (after 10 days of normal food). PH3 staining (red), Nuclei (blue). The white arrows show PH3-positive cells. Scale bar: 50 μm. (C) Representative images of the posterior midgut of flies with indicated genotypes at 29 °C for 10 days. PH3 staining (red), Nuclei (blue). The white arrows show PH3-positive cells. Scale bar: 50 μm. (D) RT-qPCR analysis of Toll pathway genes (*Spz*, *Toll*, and *Dif*) and their targets (*Drs* and *Def*) in the midgut of *w^1118* flies with or without 3% DSS treatment at 25 °C for 1 day. n = 6 (three independent biological replicates with two technical replicates each). (E) Representative images of the posterior midgut of flies with indicated genotypes with 3% DSS treatment at 29 °C for 1 day (after 10 days of normal food). PH3 staining (red), Nuclei (blue). The white arrows show PH3-positive cells. Scale bar: 50 μm. (F) Representative images of the posterior midgut of flies with indicated genotypes with 5 mM Paraquat treatment at 29 °C for 1 day (after 10 days of normal food). PH3 staining (red), Nuclei (blue). The white arrows show PH3-positive cells. Scale bar: 50 μm. (G) Quantification of PH3-positive cells per adult midgut of the indicated genotypes with 5 mM Paraquat treatment at 29 °C for 1 day (after 10 days of normal food). n = 17, 28, 36, 26, 27, 25, 36, 29 (from left to right). (H) Quantification of PH3-positive cells per adult midgut of the indicated genotypes upon *P.e* infection at 29 °C for 1 day (after 10 days of normal food). *w^1118*, n = 18; *Dif-RNAi*, n = 24. (I) Quantification of PH3-positive cells per adult midgut of the indicated genotypes with or without 3% DSS or 5 mM Paraquat treatment at 29 °C for 1 day (after 10 days of normal food). n = 21, 28, 24, 21, 22, 23, 20, 28, 23 (from left to right). (J) Representative images of the posterior midgut of flies with indicated genotypes with or without 3% DSS or 5 mM Paraquat treatment at 29 °C for 1 day (after 10 days of normal food). PH3 staining (red) and Nuclei (blue). The white arrows show PH3-positive cells. Scale bar: 50 μm. (K) Distribution of Dif CUT&Tag signals on different chromosomes. (L) Distribution of dorsal CUT&Tag signals on different chromosomes. (M) Heatmap depicts Pearson correlation of Dif, dorsal, and H3K4me3 CUT&Tag signals. (N) Distribution of Dif, dorsal, and H3K4me3 CUT&Tag peaks across different functional genome regions. Data information: In (D), data are presented as mean ± SEM. In (G–I), box plots indicate median (center line), 25th–75th percentiles (box) and minima/maxima within 1.5 × interquartile range (whiskers); outliers are shown as individual points. Statistical significance was determined using a two-tailed unpaired *t*-test (D, G, H) or one-way ANOVA followed by Tukey's multiple comparisons test (I) (*$p < 0.05$; **$p < 0.01$; ***$p < 0.001$). Exact *p* values are provided in Table EV1.

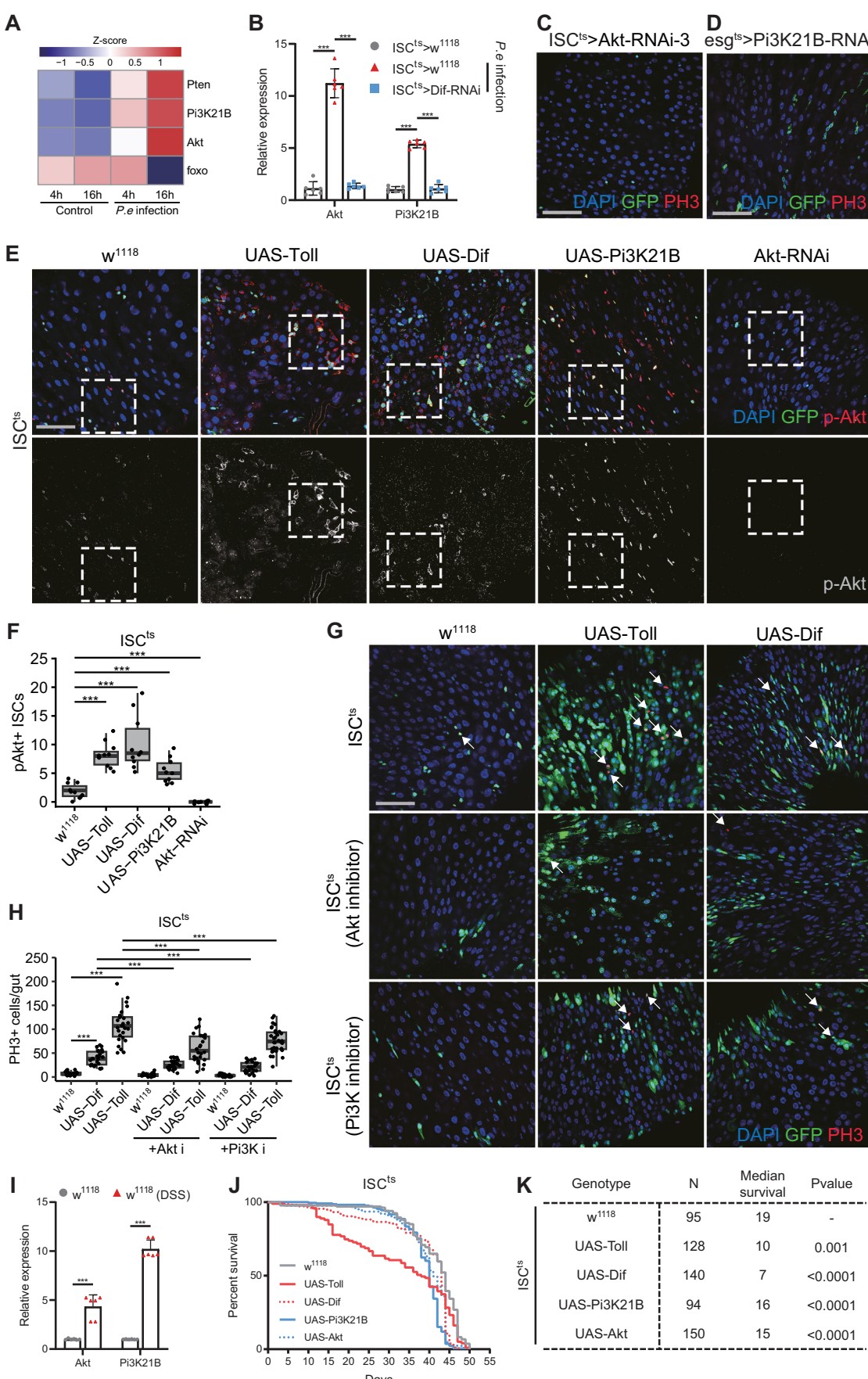

**Figure EV3. Toll signaling promotes ISC proliferation by activating the PI3K/Akt pathway.**

(A) Heatmap showing expression of *Pten*, *Pi3K21B*, *Akt,* and *foxo* in the intestine upon *P.e* infection at 4 and 16 h post-challenge (using dataset GSE128489). Heatmap displays log2(FPKM + 1) data scaled using a *Z*-score. (B) RT-qPCR analysis of *Akt* and *Pi3K21B* in *ISC^{ts} > w^{1118}* and *ISC^{ts}>Dif* flies with or without *P.e* infection at 29 °C for 1 day (after 10 days of normal food). *n* = 6 (three independent biological replicates with two technical replicates each). (C) Representative images of the posterior midgut of flies with indicated genotypes at 29 °C for 10 days. PH3 staining (red), Nuclei (blue), and ISC (green). The white arrows show PH3-positive cells. Scale bar: 50 μm. (D) Representative images of the posterior midgut of flies with indicated genotypes at 29 °C for 10 days. PH3 staining (red), Nuclei (blue), and *esg > GFP* (green). The white arrows show PH3-positive cells. Scale bar: 50 μm. (E) Representative images of the posterior midgut of flies with indicated genotypes at 29 °C for 8 days. p-Akt staining (red), Nuclei (blue), and ISC (green). The white box indicates the region enlarged in Fig. 3D. Scale bar: 50 μm. (F) Quantification of p-Akt-positive ISCs in the posterior midgut of flies with the indicated genotypes after 8 days at 29 °C. *n* = 10 independent biological samples. (G) Representative images of the posterior midgut of flies with indicated genotypes with or without 50 mM Akt or Pi3K inhibitor treatment at 29 °C for 10 days. PH3 staining (red), Nuclei (blue), and ISC (green). The white arrows show PH3-positive cells. Scale bar: 50 μm. (H) Quantification of PH3-positive cells per adult midgut of the indicated genotypes at 29 °C for 10 days. "Akt i" and "PI3K i" denote Akt and PI3K inhibitors, respectively. *n* = 28, 25, 27, 24, 26, 26, 25, 24, 32 (from left to right). (I) RT-qPCR analysis of *Pi3K21B* and *Akt* in the midgut of w^{1118} flies with or without 3% DSS treatment at 25 °C for 1 day. *n* = 6 (three independent biological replicates with two technical replicates each). (J) Lifespan curves of female flies of the indicated genotypes at 29 °C. The number of flies ranged from 94 to 150. (K) Summary of statistics of the lifespan shown in (J). Data information: In (B, I), data are presented as mean ± SEM. In (F, H), box plots indicate median (center line), 25th–75th percentiles (box) and minima/maxima within 1.5 × interquartile range (whiskers); outliers are shown as individual points. Statistical significance in (B, F, H, I) was determined using a two-tailed unpaired *t*-test (*p < 0.05; **p < 0.01; ***p < 0.001). Exact *p* values are provided in Table EV1.

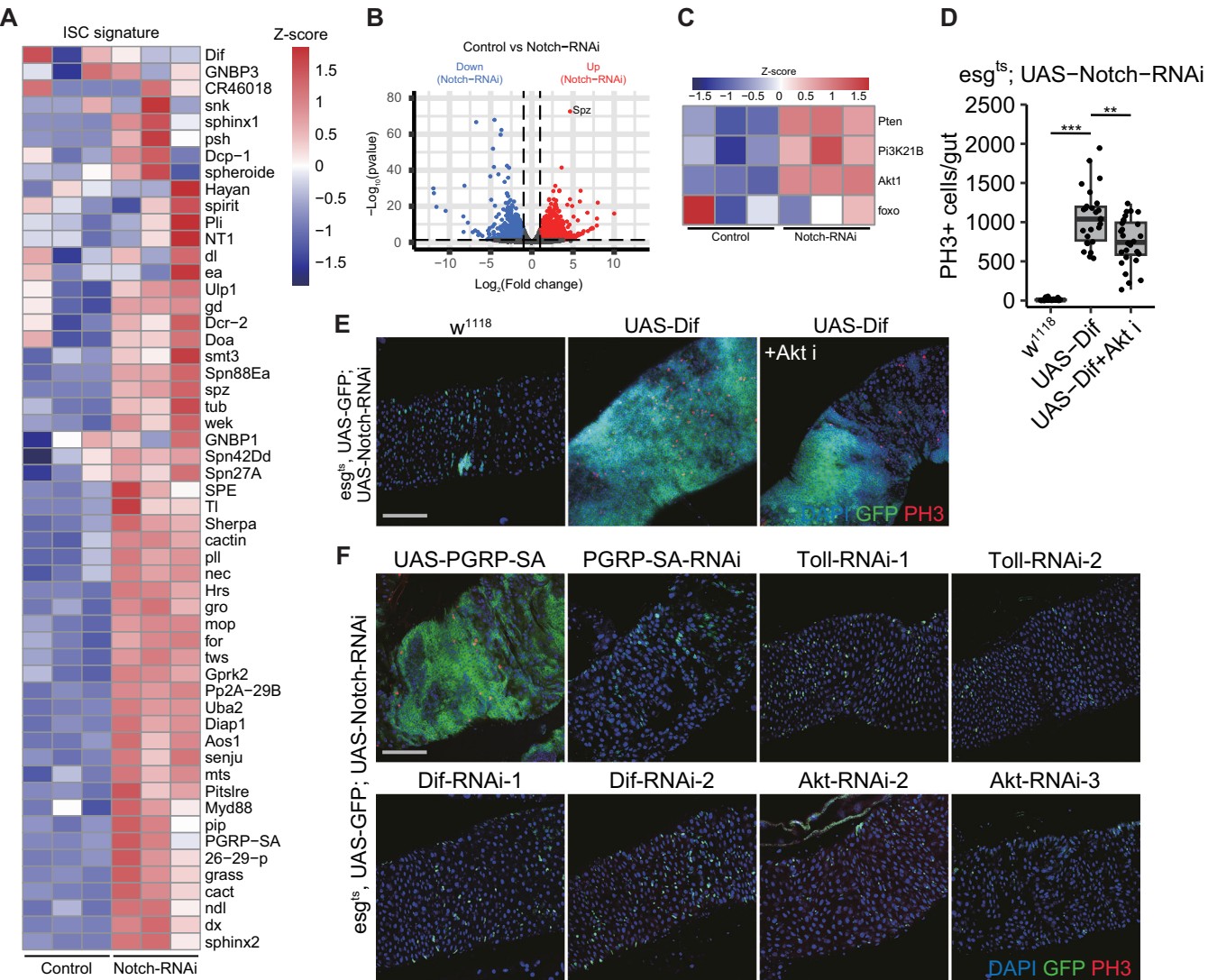

**Figure EV4. Toll signaling promotes intestinal tumor growth via the Pi3K/Akt pathway.**

(A) Heatmap showing expression of Toll pathway genes in ISCs of the Notch-depleted tumor intestine (using dataset GSE112331). Heatmap displays DESeq2 normalized counts scaled using a Z-score. (B) Volcano plot showing differentially expressed genes (blue and red points) identified by RNA-seq in ISCs (Notch-RNAi versus control) (using dataset GSE112331). The most upregulated significant gene (Spz) is shown. (C) Heatmap showing expression of Pten, Pi3K21B, Akt, and foxo in ISCs of the Notch-depleted tumor intestine (using dataset GSE112331). Heatmap displays DESeq2 normalized counts scaled using a Z-score. (D) Quantification of PH3-positive cells per adult midgut of the indicated genotypes at 29 °C for 4 days. $n =$ 33, 26, 28 (from left to right). "Akt i" denotes Akt inhibitor. (E) Representative images of the posterior midgut of flies with indicated genotypes with or without 100 mM Akt inhibitor at 29 °C for 4 days. PH3 staining (red), Nuclei (blue), and esg > GFP (green). Scale bar: 100 μm. (F) Representative images of the posterior midgut of flies with indicated genotypes at 29 °C for 8 days. PH3 staining (red), Nuclei (blue), and esg > GFP (green). Scale bar: 100 μm. Data information: In (D), box plots indicate median (center line), 25th–75th percentiles (box) and minima/maxima within 1.5 × interquartile range (whiskers); outliers are shown as individual points. Statistical significance in (D) was determined using a two-tailed unpaired t-test (*$p < 0.05$; **$p < 0.01$; ***$p < 0.001$). Exact p values are provided in Table EV1.

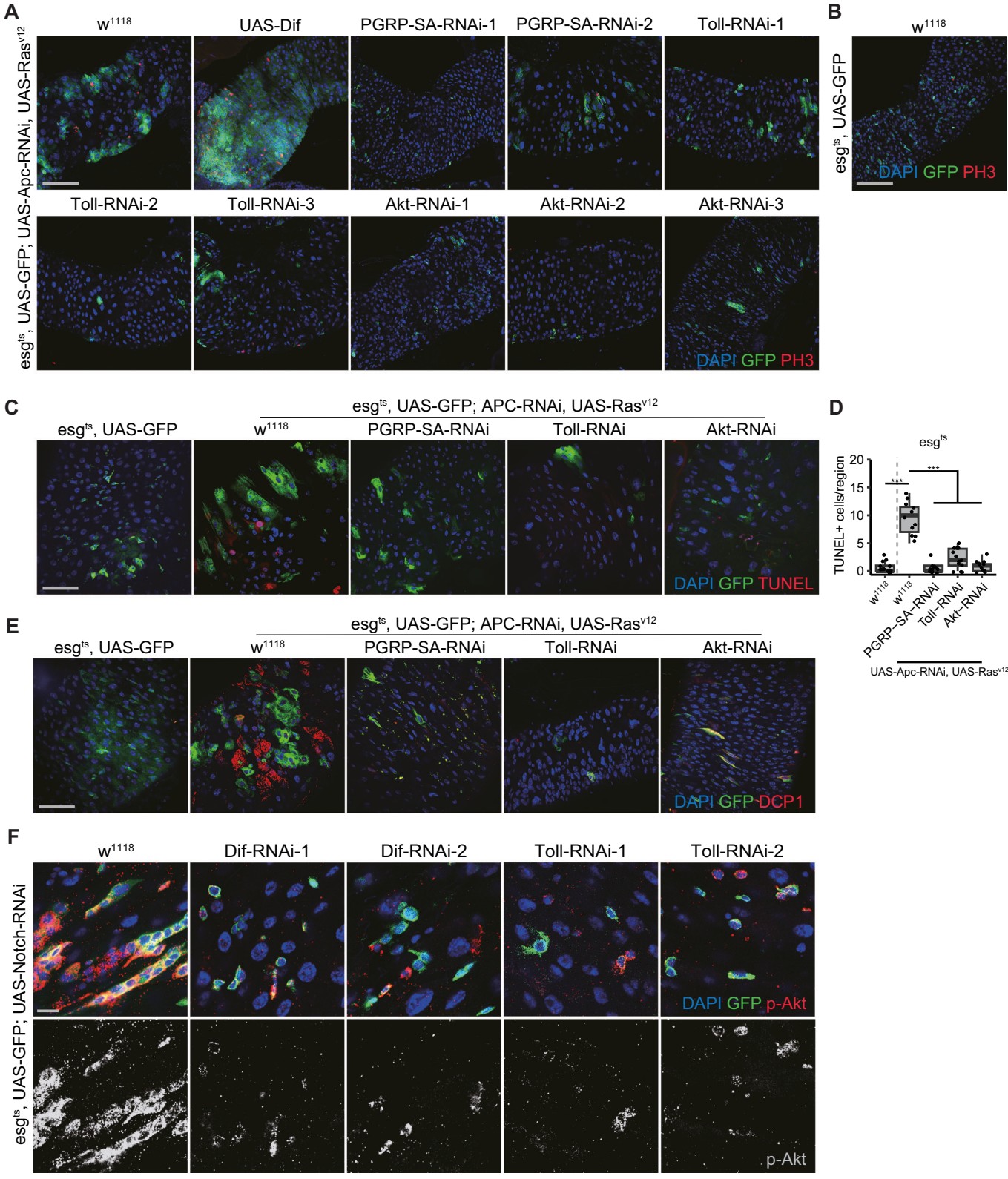

**Figure EV5. Toll signaling regulates intestinal tumor growth via the PI3K/Akt pathway.**

(A) Representative images of the posterior midgut of flies with indicated genotypes at 29 °C for 16 days. PH3 staining (red), Nuclei (blue), and *esg* > *GFP* (green). Scale bar: 100 μm. (B) Representative images of the posterior midgut of flies with indicated genotypes at 29 °C for 16 days. PH3 staining (red), Nuclei (blue), and *esg* > *GFP* (green). Scale bar: 100 μm. (C) Representative images of the posterior midgut of flies with indicated genotypes at 29 °C for 16 days. TUNEL staining (red), Nuclei (blue), and *esg* > *GFP* (green). Scale bar: 50 μm. (D) Quantification of TUNEL-positive cells in the posterior midgut of flies with the indicated genotypes after 16 days at 29 °C. $n = 15$, 11, 12, 13, 12 (from left to right). (E) Representative images of the posterior midgut of flies with indicated genotypes at 29 °C for 16 days. DCP1 staining (red), Nuclei (blue), and *esg* > *GFP* (green). Scale bar: 50 μm. (F) Representative images of the posterior midgut of flies with indicated genotypes at 29 °C for 8 days. p-Akt staining (red), Nuclei (blue), and *esg* > *GFP* (green). Scale bar: 10 μm. Data information: In (D), box plots indicate median (center line), 25th–75th percentiles (box) and minima/maxima within 1.5 × interquartile range (whiskers); outliers are shown as individual points. Statistical significance in (D) was determined using a two-tailed unpaired *t*-test ($^*p < 0.05$; $^{**}p < 0.01$; $^{***}p < 0.001$). Exact *p* values are provided in Table EV1.

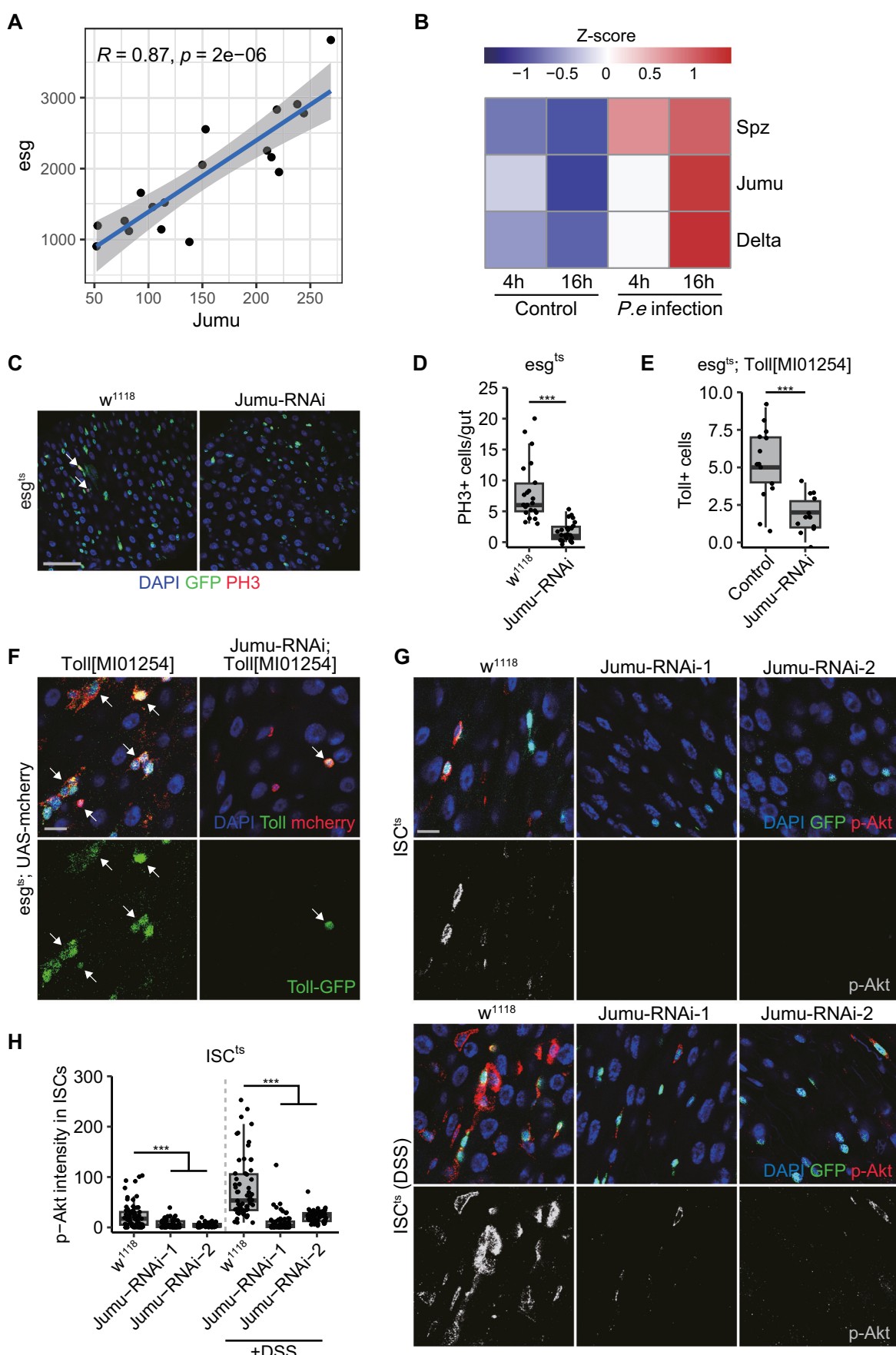

◄ **Figure EV6.  Jumu/Spz/Toll cascade drives stem cell proliferation through Akt.**

(A) Scatter plot showing expression correlation between *esg* and *Jumu*. Each point represents an individual sample from RNA-seq data, with *Jumu* expression on the x-axis and *esg* expression on the y-axis. (B) Heatmap showing expression of *Spz, Jumu* and *Delta* in the intestine upon *P.e* infection at 4 and 16 h post-challenge (using dataset GSE128489). Heatmap displays log2(FPKM + 1) data scaled using a *Z*-score. (C) Representative images of the posterior midgut of flies with indicated genotypes at 29 °C for 10 days. PH3 staining (red), Nuclei (blue), and *esg > GFP* (green). The white arrows show PH3-positive cells. Scale bar: 50 μm. (D) Quantification of PH3-positive cells per adult midgut of the indicated genotypes at 29 °C for 10 days. *w*^*1118*^, n = 22; *Jumu-RNAi*, n = 23. (E) Quantification of Toll-positive cells in the posterior midgut of flies with indicated genotypes at 29 °C for 10 days. *n* = 14 independent biological samples. (F) Representative images of the midgut of flies with indicated genotypes at 29 °C for 10 days. Nuclei (blue), *Toll-GFP* (green), *esg>mCherry* (red). The white arrows show GFP-positive ISCs. Scale bar: 10 μm. (G) Representative images of the posterior midgut of flies with indicated genotypes with or without 3% DSS treatment at 29 °C for 1 day (after 10 days of normal food). p-Akt staining (red), Nuclei (blue), and ISC (green). Scale bar: 10 μm. (H) Intensity statistics of p-Akt in the posterior midgut of flies with indicated genotypes with or without 3% DSS treatment at 29 °C for 1 day (after 10 days of normal food). n = 73, 57, 50, 60, 64, 66 (from left to right). Data information: In (D, E, H), box plots indicate median (center line), 25th–75th percentiles (box) and minima/maxima within 1.5 × interquartile range (whiskers); outliers are shown as individual points. Statistical significance in (D, E, H) was determined using a two-tailed unpaired *t*-test (*p < 0.05; **p < 0.01; ***p < 0.001). Exact *p* values are provided in Table EV1.

