## [Peer Review File · EMBO Reports]

Toll signaling controls stem cell proliferation in intestinal regeneration and tumorigenesis

Guofan Peng, Shichao Yang, Yuexia Zhang, Yu Zhao, Xiaoyun Huang, Shengen Yi, Lei Gu, Ganqian Zhu, Kewei Zheng, Huijun Zhou, Kang Han, and Jun Zhou

Corresponding author(s): Jun Zhou (junzhou82@hnu.edu.cn), Kang Han (hankang1988@hnu.edu.cn), Huijun Zhou (zhouhuijun@hnca.org.cn)

Review Timeline:

Submission Date:	5th Mar 25
Editorial Decision:	7th Apr 25
Revision Received:	2nd Sep 25
Editorial Decision:	21st Oct 25
Revision Received:	12th Dec 25
Accepted:	12th Jan 26

Editor: Achim Breiling

Transaction Report:

Dear Prof. Zhou,

Thank you for the transfer of your manuscript to EMBO reports. I have now received the reports from the three referees that were asked to evaluate your study, which can be found at the end of this email.

As you will see, the referees think that these findings are of interest. However, they have several comments, concerns, and suggestions, indicating that a major revision of the manuscript is necessary to allow publication of the study in EMBO reports. As the reports are below, and all the referee concerns need to be addressed, I will not detail them here.

Given the constructive referee comments, I would like to invite you to revise your manuscript with the understanding that the concerns of the referees must be addressed in the revised manuscript and in a detailed point-by-point response. Acceptance of your manuscript will depend on a positive outcome of a second round of review. It is EMBO reports policy to allow a single round of revision only and acceptance of the manuscript will therefore depend on the completeness of your responses included in the next, final version of the manuscript.

- 1) a .docx formatted version of the final manuscript text (including legends for main figures, EV figures and tables), but without the figures included. Figure legends should be compiled at the end of the manuscript text.
- 2) individual production quality figure files as .eps, .tif, .jpg (one file per figure), of main figures and EV figures. Please upload these as separate, individual files upon re-submission.

- 4) a complete author checklist, which you can download from our author guidelines (<https://www.embopress.org/page/journal/14693178/authorguide>). Please insert page numbers in the checklist to indicate where the requested information can be found in the manuscript. The completed author checklist will also be part of the RPF.

- 5) that primary datasets produced in this study (e.g. RNA-seq, ChIP-seq, structural and array data) are deposited in an

appropriate public database. If no primary datasets have been deposited, please also state this in a dedicated section (e.g. 'No primary datasets have been generated and deposited'), see below.

The accession numbers and database should be listed in a formal "Data Availability" section that follows the model below. This is now mandatory (like the COI statement). Please note that the Data Availability Section is restricted to new primary data that are part of this study. This section is mandatory. As indicated above, if no primary datasets have been deposited, please state this in this section

Data availability

8) Regarding data quantification and statistics, please make sure that the number "n" for how many independent experiments were performed, their nature (biological versus technical replicates), the bars and error bars (e.g. SEM, SD) and the test used to calculate p-values is indicated in the respective figure legends (also for EV and Appendix figures). Please also check that all the p-values are explained in the legend, and that these fit to those shown in the figure. Please provide statistical testing where applicable. Please avoid the phrase 'independent experiment', but clearly state if these were biological or technical replicates. Please also indicate (e.g. with n.s.) if testing was performed, but the differences are not significant. In case n=2, please show the data as separate datapoints without error bars and statistics. See also: <http://www.embopress.org/page/journal/14693178/authorguide#statisticalanalysis>

9) Please add scale bars of similar style and thickness to microscopic images, using clearly visible black or white bars (depending on the background). Please place these in the lower right corner of the images themselves. Please do not write on or near the bars in the image but define the size in the respective figure legend.

10) Please also note our reference format:

12) We now use CRedit to specify the contributions of each author in the journal submission system. CRedit replaces the author contribution section. Please use the free text box to provide more detailed descriptions and do NOT provide your final manuscript text file with an author contributions section. See also our guide to authors: <https://www.embopress.org/page/journal/14693178/authorguide#authorshipguidelines>

13) All Materials and Methods need to be described in the main text using our 'Structured Methods' format, which is required for

all research articles. According to this format, the Methods section should include a Reagents and Tools Table (listing key reagents, experimental models, software, and relevant equipment and including their sources and relevant identifiers), uploaded as separate file, and a Methods section in which we encourage the authors to describe their methods using a step-by-step protocol format with bullet points, to facilitate the adoption of the methodologies across labs. More information on how to adhere to this format as well as downloadable templates (.doc) for the Reagents and Tools Table can be found in our author guidelines (section 'Structured Methods'):

14) Please order the sections like this, using (only) these names:

Title page - Abstract - Keywords - Introduction - Results - Discussion - Methods - Data availability section - Acknowledgements (including the funding information) - Disclosure and Competing Interests Statement - References - Figure legends - Expanded View Figure legends

15) Please make sure that all the funding information is also entered into the online submission system and that it is complete and similar to the one in the acknowledgement section of the manuscript text file.

Please note that corresponding authors are required to supply an ORCID ID upon submission of a revised manuscript and an institutional e-mail address. Please make sure that co-corresponding author Han Kang provides an ORCID in the submission system. Please find instructions on how to link the ORCID ID to the account in our manuscript tracking system in our Author guidelines: <http://www.embopress.org/page/journal/14693178/authorguide#authorshipguidelines>

I look forward to seeing a revised form of your manuscript when it is ready.

Yours sincerely,

Referee #1:

In this manuscript, Peng et al. demonstrate the involvement of the Toll pathway in the regulation of proliferation of the adult intestinal stem cells of *Drosophila melanogaster*. They show that this requirement happens in resting conditions but, most clearly, during regeneration in response to infection or chemical damage, and during tumour growth of two genetic origins (loss of Notch or simultaneous excess of Wnt and MAPK signalling). In the case of tumours, they show that the contribution of this pathway accelerates disease progression towards the death of the host. The authors propose that, in this function, Toll, activated by Spatzle, results in the activation of Dif and Dorsal, which activate the expression of PI3K and Akt, which in turn promote proliferation. The connection between Spz/Toll/Dif-Dorsal and PI3K/Akt seems preserved in the different contexts (homeostasis, regeneration, tumour growth). Finally, the authors provide evidence that Jumu, a transcription factor that increases expression after tissue damage, but that is also overexpressed in tumours, activates in turn the expression of Spz, providing a trigger for the Toll pathway to respond to damage/tumours.

The manuscript presents abundant data, sufficiently well reported, and the main message is well supported. I only have one major issue (one that requires additional experiments), and a few minor points that I think can be addressed in writing only. I must mention that the quality of the English is not ready for publication, and this must be addressed too.

Major:

The justification of jumu as part of the pathway hinges on its requirement for expression of spz. While the phenotypic effect of jumu RNAi on proliferation and tumour expansion seems clear, the reduction of spz mRNA observed with qPCR (a mere 2-fold)

seems not enough to justify the biological effects observed. Maybe jumu is doing something else entirely. I would suggest that this expression effect is tested with spz[MI02318] and that an epistasis analysis is made - the overexpression of Spz should render the knock-down of jumu ineffectual.

Minor:

- The most important non-experimental matter that I need to point out is that the authors seem to overlook that Spatzle needs to be processed in order to signal. The authors explicitly state this, true, but they do not address it in any way. There are multiple proteases between PGRP-SA and Spatzle that are required for Spatzle to bind Toll, most famously Easter and/or Spatzle-processing enzyme (SPE). The authors make no effort to show that these enzymes are expressed in the tissue in the circumstances they analyse. The same goes for PGRP-SA. Moreover, in the discussion, the authors state "We ... provide evidence that PGRP-SA and Spatzle might function as extracellular insulators that induce Toll receptor activity in a microorganism independent manner." But there is not enough data to make this claim, as the experiments have not been performed in axenic conditions. Rather, the molecular identity and tissue distribution of the trigger for the activation of Spatzle remains unexplored in this work -- which is fine, but should be acknowledged. So, I do not wish to request additional experiments, but I feel the authors should state candidly in a paragraph of the discussion what the situation is.

- Towards the end of the introduction, the authors state that enteroblasts can differentiate into enterocytes and enteroendocrine cells. This is incorrect. Enteroendocrine cells arise from pre-enteroendocrine cells, which is a separate, short-lived precursor. Please see Zeng and How, 2015 (10.1242/dev.113357).

- An earlier report than those cited in the manuscript for the role of Insulin signalling in controlling proliferation of the fly intestinal stem cell is Biteau, Karpac et al., 2010 (10.1371/journal.pgen.1001159).

- The authors refer to a transgene as "Su(H)-Gal80ts". Presumably, they mean Su(H)-Gal80 combined with tub-Gal80ts. This should be explained. Also, it should be explained when the tissues were extracted at the permissive or non-permissive temperature.

- What is Toll[10B] ? Presumably, a constitutively active mutant?

- At the end of section "Toll receptor mediated Pi3K/Akt pathway to control stem cell activity" (it would be handy to have line numbers, by the way), the authors conclude that "...Toll pathway controls the expression and activity of PI3K and Akt..." but in figure 3E-H, the epistasis is only shown for Akt. I would rephrase this as "...Toll pathway controls the expression and activity of Akt and, given the molecular data from figure 2, probably PI3K as well..." or something to that effect.

- Figure 4C - why is this heatmap showing only 9 genes? I would expect it to contain all the Toll signalling pathway genes from figure 4B, at least. If these 9 genes are particularly informative, then this should be justified separately, maybe in a supplementary table that indicates why these genes, and not all the others that are part of the Toll pathway or typical downstream genes of Toll or Akt/PI3K. For instance, why are PTEN and Foxo not mentioned here or anywhere else in the paper? Why are not mentioned as readouts typical downstream genes of Dif/Dorsal (e.g. AMPs)?

- Figure 4I - the authors state that "...Toll or Dif inhibition significantly suppressed the p-Akt activity in intestinal tumor (Fig. 4H-I)." However, while the phenotype of suppression of proliferation in Fig 4H is clear, I am not equally convinced that this reduces phospho-Akt. I see that, with the reduction of the number of GFP+ cells, the territory where p-Akt signal is found is also reduced. But the expectation is that the intensity of the phospho-Akt antibody should go down in individual cells. This requires a quantification of the antibody signal, with a proper normalisation approach (not difficult, but not trivial either).

- The authors do not cite previous work done on jumu in intestinal stem cells: Doupé et al., 2018 (10.1073/pnas.1719169115), where the authors define it as a stem cell-specific TF and show that its knock-down in the esg+ population leads to a slight increase in GFP+ labelled cells (which is the opposite of what you would expect according to the model proposed here, albeit with much less data to support the observation). Further, you characterise jumu as a "stress-inducible factor" but jumu has been identified as a stem-cell specific factor in homeostatic conditions in the work of Doupé et al. (2018) as well as earlier in Dutta et al., 2015 (10.1016/j.celrep.2015.06.009). This is overlooked in the discussion.

- The section entitled "jumu is essential for spz expression to potentiate Toll activation during regeneration" does not show what the title says, only correlation. This seems a title for the following section.

Comments on Methods:

- The methods say, regarding infection experiments: "... statistical significances ... between two groups were performed using Gehan-Breslow-Wilcoxon tests...". Can the authors justify why this test was chosen, when survival analyses are usually performed using Log-rank tests (with the most relaxed assumptions)?

- The 'general' statistical analysis is described as two-tailed student's tests, but no mention is made of tests of the assumption

(normality), like Shapiro-Wilk tests. Are the authors certain that this is the appropriate test for their analyses. Also, it is not clear whether this covers everything that is not bioinformatics and the infection survival - but there is also the survival analyses for the tumour effects, whose statistical analysis is not described.

- The description of the Cut and Tag method, which is not routine in the *Drosophila* midgut, is not at the same level of detail compared with the description of the histofluorescence (which is more established). The paper would be more valuable to the community if this was described more accurately, especially aspects that are not in any manufacturer's instructions (e.g. single cell suspension or the exact composition, with catalog numbers, of the dig-wash buffer, timings etc).

Comments on Figures:

- Heatmaps in S1G, S4A, 4C- what are the numerical units? Where is the data coming from? Besides referencing it, it would be helpful to have the samples described in the legend at least, if not the main text where this is referenced.

- Figure 6I - the legends says that 'jumu peaks are highlighted in orange' but one can see several other peaks that are not highlighted. Why is that? The criterion for what is a peak seems unexplained.

- Figure 5A,C, please indicate numbers of flies per curve (or a range for all).

- Coexpression panels from Figures 1A, 3B and 4I should have the different channels shown in separate panels, in addition.

Comments on Writing:

- I am sorry to say this but the manuscript has numerous points of ambiguity and lack of clarity, as well as minor grammatical and punctuation mistakes, and formatting errors -- too numerous for me to single them out here. As a non-native speaker myself, I can only sympathise, but for the sake of the reader, this is an important necessary improvement. In general, I can point out at some persistent errors:

* Very often the 's' for the 3rd person singular in the verb is missing.

* In the introduction it is difficult to follow which statements belong to which organism or research system. This should be made explicit.

* Formatting of genes and transgenes (capitalisation, use of italics) does not follow conventions. Notch, jumu and spz are cases in point. Another are UAS-X transgenes, which should be in italics.

- Specific examples of imprecise meaning:

* In the main text, where Figure 3EF is referred to, the phenotype is described as "stem cell depletion", but the figures are supposedly depicting the number of mitotic ISCs.

* "colorectal tumorigenesis, renal fibrosis and other neurodegenerative diseases" -- "others" seems out of place here.

* "We ... examined the expression ... of Toll and PI3K/Akt dependent gene expression..." -- why 'dependent'?

* The section title "The proactive role of Toll/Akt/PI3K pathway in tumor growth" is vague. What does proactive mean? I would suggest the authors just say what the section shows - that the proliferative capacity of the tumours seen in the previous section correlates with disease progression.

Referee #2:

Summary

In this paper, the authors show a non-immune function of Spz/Toll signaling in intestinal stem cell proliferation in homeostasis, regeneration and tumour-like pathophysiological conditions. Toll signaling drives intestinal stem cell proliferation by regulating the transcription of insulin signalling targets, PI3K and Akt. Through molecular and bioinformatic analysis, the authors show that NF- κ B factors Dif and Dorsal are transcriptionally regulating the expression of PI3K and Akt downstream of Toll signaling in an insulin-independent manner. Further, epistatic studies, show that downregulating Toll activation or Akt activation can rescue the proliferation of intestinal tumors and improve the lifespan of the animals. Transcriptomic and genetic analysis suggest that the stress-responsive factor Jumu regulates the expression of Spz, and inactivation of the Jumu/Spz/Toll cascade reduces intestinal regeneration as well as tumorigenesis .

This represents an intriguing non-immune role of Toll signaling in adult tissue homeostasis in *Drosophila*. Several pertinent questions have to be addressed to consolidate the results.

Major comments

1. Figure 1D and E: Apart from the effect on ISC proliferation, is there a role for the studied genes in other cell types (EC and EE) in the intestine? Is it the survival and/or the differentiation of ISCs affected?

2. In S3C, expression of p-akt can be also seen in non GFP cells. Are these enterocytes? If so, have the authors checked whether there is a contribution of these cells in Toll mediated stem cell proliferation?

3. In figure S5D *esg>GFP*; Notch RNAi looks like wildtype gut with minimal to no proliferation whereas and figure 4D *esg>GFP*;

Notch RNAi shows gut with more GFP cells and more ph3 incorporation. Why is this variability? Same is visible in the respective ph3 quantitation as well.

4. Figure S5 E: Even though there is less ph3 incorporation, the nuclei number also seems to be higher in akt-RNAi and toll-RNAi combined with Apc, rasv12 compared to the Apc,rasv12 only guts. What is the status of cell death in these genetic backgrounds?

5. Figure S5 E: Why is there significant loss of stem cell specific GFP expression in akt-RNAi and toll-RNAi. An image of a wildtype gut also should be incorporated for reference in all these analyses.

6. Does blocking systemic insulin has any impact onto toll mediated activation of Akt and Pi3K and further tumorigenesis?

7. Does overexpression of akt or toll independently of tumors have an impact on the life span of animals?

8. Does overexpression of toll in Jumu loss of function rescues stem cell proliferation in homeostatic condition?

9. Does overexpression of Jumu induce hyper-phosphorylation of Akt as observed in UAS-Dif and UAS-toll?

Minor comments

1. There are grammatical and spelling errors throughout the manuscript. Please rectify/edit.

2. S2A Dif GFP expression and Figure 1A cactus antibody staining looks nonspecific. Better representation is needed.

Referee #3:

In this manuscript, Peng, Zhou and colleagues report data showing that the Spz/Toll/Dorsal signaling pathway, best known for its functions in innate immunity, also impacts Intestinal Stem Cell (ISC) function and regeneration in the *Drosophila* intestine. The analysis is fairly extensive and many of the findings are novel and interesting. Much of the data support the authors' conclusions, but not all, and the quality of the data is variable. One strength is that the paper provides new insights into the non-immune function of Toll signaling, which has traditionally been studied for its role in innate immunity. This research broadens our understanding of Toll receptor functions. Another strength is the identification of Jumu, which appears to regulate Spz expression and subsequently activates Toll signaling. As to detractions, the most serious issue with this manuscript is that the authors' conclusions about Spz/Toll signaling regulating ISCs via the PI3K/AKT pathway are not warranted by the data they present (see below). In addition, there are a number of assays and controls that are not sufficient, and which should be improved in order to meet typical standards in this competitive field (see details below). For these reasons, and despite the fact that Spz/Toll signaling is clearly important in ISCs, we cannot recommend publication of this work in its present form. Some significant issues with the paper are detailed below.

1. First and foremost, the paper has not demonstrated that Spz/Toll promotes ISC divisions by activating PI3K/AKT signaling. The authors make this conclusion based on their observation that the mRNA expression of AKT and PI3K21B (a P60 non-catalytic subunit of PI3K) rises if Dif, a transcription factor downstream of Toll, is overexpressed (Fig 3A). Additionally, Dif overexpression strongly stimulates ISC proliferation, and this effect can be blocked by AKT-RNAi. However, there is ample data in the literature showing that levels of AKT and PI3K-P60 mRNA do not typically determine functional PI3K activity (which is not measured here) and, consistently, the authors also show that overexpression of either AKT or PI3K21B has little if any effect on ISC proliferation. Thus, the data presented here show that Dif is pro-proliferative and that AKT and P60 are required for ISC proliferation, but not (importantly) that PI3K activity is pro-proliferative or that Spz/Toll/Dif regulates PI3K activity. (The pAKT stains shown in Fig 3B, 4I and S4C are quite inconclusive and would need to be quantified to be considered seriously.) The authors need to do their homework on how PI3K/AKT activity is regulated (it's by membrane association and phosphorylation), and what its known effects on ISC proliferation are. Presently, their conclusions drawn about the pathway from the data in Figures 3D-3E and 5C-5D are invalid.
2. The initial characterization of the pathway (Fig 1) is superficial and incomplete. It could be improved by assaying the effects of other components (Dorsal-RNAi, Cactus-RNAi, MyD88-RNAi) with and without DSS treatment, or infection.
3. Image quality: The images in Figure 1 (C, D, E) do not accurately represent their corresponding PH3 counts in panels F and G. The resolution is low and may lead to misinterpretation of the data. In the posterior midgut, are the PGRP-SA-RNAi cells washed off or not present? This needs clarification.
4. The manuscript does not clearly state whether the CUT&Tag data were collected under damage/DSS conditions or homeostatic conditions. Clarification is needed for proper interpretation. Also, the description of the CUT&Tag method in the Results section is inaccurate (it doesn't involve "pull-down" of transcription factors).
5. There is insufficient explanation of Jumu's role: While the data suggesting Jumu regulates Spz expression is compelling, the description of this mechanism is vague. Providing more detail on how Jumu interacts with Spz would strengthen this paper.
6. All imaging data are shown only in merged panels. While this may be acceptable for less critical figures, it becomes problematic when used to obscure complex or difficult-to-explain phenotypes. To ensure scientific transparency, the authors should include separate channel panels for Figures 1A, 1B, 3B, 4I, S2A, and S4C.
7. Figure S2A contains essential data and should be moved to Figure 1A. Additionally, we ask the authors to clarify whether the progenitor-specific expression of Spz, Toll, Dif, Dorsal, and Cactus occurs in all progenitor cells or only a subset. Is this expression pattern consistent throughout the midgut, or confined to a specific region?
8. Although most Toll pathway components are expressed in progenitors, this does not directly reflect pathway activation. The Drosomycin-GFP or -lacZ reporter is a commonly used tool to visualize Toll pathway activation. We recommend that the authors use this, or another validated pathway activity reporter to determine baseline activity in progenitor cells under normal conditions, and whether DSS or infection alters this activity, as proposed. Further, is there any progenitor-to-EC signal shift under stress?

9. While Dif and Dorsal can form heterodimers, they don't need to form a complex for function, as they can also act independently. Dif is known to primarily mediate immune responses and Dorsal is involved in both developmental and immune processes. To better understand the role of Toll signaling in ISCs, we suggest that the authors analyze the loss-of-function effect of Dorsal. This applies to Figures 1C-1H.
10. The Toll pathway is primarily activated by Gram-positive bacteria and fungi, while the IMD pathway responds to Gram-negative bacteria. Using *Staphylococcus aureus* (S.a.) to rule out IMD pathway involvement is inappropriate without direct evidence that IMD responds to S.a. in the *Drosophila* gut. The authors should explain why they used S.a. instead of *Pseudomonas entomophila* (P.e.) in Figs. S1C-1D. Additionally, since the authors propose a progenitor-specific function for the Toll pathway, the EC-specific knockdown in Figure S1B is not suitable to support the claim that "intestinal Toll activity may not be essential for survival." This experiment should be repeated using *esgTS*.
11. In Figure S4A, P.e. is shown to upregulate Pi3K21B and Akt expression. To fully support their proposed model the authors should provide evidence that P.e. (a Gram-negative bacterium) activates Toll signaling, and that this upregulation is dependent on Toll pathway activity.
12. The gene expression analysis in Fig 2 is not very good. An overall summary of the CUT&Tag targets of Dorsal and Dif is not presented, and Fig 2B and 2D are not very useful. Fig 2F seems to be selected data; it would be more helpful to see a graph of all the top GO categories affected.
13. The English grammar is quite poor in many instances, sometimes to the point of obscuring the meaning of the text.
14. Figures 4D and S5D present inconsistent controls. Why does the control in S5D not show a tumor phenotype?
15. Figures 6E-6F lack sufficient methodological detail in the Results section and figure legends. In Figure 6D, most pro-mitotic genes are not upregulated during the damage stages (d1, d4), when regeneration is needed, but increase during the recovery phase (d5, d8), when mitosis should be decreasing to terminate regeneration. Please explain this.
16. In Figure 7A, the authors should use *spz*-GFP and Toll-GFP reporters to confirm whether the downregulation of *spz* and Toll is progenitor cell-autonomous. In Figure 7D, the DSS treatment appears ineffective and should be repeated.
17. Minor point: the expression patterns of *Adh-Gal4* and *Myo1A-Gal4* should be clearly described in the relevant figure legends.
18. Please add information to the figures or the figure legends detailing the timing of the experiments.

Response letter (EMBOR-2025-61477-T)

Referee #1:

In this manuscript, Peng et al. demonstrate the involvement of the Toll pathway in the regulation of proliferation of the adult intestinal stem cells of *Drosophila melanogaster*. They show that this requirement happens in resting conditions but, most clearly, during regeneration in response to infection or chemical damage, and during tumour growth of two genetic origins (loss of Notch or simultaneous excess of Wnt and MAPK signalling). In the case of tumours, they show that the contribution of this pathway accelerates disease progression towards the death of the host. The authors propose that, in this function, Toll, activated by Spatzle, results in the activation of Dif and Dorsal, which activate the expression of PI3K and Akt, which in turn promote proliferation. The connection between Spz/Toll/Dif-Dorsal and PI3K/Akt seems preserved in the different contexts (homeostasis, regeneration, tumour growth). Finally, the authors provide evidence that Jumu, a transcription factor that increases expression after tissue damage, but that is also overexpressed in tumours, activates in turn the expression of Spz, providing a trigger for the Toll pathway to respond to damage/tumours.

The manuscript presents abundant data, sufficiently well reported, and the main message is well supported. I only have one major issue (one that requires additional experiments), and a few minor points that I think can be addressed in writing only. I must mention that the quality of the English is not ready for publication, and this must be addressed too

Response:

We thank the reviewer for his encouragement and suggestions. We have now addressed the comments accordingly and rewrote the text that improved the quality of English for reading.

Major:

The justification of jumu as part of the pathway hinges on its requirement for expression of spz. While the phenotypic effect of jumu RNAi on proliferation and tumour expansion seems clear, the reduction of spz mRNA observed with qPCR (a mere 2-fold) seems not enough to justify the biological effects observed. Maybe jumu is doing something else entirely. I would suggest that this expression effect is tested with spz[MI02318] and that an epistasis analysis is made - the overexpression of Spz should render the knock-down of jumu ineffectual.

Response:

Thank you for your insightful comments. We have conducted additional experiments, including using spz[MI02318] and performing an epistasis analysis in which Spz overexpression was tested in the jumu RNAi background. We found that Jumu knockdown markedly attenuated Spz-GFP signals compared to controls (new Fig. 7J and K). Overexpression of Spz induced excessive ISC proliferation, which was unaffected by simultaneous Jumu knockdown, indicating that Jumu acts upstream of Spz (new Fig. 7C). These results support the role of jumu in regulating spz expression to regulate stem cell proliferation during intestinal homeostasis and regeneration.

Minor:

- The most important non-experimental matter that I need to point out is that the authors seem to overlook that Spätzle needs to be processed in order to signal. The authors explicitly state this, true, but they do not address it in any way. There are multiple proteases between PGRP-SA and Spätzle that are required for Spätzle to bind Toll, most famously Easter and/or Spätzle-processing enzyme (SPE). The authors make no effort to show that these enzymes are expressed in the tissue in the circumstances they analyse. The same goes for PGRP-SA. Moreover, in the discussion, the authors state "We ... provide evidence that PGRP-SA and Spätzle might function as extracellular insulators that induce Toll receptor activity in a microorganism independent manner." But there is not enough data to make this claim, as the experiments have not been performed in axenic conditions. Rather, the molecular identity and tissue distribution of the trigger for the activation of Spätzle remains unexplored in this work -- which is fine, but should be acknowledged. So, I do not wish to request additional experiments, but I feel the authors should state candidly in a paragraph of the discussion what the situation is.

Response:

We sincerely thank the reviewer for highlighting the important point regarding the processing of Spätzle (Spz) and the expression of upstream proteases and recognition proteins in our study. We acknowledge that our manuscript focuses primarily on the downstream effects of Spz/Toll signaling in regulating intestinal stem cell (ISC) proliferation and tumorigenesis, and we did not explicitly address the upstream mechanisms of Spz activation, including the roles of proteases such as Easter or Spz-processing enzyme (SPE) and the expression of PGRP-SA. While our study demonstrates that Spz and Toll signaling components are critical for ISC function in the *Drosophila* intestine, we agree that the molecular details of Spz activation, including the tissue-specific expression of proteases and pathogen recognition proteins like PGRP-SA, remain unexplored in our dataset. To

address this, we have added a new paragraph to the Discussion section to candidly discuss the limitations of our study concerning Spz processing and the upstream activation cascade (Page 12, Lines 355–365)). This paragraph, provided below, incorporates the reviewer’s points and clarifies the need for future investigations into the expression and roles of Easter, SPE, PGRP-SA, and other upstream factors in the *Drosophila* intestine during regeneration and tumorigenesis. We believe this addition strengthens the manuscript by transparently addressing these gaps and aligning with the reviewer’s suggestion for a forthright discussion.

“In *Drosophila*, Toll signaling is initiated by pathogen recognition through peptidoglycan recognition proteins (e.g., PGRP-SA) and glucan-binding proteins (e.g., GGBP1), which trigger a serine protease cascade culminating in the cleavage of Spz by the Spz-processing enzyme (SPE) or other proteases, such as Easter, enabling Spz to bind and activate Toll (Valanne et al, 2022). Our study demonstrates a critical role for Spz/Toll signaling in ISC proliferation during regeneration and tumorigenesis but does not explore the tissue-specific expression or activity of PGRP-SA, GGBP1, SPE, or Easter in the *Drosophila* intestine. While we hypothesize that Spz activation may occur independently of microbial triggers in these contexts, the molecular identity and distribution of the upstream factors initiating this cascade remain uncharacterized in our dataset. Future studies aimed at dissecting the proteolytic machinery and upstream recognition events in intestinal cells will be essential to fully understand how Spz/Toll signaling is initiated during regeneration and tumorigenesis.”

- Towards the end of the introduction, the authors state that enteroblasts can differentiate into enterocytes and enteroendocrine cells. This is incorrect. Enteroendocrine cells arise from pre-enteroendocrine cells, which is a separate, short-lived precursor. Please see Zeng and How, 2015 (10.1242/dev.113357).

Response:

We thank the reviewer for pointing this out. We have corrected the statement in the Introduction (Page 4, Lines 80–83) to clarify that enteroendocrine cells derive from pre-enteroendocrine cells, as demonstrated by Zeng and Hou (2015), while enteroblasts primarily differentiate into enterocytes.

- An earlier report than those cited in the manuscript for the role of Insulin signalling in controlling proliferation of the fly intestinal stem cell is Biteau, Karpac et al., 2010 (10.1371/journal.pgen.1001159).

Response:

We thank the referee for this valuable suggestion and for pointing us to the highly relevant references. We have now cited this indicated reference in the revised manuscript

- The authors refer to a transgene as "Su(H)-Gal80ts". Presumably, they mean Su(H)-Gal80 combined with tub-Gal80ts. This should be explained. Also, it should be explained when the tissues were extracted at the permissive or non-permissive temperature.

Response:

Thank you for pointing this out. The "Su(H)-Gal80ts" (Su(H)-Gal80, tub-Gal80ts) construct indeed combines the Su(H)-Gal80 with tub-Gal80ts (a temperature-sensitive Gal80 variant under the tubulin promoter), as correctly inferred (Rodriguez-Fernandez *et al*, 2019). In ISC^{ts}>GFP (*esg-Gal4, UAS-GFP, Su(H)-Gal80^{ts}*) flies, the Su(H)GBE-Gal80 transgene suppresses Gal4 activity specifically in Su(H)GBE-positive EBs, thereby restricting *esg-Gal4*-driven expression to ISCs while leaving it inactive in EBs. This genetic setup allows ISC-specific manipulation and visualization. We have clarified this and cite the related reference in the Results section (Page 5, Lines 120–123).

- What is Toll[10B] ? Presumably, a constitutively active mutant?

Response:

Thanks for this question. Toll[10B] is a constitutively active gain-of-function mutant of the Toll receptor generated by a cysteine-to-threonine substitution (G2.916→A) in the extracellular domain, which disrupts ligand-independent autoinhibition and leads to sustained activation of downstream signaling pathways (Maxton-Küchenmeister *et al*, 1999; Schneider *et al*, 1991). We have clarified this definition in the revised manuscript (Results section, Page 7, Lines 174–177) and cited relevant literature to contextualize its use in our experimental system.

- At the end of section "Toll receptor mediated Pi3K/Akt pathway to control stem cell activity" (it would be handy to have line numbers, by the way), the authors conclude that "...Toll pathway controls the expression and activity of PI3K and Akt..." but in figure 3E-H, the epistasis is only shown for Akt. I would rephrase this as "...Toll pathway controls the expression and activity of Akt and, given the molecular data from figure 2, probably PI3K as well..." or something to that effect.

Response:

We sincerely appreciate the referee's insightful comment. As suggested, we have revised the statement in the "Toll receptor mediated Pi3K/Akt pathway to control stem cell activity" section (now with line numbers: Page 7, Lines 198–201): "Hence, these results suggest that the Toll pathway

controls the expression and activity of Akt to regulate stem cell proliferation in normal homeostasis and intestinal regeneration, and molecular evidence in Figure 2 suggests that PI3K also functions downstream of the Toll pathway, potentially contributing to similar cellular responses.”

- Figure 4C - why is this heatmap showing only 9 genes? I would expect it to contain all the Toll signalling pathway genes from figure 4B, at least. If these 9 genes are particularly informative, then this should be justified separately, maybe in a supplementary table that indicates why these genes, and not all the others that are part of the Toll pathway or typical downstream genes of Toll or Akt/PI3K. For instance, why are PTEN and Foxo not mentioned here or anywhere else in the paper? Why are not mentioned as readouts typical downstream genes of Dif/Dorsal (e.g. AMPs)?

Response:

Thank you for this comment. In the revised manuscript, we have replaced the original heatmap with a comprehensive version that now includes all Toll signaling pathway genes (SI Appendix, Fig. S6A), along with PTEN and Foxo (SI Appendix, Fig. S6C). PTEN was upregulated, whereas Foxo showed no obvious change. In addition, in the Notch-RNAi dataset (derived from FACS-sorted ISCs), the expression of typical downstream targets of Dif/Dorsal was very low: the average reads of Def were fewer than 30 and those of Mtk fewer than 10 across all groups, with no significant differences. These findings suggest that Def and Mtk are not key downstream genes in ISCs. Therefore, we did not include these results in the main text.

In the RNA-seq dataset from *Pseudomonas entomophila* (*P.e*) infection, PTEN, Def, and Mtk were all upregulated (SI Appendix, Fig. S1K and S5A), whereas Foxo showed no obvious change (SI Appendix, Fig. S5A). We have added these heatmap results and a corresponding discussion regarding these results in the revised manuscript (Page 11, Lines 323–333).

“Pten is a lipid phosphatase that antagonizes the PI3K/Akt pathway (Mukherjee et al, 2021). Both physiological and oncogenic activation of PI3K signaling elevate the expression of its negative regulator, Pten (Mukherjee et al, 2021). Consistent with this, our RNA-seq analysis revealed significant upregulation of Pten upon infection or in intestinal tumors, concomitant with activation of the Toll/PI3K/Akt pathway (SI Appendix, Fig. S5A and S6C), suggesting a compensatory feedback mechanism to maintain pathway homeostasis. A key downstream effector, Foxo, is phosphorylated by Akt, leading to its cytoplasmic retention and functional inactivation (Mukherjee et al, 2021). Accordingly, Foxo transcript levels remained stable in intestinal tumors or were slightly reduced upon infection (SI Appendix, Fig. S5A, S6C). This suggests that Akt primarily regulates Foxo through phosphorylation-mediated inactivation rather than by modulating its mRNA expression. Together, these findings highlight a finely balanced regulatory network in which Toll signaling modulates PI3K/Akt activity to control stem cell proliferation.”

- Figure 4I - the authors state that "...Toll or Dif inhibition significantly suppressed the p-Akt activity in intestinal tumor (Fig. 4H-I)." However, while the phenotype of suppression of proliferation in Fig 4H is clear, I am not equally convinced that this reduces phospho-Akt. I see that, with the reduction of the number of GFP+ cells, the territory where p-Akt signal is found is also reduced. But the expectation is that the intensity of the phospho-Akt antibody should go down in individual cells. This requires a quantification of the antibody signal, with a proper normalisation approach (not difficult, but not trivial either).

Response:

We sincerely appreciate the referee's critical feedback on the quantification of p-Akt activity. To address the concern, we have performed p-Akt intensity analysis per ISC using ImageJ: mean fluorescence intensity of per ISCs was measured using the mean gray value function, and then subtracting the mean fluorescent background calculated from 3-5 ROI neighboring blank areas. We have also added negative control validation—Akt-RNAi flies (negative control) exhibited a 85.2% reduction in p-Akt intensity ($p < 0.001$) compared to w^{1118} flies, confirming antibody specificity and signal authenticity (new Fig. 3B-D, SI Appendix, Fig. S5D). Detailed methods are expanded in Materials & Methods (Page 18, Lines 523–526).

Our results showed that Toll or Dif inhibition significantly decreased both the number of p-Akt-positive stem cells and p-Akt levels per ISC.

- The authors do not cite previous work done on jumu in intestinal stem cells: Doupé et al., 2018 (10.1073/pnas.1719169115), where the authors define it as a stem cell-specific TF and show that its knock-down in the *esg+* population leads to a slight increase in GFP+ labelled cells (which is the opposite of what you would expect according to the model proposed here, albeit with much less data to support the observation). Further, you characterise jumu as a "stress-inducible factor" but jumu has been identified as a stem-cell specific factor in homeostatic conditions in the work of Doupé et al. (2018) as well as earlier in Dutta et al., 2015 (10.1016/j.celrep.2015.06.009). This is overlooked in the discussion.

Response:

We thank the reviewer for highlighting the prior work on jumu in intestinal stem cells (Doupé et al., 2018; Dutta et al., 2015). These studies indeed identified jumu as a stem-cell-specific transcription factor under homeostatic conditions. In our study, phospho-histone H3 (PH3)-positive cells were manually quantified across the entire midgut. While Doupé et al. reported a modest increase in

GFP+ ISCs in the posterior hindgut upon jumu RNAi, our systemic analysis of the entire midgut revealed a significant reduction in pH3+ cells and impaired proliferation. This discrepancy likely reflects methodological differences: their quantification relied on localized posterior midgut GFP+ cell counts in single sections, whereas our whole-midgut approach captures global PH3+ cell dynamics, including homeostatic conditions and stress-induced regenerative responses.

We have incorporated the following additions to the Discussion section and discussed the divergence between our findings and those of Doupé et al., and clarifies Jumu's context-dependent roles (Page 13, Lines 366-375).

"Notably, we characterized Jumu functions as the upstream regulator of Spz to coordinate Toll activity in the intestine. Previous studies have established Jumu as a stem cell-specific transcription factor under homeostatic conditions in *Drosophila* (Doupé et al, 2018; Dutta et al, 2015). While Doupé et al. observed a modest increase in GFP+ ISCs in the posterior hindgut upon Jumu RNAi, our whole-midgut analysis showed that Jumu knockdown reduced PH3+ ISCs, impairing proliferation during homeostasis, regeneration, and tumorigenesis. This discrepancy may stem from variations in the tissue regions examined (posterior hindgut vs. entire midgut) and the quantification methodologies employed (localized sectional counts vs. whole-midgut assessment). Importantly, we demonstrated that Jumu directly binds the Spz promoter to drive its expression, thereby activating the Toll/PI3K/Akt signaling cascade and regulating ISC proliferation and tumor growth."

- The section entitled "Jumeaux is essential for spz expression to potentiate Toll activation during regeneration" does not show what the title says, only correlation. This seems a title for the following section.

Response:

Thank you for pointing this out. Upon careful consideration, we agree that the current title may overstate the mechanistic evidence presented in this particular section, which primarily establishes a correlative relationship between Jumeaux and Spz expression during regeneration. To better reflect the content, we revise the title to "The expression of Jumeaux is correlated with Spz in intestine" (Page 9, Line 247 in the revised Ms).

Comments on Methods:

- The methods say, regarding infection experiments: "... statistical significances ... between two groups were performed using Gehan-Breslow-Wilcoxon tests...". Can the authors justify why this test

was chosen, when survival analyses are usually performed using Log-rank tests (with the most relaxed assumptions)?

Response:

We thank the reviewer for this comment. We have re-analyzed the survival data using the Log-rank (Mantel-Cox) test, which is the standard method for survival analysis and makes fewer assumptions regarding the hazard functions. The results are consistent with our previous findings using the Gehan-Breslow-Wilcoxon test, supporting the conclusions presented in the manuscript. We have updated the all figures of lifespan curves and methods section accordingly to indicate that the Log-rank test was used for all survival curve comparisons.

- The 'general' statistical analysis is described as two-tailed student's tests, but no mention is made of tests of the assumption (normality), like Shapiro-Wilk tests. Are the authors certain that this is the appropriate test for their analyses. Also, it is not clear whether this covers everything that is not bioinformatics and the infection survival - but there is also the survival analyses for the tumour effects, whose statistical analysis is not described.

Response:

We thank the reviewer for this insightful comment. For all comparisons of continuous variables, we have now tested the normality of the data using the Shapiro-Wilk test. For datasets following a normal distribution, comparisons between two groups were performed using a two-tailed Student's t-test. For multiple-group comparisons, one-way analysis of variance (ANOVA) was applied, followed by Tukey's post hoc test for pairwise comparisons. The Log-rank (Mantel-Cox) test was used to compare survival curves across genotypes. For differential expression analyses of bulk RNA-seq data, P values were calculated in DESeq2 using the Wald test. Pearson correlation coefficients and corresponding pvalues were calculated using the R function cor.test. Gene enrichment analysis was performed using a hypergeometric distribution algorithm to assess the significance of gene enrichment (Page 17-18, Lines 511-519).

Regarding survival analyses for tumor experiments, we have clarified that the Log-rank (Mantel-Cox) test was used to compare survival curves across genotypes, consistent with the standard method for survival analyses. We have updated the Materials and Methods section to explicitly describe the statistical tests for all datasets, including both infection and tumor survival experiments, to ensure transparency and reproducibility (Page 17, Lines 515-516).

- The description of the Cut and Tag method, which is not routine in the *Drosophila* midgut, is not at the same level of detail compared with the description of the histofluorescence (which is more established). The paper would be more valuable to the community if this was described more accurately, especially aspects that are not in any manufacturer's instructions (e.g. single cell suspension or the exact composition, with catalog numbers, of the dig-wash buffer, timings etc).

Response:

We appreciate the reviewer's comment. The CUT&Tag experiments were performed following the manufacturer's instructions of the commercial kit. The only deviation from the standard protocol concerns the preparation of single-cell suspensions from *Drosophila* midgut, for which we have developed an optimized procedure. The detailed steps of single-cell suspension preparation have been described in the Methods section. (Methods, subsection " CUT&Tag assay").

Comments on Figures:

- Heatmaps in S1G, S4A, 4C- what are the numerical units? Where is the data coming from? Besides referencing it, it would be helpful to have the samples described in the legend at least, if not the main text where this is referenced.

Response:

The heatmaps in Figures S1G, S4A, and 4C display normalized gene expression values, represented as Z-scores based on log₂-transformed FPKM from RNA-seq data. The color gradient reflects relative expression levels, with red indicating high expression and blue indicating low expression. The data sources are explicitly labeled in the figure legends (e.g., GSE128489 and GSE112331).

- Figure 6I - the legends says that 'jumu peaks are highlighted in orange' but one can see several other peaks that are not highlighted. Why is that? The criterion for what is a peak seems unexplained.

Response:

Thank you for pointing this out. In Figure 6I, the orange highlights indicate CUT&Tag peaks that overlap with the spz locus and pass our stringent peak-calling criteria. In our analysis, peaks were definitively identified based on both statistical significance (MACS2, $q < 0.01$) and genomic context. Specifically, among the significant peaks passing the q-value threshold, the one closest to the

transcription start site (TSS) was selected as the representative peak for downstream analysis and visualization. The additional signal profiles visible in the track may represent sequencing background and lower-confidence enrichments that did not meet these criteria and therefore were not marked as confident peaks. We have revised the figure legend to clarify that only these high-confidence peaks ($q < 0.01$, TSS-proximal) are highlighted in orange. In addition, we provide new evidence that show Jumu binds to the TSS of Spz locus in presence or absence of DSS treatment (New Figure 7A). We further showed Jumu RNAi suppressed the expression of Spz in the intestine (Figure 7B). These results together suggest Jumu bound the gene locus of Spz and regulate its transcription.

- Figure 5A,C, please indicate numbers of flies per curve (or a range for all).

Response:

Thank you for your comment. The numbers of flies used for each survival curve in Figure 5A and 5C are summarized in the table provided in Figure 5B and 5D. We have revised the figure legend to clarify this information.

- Coexpression panels from Figures 1A, 3B and 4I should have the different channels shown in separate panels, in addition.

Response:

The coexpression panels in Figures 1A, 3B, and 4I (new fig. 1A, 3D, and SI Appendix, Fig. S7F) was revised to display individual fluorescence channels as separate insets alongside merged images.

Comments on Writing:

- I am sorry to say this but the manuscript has numerous points of ambiguity and lack of clarity, as well as minor grammatical and punctuation mistakes, and formatting errors -- too numerous for me to single them out here. As a non-native speaker myself, I can only sympathise, but for the sake of the reader, this is an important necessary improvement. In general, I can point out at some persistent errors:

* Very often the 's' for the 3rd person singular in the verb is missing.

* In the introduction it is difficult to follow which statements belong to which organism or research system. This should be made explicit.

* Formatting of genes and transgenes (capitalisation, use of italics) does not follow conventions.

Notch, jumu and spz are cases in point. Another are UAS-X transgenes, which should be in italics.

- Specific examples of imprecise meaning:

* In the main text, where Figure 3EF is referred to, the phenotype is described as "stem cell depletion", but the figures are supposedly depicting the number of mitotic ISCs.

* "colorectal tumorigenesis, renal fibrosis and other neurodegenerative diseases" -- "others" seems out of place here.

* "We ... examined the expression ... of Toll and PI3K/Akt dependent gene expression..." -- why 'dependent'?

* The section title "The proactive role of Toll/Akt/PI3K pathway in tumor growth" is vague. What does proactive mean? I would suggest the authors just say what the section shows - that the proliferative capacity of the tumours seen in the previous section correlates with disease progression.

Response:

We sincerely thank the reviewer for the careful reading and constructive feedback on the clarity and language of our manuscript. We have thoroughly revised the text with special attention to grammar, punctuation, and formatting, and we believe that these changes have significantly improved readability. To address the reviewer's specific concerns, we carefully corrected all instances of missing third-person singular "-s" in verbs.

In the Introduction, we have clarified the description of studies from different organisms, making explicit which findings are from *Drosophila* and which are from mammalian systems.

We revised the formatting of genes and transgenes according to standard conventions: gene symbols (Notch, jumu, spz etc.) are italicized, while proteins are presented in Roman type, and UAS-transgenes are italicized as well.

We have corrected the description in the Results where "stem cell depletion" was incorrectly used; it now accurately refers to "proliferation inhibition".

The phrase "colorectal tumorigenesis, renal fibrosis and other neurodegenerative diseases" has been revised to "colorectal tumorigenesis, renal fibrosis, and neurodegenerative diseases" for precision.

The description "Toll and PI3K/Akt-dependent gene expression" has been rephrased to "Toll and PI3K/Akt-regulated genes" to avoid redundancy.

The section title "The proactive role of Toll/Akt/PI3K pathway in tumor growth" has been revised to "Toll/PI3K/Akt signaling drives intestinal tumor growth," which better reflects the results.

Referee #2:

Summary

In this paper, the authors show a non-immune function of Spz/Toll signaling in intestinal stem cell proliferation in homeostasis, regeneration and tumour-like pathophysiological conditions. Toll signaling drives intestinal stem cell proliferation by regulating the transcription of insulin signalling targets, PI3K and Akt. Through molecular and bioinformatic analysis, the authors show that NF- κ B factors Dif and Dorsal are transcriptionally regulating the expression of PI3K and Akt downstream of Toll signaling in an insulin-independent manner. Further, epistatic studies, show that downregulating Toll activation or Akt activation can rescue the proliferation of intestinal tumors and improve the lifespan of the animals. Transcriptomic and genetic analysis suggest that the stress-responsive factor Jumu regulates the expression of Spz, and inactivation of the Jumu/Spz/Toll cascade reduces intestinal regeneration as well as tumorigenesis .

This represents an intriguing non-immune role of Toll signaling in adult tissue homeostasis in *Drosophila*. Several pertinent questions have to be addressed to consolidate the results.

Major comments

1. Figure 1D and E: Apart from the effect on ISC proliferation, is there a role for the studied genes in other cell types (EC and EE) in the intestine? Is it the survival and/or the differentiation of ISCs affected?

Response:

We thank the reviewer for this insightful question. To address whether the studied genes may have a role in other intestinal cell types, our collaborator Petros Ligoxygakis lab performed RNAi knockdown of Toll, Dorsal, and Dif in ECs and EBs, but did not observe significant changes in the number of PH3+ cells, suggesting no direct effect on proliferation from these cell types. In addition, we examined EE differentiation using pros as a marker, and found that ISC-specific knockdown of PGRP-SA, Toll, or Spz promoted a shift of ISC fate towards EE differentiation (see figure below). While this phenotype is intriguing, it was not the main focus of the present study. Interestingly, we also observed that ISC-specific overexpression of Toll, Dif, Akt, or Pi3K21B significantly reduced the lifespan of flies (SI Appendix, Fig. S5G and H), supporting the idea that hyperactivation of this pathway in ISCs has deleterious effects at the organismal level.

Figure for referees with unpublished data not shown.

2. In S3C, expression of p-akt can be also seen in non GFP cells. Are these enterocytes? If so, have the authors checked whether there is a contribution of these cells in Toll mediated stem cell proliferation?

Response:

We appreciate the reviewer's careful observation. The p-Akt signal detected in non-GFP cells in Fig. S3C may indeed represent enterocytes; however, the intensity of p-Akt staining in ECs is relatively weak compared to that in ISCs, and the signal is difficult to unambiguously evaluate. Importantly, Petros Ligoxygakis lab knocked down Toll, Dorsal, or Dif specifically in ECs and did not detect significant changes in PH3⁺ cell numbers. These data suggest that Toll signaling in ECs does not contribute to ISC proliferation, either because Toll is not expressed in ECs or it does not regulate proliferation through Akt in this cell type.

3. In figure S5D *esg>GFP*; Notch RNAi looks like wildtype gut with minimal to no proliferation whereas and figure 4D *esg>GFP*; Notch RNAi shows gut with more GFP cells and more ph3 incorporation. Why is this variability? Same is visible in the respective ph3 quantitation as well.

Response:

We thank the reviewer for pointing out the apparent variability between Figure 4D and Figure S5D. The difference arises from the experimental time points used. In Figure 4D (new fig 4H), *esg>GFP*;

Notch RNAi guts were analyzed at day 8, when robust tumors are established. However, UAS-Dif flies carrying the same tumor background do not survive to day 8 (Fig. 5C). Therefore, in Figure S5D (new SI Appendix, Fig. S6D and E), we analyzed guts at day 4 to enable comparison with the UAS-Dif background. At this earlier time point, the effect of Notch RNAi is weaker, probably due to residual Notch activity and compensatory mechanisms, leading to a phenotype that appears closer to wild-type. Nonetheless, activation of UAS-Dif rapidly promotes tumor growth even within this shorter time window.

To minimize confusion, we have repeated the experiment and replaced the control Notch RNAi images at day 4 with clearer examples showing tumor-like features (although still weaker than at day 8) (new SI Appendix, Fig. S6D and E). Regarding PH3 counts, they are relatively low in day-4 Notch RNAi guts due to the weaker tumor phenotype, while in UAS-Dif tumors proliferation is extremely strong (average >1000 PH3+ cells), which accentuates the apparent difference when plotted together.

4. Figure S5 E: Even though there is less ph3 incorporation, the nuclei number also seems to be higher in akt-RNAi and toll-RNAi combined with Apc, rasv12 compared to the Apc,rasv12 only guts. What is the status of cell death in these genetic backgrounds?

Response:

As suggested by the reviewer, we examined apoptosis in these tumor models using both DCP1 and TUNEL staining. Consistent with the reviewer's observation, *Apc-RNAi*, *UAS-Ras^{V12}* tumors show a high level of cell death. Importantly, we found that knockdown of Dif, Toll, or Akt in the *Apc-RNAi*, *UAS-Ras^{V12}* background significantly reduced the number of apoptotic cells (new SI Appendix, Fig. S7C, D and E). This suggests that Toll/Akt signaling promotes proliferation and induce death of neighbouring cells in the tumor context, which is consistent with our previous observation that tumor cells eliminate surrounding wildtype cells for their expansion and overgrowth (Zhou *et al*, 2021). Inactivation of Toll/Akt pathway inhibit tumor growth, as a result, the death of surrounding cell is also reduced.

5. Figure S5 E: Why is there significant loss of stem cell specific GFP expression in akt-RNAi and toll-RNAi. An image of a wildtype gut also should be incorporated for reference in all these analyses.

Response:

We thank the reviewer for his comments and suggestion. The loss of stem cell specific GFP expression after Akt and Toll-RNAi are because of Toll/Akt pathway is not only required for stem cell proliferation,

but also for the maintenance of stem cell. In consistent with our observation, Ohlstein and colleagues et al., showed InR/PI3K/Akt pathway is required for stem cell activation and survival (Choi *et al*, 2011). In the *Apc*, *Ras*^{V12} tumor background, stem cells are strongly driven into an activated state by EGFR and Wnt signaling. Blocking the Toll/Akt pathway impairs both the proliferation and the survival of stem cells, leading to the observed loss of GFP-positive cells. To address the reviewer's request, we have now included an *esg*^{ts>w}¹¹¹⁸ control gut as a wild-type reference in Figure S5.

6. Does blocking systemic insulin has any impact onto toll mediated activation of Akt and Pi3K and further tumorigenesis?

Response:

We thank the reviewer for this insightful question. Our new data indicate that overexpression of Toll or Dif strongly enhanced stem cell proliferative activity, whereas treatment with Akt or PI3K inhibitors significantly suppressed this hyperproliferative phenotype (SI Appendix, Fig. S5I, J). Importantly, Akt inhibition effectively reduced both baseline tumor growth and the excessive proliferation induced by UAS-Spz or UAS-Dif (Fig. 4C and F, SI Appendix, Fig. S6D and E). These results suggest that the activation of Toll signaling promotes Akt/PI3K pathway activity in intestinal stem cells to drive intestinal tumorigenesis.

7. Does overexpression of akt or toll independently of tumors have an impact on the life span of animals?

Response:

We thank the reviewer for raising this concern. As suggested, we performed new experiments by overexpression of Akt or Toll independently of tumor condition and observed that ISC-specific overexpression of Toll, Dif, Akt, or Pi3K21B significantly reduced the lifespan of flies (SI Appendix, Fig. S5G and H), suggesting that sustained activation of proliferation adversely affects overall organismal health (SI Appendix, Fig. S5G and H).

8. Does overexpression of toll in Jumu loss of function rescues stem cell proliferation in homeostatic condition?

Response:

We thank the reviewer for the insightful question. To address this, we generated flies (*ISC^{ts}>UAS-Toll; Jumu-RNAi*) carrying both UAS-Toll and Jumu-RNAi and examined ISC proliferation (PH3 staining). We found that overexpression of Spz induced robust ISC proliferation, which was unaffected by simultaneous Jumu knockdown (Fig. 7C). These results indicate that Jumu acts upstream of Spz/Toll signaling, and loss of Jumu cannot block the proliferative effect of Toll overexpression under homeostatic conditions.

9. Does overexpression of Jumu induce hyper-phosphorylation of Akt as observed in UAS-Dif and UAS-toll?

Response:

We appreciate the reviewer's suggestion. Unfortunately, we did not obtain Jumu overexpression flies and therefore could not directly test whether Jumu overexpression induces Akt hyper-phosphorylation. However, we examined Jumu knockdown flies and found that p-Akt staining was significantly reduced, and DSS-induced Akt activation was suppressed (SI Appendix, Fig. S9E and F). These results support the conclusion that Jumu positively regulates Akt activity in intestinal stem cells.

Minor comments

1. There are grammatical and spelling errors throughout the manuscript. Please rectify/edit.

Response:

We sincerely thank the reviewer for pointing this out. We have carefully revised the manuscript to correct grammatical, spelling, and formatting errors throughout, ensuring improved clarity and readability for the readers.

2. S2A Dif GFP expression and Figure 1A cactus antibody staining looks nonspecific. Better representation is needed.

Response:

We thank the reviewer for the comment. To address this concern, we have repeated the experiments using *esg^{ts}-mCherry/Dif-GFP* flies and performed Cactus staining in *ISC^{ts}* intestines (new Fig 1A and SI Appendix, Fig. S2A). The revised images are now provided a clearer and more specific representation of Dif and Cactus expression.

Referee #3:

In this manuscript, Peng, Zhou and colleagues report data showing that the Spz/Toll/Dorsal signaling pathway, best known for its functions in innate immunity, also impacts Intestinal Stem Cell (ISC) function and regeneration in the *Drosophila* intestine. The analysis is fairly extensive and many of the findings are novel and interesting. Much of the data support the authors' conclusions, but not all, and the quality of the data is variable. One strength is that the paper provides new insights into the non-immune function of Toll signaling, which has traditionally been studied for its role in innate immunity. This research broadens our understanding of Toll receptor functions. Another strength is the identification of Jumu, which appears to regulate Spz expression and subsequently activates Toll signaling. As to detractors, the most serious issue with this manuscript is that the authors' conclusions about Spz/Toll signaling regulating ISCs via the PI3K/AKT pathway are not warranted by the data they present (see below). In addition, there are a number of assays and controls that are not sufficient, and which should be improved in order to meet typical standards in this competitive field (see details below). For these reasons, and despite the fact that Spz/Toll signaling is clearly important in ISCs, we cannot recommend publication of this work in its present form. Some significant issues with the paper are detailed below.

Response: We thank the reviewer for his constructive comments and support.

1. First and foremost, the paper has not demonstrated that Spz/Toll promotes ISC divisions by activating PI3K/AKT signaling. The authors make this conclusion based on their observation that the mRNA expression of AKT and PI3K21B (a P60 non-catalytic subunit of PI3K) rises if Dif, a transcription factor downstream of Toll, is overexpressed (Fig 3A). Additionally, Dif overexpression strongly stimulates ISC proliferation, and this effect can be blocked by AKT-RNAi. However, there is ample data in the literature showing that levels of AKT and PI3K-P60 mRNA do not typically determine functional PI3K activity (which is not measured here) and, consistently, the authors also show that overexpression of either AKT or PI3K21B has little if any effect on ISC proliferation. Thus, the data presented here show that Dif is pro-proliferative and that AKT and P60 are required for ISC proliferation, but not (importantly) that PI3K activity is pro-proliferative or that Spz/Toll/Dif regulates PI3K activity. (The pAKT stains shown in Fig 3B, 4I and S4C are quite inconclusive and would need to be quantified to be considered seriously.) The authors need to do their homework on how PI3K/AKT activity is regulated (it's by membrane association and phosphorylation), and what its

known effects on ISC proliferation are. Presently, their conclusions drawn about the pathway from the data in Figures 3D-3E and 5C-5D are invalid.

Response:

We sincerely thank the reviewer for their critical and insightful feedback regarding the evidence for Spz/Toll-mediated activation of PI3K/Akt signaling in intestinal stem cell (ISC) proliferation. We acknowledge the reviewer's concern that mRNA levels of Akt and Pi3K21B alone, as observed with Dif overexpression (Fig. 3A), do not fully demonstrate functional PI3K/Akt activity, given that PI3K/Akt signaling is primarily regulated by membrane association and phosphorylation. To address this, we have conducted additional experiments and revised our manuscript to strengthen the evidence for the role of PI3K/Akt activity downstream of Spz/Toll signaling.

Firstly, we quantified phospho-Akt (p-Akt) levels in ISCs to directly assess PI3K/Akt activity. Using ImageJ, we measured mean fluorescence intensity per ISC with the mean gray value function, subtracting average background fluorescence from 3–5 regions of interest (ROIs) in adjacent blank areas. Negative control validation with Akt-RNAi flies showed an 85.2% reduction in p-Akt intensity ($p < 0.001$) compared to w^{1118} flies, confirming antibody specificity and signal reliability (new Fig. 3B-D, SI Appendix, Fig. S5D; detailed in Materials and Methods, page 18, lines 523–526). These results revealed that inhibition of Toll or Dif significantly reduced both the number of p-Akt-positive ISCs and p-Akt levels per cell, directly linking Toll signaling to Akt activation (Fig. 3B-D, SI Appendix, Fig. S5D). Secondly, to evaluate the functional role of PI3K/Akt activity in ISC proliferation, we treated flies with Akt and Pi3K inhibitors, which significantly suppressed the hyperproliferative phenotype induced by Toll or Dif overexpression (SI Appendix, Fig. S5I, J). Notably, Akt inhibitor treatment effectively reduced both baseline tumor growth and excessive proliferation driven by UAS-Spz or UAS-Dif (new Fig. 4C and F, SI Appendix, Fig. S6D and E) confirming that Akt activity is pro-proliferative downstream of Toll signaling. Additionally, we note literature evidence that Delta-driven UAS-Pi3K overexpression or PTEN (a negative regulator of Akt) knockdown increases ISC numbers (Amcheslavsky *et al*, 2014; Foronda *et al*, 2014), supporting the pro-proliferative role of PI3K activity in this context.

Thus, these data support that Toll signaling promotes the activation of Pi3K/Akt to drive stem cell proliferation and tumorigenesis.

2. The initial characterization of the pathway (Fig 1) is superficial and incomplete. It could be improved by assaying the effects of other components (Dorsal-RNAi, Cactus-RNAi, MyD88-RNAi) with and without DSS treatment, or infection.

Response:

We appreciate the reviewer's suggestion to expand the characterization of the Toll pathway. To address this, we performed ISC-specific RNAi knockdown of additional key components, including Dorsal, Cactus, and MyD88, in addition to PGRP-SA, Spz, Toll, and Dif, and assessed ISC proliferation by PH3 staining.

Our results show that knockdown of Dorsal or MyD88 strongly reduced PH3-positive ISCs, consistent with their role in Toll signaling. In contrast, Cactus knockdown, which removes inhibition on the pathway, increased ISC proliferation (New Fig. 1D, SI Appendix, Fig. S2B). To test regenerative responses, we applied DSS and Paraquat treatments. Inhibition of Toll pathway components dampened the DSS- and Paraquat-induced ISC proliferation, whereas knockdown of Cactus enhanced regenerative responses (New Fig. 1C and E, SI Appendix, Fig. S2C–E). Similarly, ISC-specific Dif knockdown significantly reduced PH3-positive cells following P.e infection, further confirming that the Toll pathway mediates ISC proliferative responses under stress (SI Appendix, Fig. S3B and D).

Together, these expanded experiments provide a more complete characterization of the Toll pathway in ISC proliferation under homeostatic and stress-induced conditions.

3. Image quality: The images in Figure 1 (C, D, E) do not accurately represent their corresponding PH3 counts in panels F and G. The resolution is low and may lead to misinterpretation of the data. In the posterior midgut, are the PGRP-SA-RNAi cells washed off or not present? This needs clarification.

Response:

We thank the reviewer for pointing out this. The apparent mismatch between the PH3 counts in panels F and G and the images shown in Figure 1 (C, D, E) (new Fig 1B and C, SI Appendix, Fig. S2B–D) is due to methodological considerations. For the representative images, we only imaged the posterior midgut, whereas for quantification, all PH3-positive cells across the entire gut were counted to more accurately reflect ISC proliferative activity. As a result, the total counted PH3+ cells exceed what is visible in the representative images. We have supplemented the description in the Methods section to clarify this point (Page 17, Lines 505–506).

To clarify this and improve resolution, we removed the GFP channel in Figure 1 (C, D, E), leaving only DAPI and PH3 channels, which allows the PH3 signal to be seen more clearly. For the PGRP-SA-RNAi cells, we confirm their presence throughout the midgut and have replaced the original images with higher-quality versions (SI Appendix, Fig. S2B–D).

4. The manuscript does not clearly state whether the CUT&Tag data were collected under damage/DSS conditions or homeostatic conditions. Clarification is needed for proper interpretation. Also, the description of the CUT&Tag method in the Results section is inaccurate (it doesn't involve "pull-down" of transcription factors).

Response:

We appreciate the reviewer's valuable comments regarding the experimental conditions and methodology description. The CUT&Tag data were collected under homeostatic conditions (without DSS treatment), as now clearly stated in the Results section (Page 6, Lines 148–149). We have corrected the inaccurate "pull-down" description in the Results section to accurately reflect that CUT&Tag uses antibody-guided Tn5 transposition without immunoprecipitation (Page 6, Lines 149–151).

5. There is insufficient explanation of Jumu's role: While the data suggesting Jumu regulates Spz expression is compelling, the description of this mechanism is vague. Providing more detail on how Jumu interacts with Spz would strengthen this paper.

Response:

We appreciate the reviewer's comment. To provide more molecular details, we performed CUT&Tag-qPCR using two primer pairs targeting the Spz promoter, based on Jumu binding peaks (Fig. 6I), and one primer pair within an intron as a negative control. Our results show that Jumu binds strongly to the Spz promoter region, and this binding is further enhanced upon DSS-induced intestinal damage. These findings indicate that Jumu directly binds to the Spz promoter to transcriptionally regulate its expression, thereby modulating Toll pathway activity during intestinal regeneration (New Fig. 6A).

6. All imaging data are shown only in merged panels. While this may be acceptable for less critical figures, it becomes problematic when used to obscure complex or difficult-to-explain phenotypes. To ensure scientific transparency, the authors should include separate channel panels for Figures 1A, 1B, 3B, 4I, S2A, and S4C.

Response:

We have addressed this concern by displaying the individual fluorescence channels alongside the merged images for Figures 1A, 1B, 3B, 4I, S2A, and S4C (new Fig 1A, 3D and SI Appendix, Fig. S2A, S5D and S7F), ensuring full transparency of the imaging data.

7. Figure S2A contains essential data and should be moved to Figure 1A. Additionally, we ask the authors to clarify whether the progenitor-specific expression of Spz, Toll, Dif, Dorsal, and Cactus occurs in all progenitor cells or only a subset. Is this expression pattern consistent throughout the midgut, or confined to a specific region?

Response:

We have moved the essential data from Figure S2A to Figure 1A as suggested (new Fig 1A). Regarding the expression pattern, we found that Spz, Toll, Dif, Dorsal, and Cactus are predominantly expressed in the R4 region of the midgut, whereas p-Akt staining shows a similar distribution. While signals can also be detected in other regions of the midgut, they are much weaker compared to R4.

8. Although most Toll pathway components are expressed in progenitors, this does not directly reflect pathway activation. The Drosomycin-GFP or -lacZ reporter is a commonly used tool to visualize Toll pathway activation. We recommend that the authors use this, or another validated pathway activity reporter to determine baseline activity in progenitor cells under normal conditions, and whether DSS or infection alters this activity, as proposed. Further, is there any progenitor-to-EC signal shift under stress?

Response:

We thank the reviewer for the suggestion. In our Dif/Dorsal target gene analysis, we did not observe Drosomycin expression in the intestine. This is consistent with the notion that, in the intestinal context, the Toll pathway may primarily regulate stem cell proliferation rather than classical immune responses. Therefore, a Drosomycin-GFP or -lacZ reporter would not be suitable for assessing Toll activity in progenitor cells here. Furthermore, in our DSS-treated experiment, we did not see the expansion of Toll/Dif/Dorsal from progenitor to ECs, suggesting that there is no obvious progenitor-to-enterocyte signal shift under these stress conditions.

9. While Dif and Dorsal can form heterodimers, they don't need to form a complex for function, as they can also act independently. Dif is known to primarily mediate immune responses and Dorsal is involved in both developmental and immune processes. To better understand the role of Toll signaling in ISCs, we suggest that the authors analyze the loss-of-function effect of Dorsal. This applies to Figures 1C-1H.

Response:

We thank the reviewer for this insightful comment. To address the role of Dorsal independently of Dif, we specifically knocked down dorsal using the ISC^{ts} driver and assessed stem cell mitotic activity via PH3 staining. Consistent with its involvement in Toll signaling, dorsal knockdown led to a significant reduction in PH3-positive cells, similar to the effects observed with PGRP-SA, Spz, Toll, Myd88, and Dif RNAi (Fig. 1B and D, SI Appendix, Fig. S2B). These results demonstrate that Dorsal contributes to ISC proliferation downstream of Toll signaling, and reinforce the functional importance of the Toll pathway in intestinal homeostasis and regeneration.

10. The Toll pathway is primarily activated by Gram-positive bacteria and fungi, while the IMD pathway responds to Gram-negative bacteria. Using *Staphylococcus aureus* (S.a.) to rule out IMD pathway involvement is inappropriate without direct evidence that IMD responds to S.a. in the *Drosophila* gut. The authors should explain why they used S.a. instead of *Pseudomonas entomophila* (P.e.) in Figs. S1C-1D. Additionally, since the authors propose a progenitor-specific function for the Toll pathway, the EC-specific knockdown in Figure S1B is not suitable to support the claim that "intestinal Toll activity may not be essential for survival." This experiment should be repeated using *esgTS*.

Response:

We thank the reviewer for raising this point. We used *Staphylococcus aureus* (S.a.) to test whether Toll signaling is required for gut immune responses because Toll is primarily activated by Gram-positive bacteria and fungi, whereas Gram-negative bacteria like *Pseudomonas entomophila* (P.e.) do not efficiently trigger Toll in innate immunity. Using P.e. would not allow us to evaluate Toll-specific effects. Relish-RNAi flies were included as a negative control to show that IMD pathway inhibition does not confound the analysis, rather than to exclude IMD involvement.

It is important to note that intestinal stem cells are not directly required for classical innate immune defense; the primary immune function resides in enterocytes. Thus, EC-specific knockdown is not suitable to conclude that "intestinal Toll activity may not be essential for survival." Nevertheless, we performed ISC/EB-specific knockdown using *esg^{ts}*, and found that Dif depletion in progenitors significantly reduced survival upon S.a. infection (SI Appendix, Fig. S1E). This indicates that Toll signaling in stem/progenitor cells is important for intestinal regenerative responses following S.a.-induced damage, rather than for canonical immune resistance, as EC-specific Dif knockdown had minimal effect on survival (SI Appendix, Fig. S1C).

11. In Figure S4A, P.e. is shown to upregulate Pi3K21B and Akt expression. To fully support their proposed model the authors should provide evidence that P.e. (a Gram-negative bacterium) activates Toll signaling, and that this upregulation is dependent on Toll pathway activity.

Response:

We thank the reviewer for this comment. To address this point, we performed qPCR analysis and confirmed that P.e infection markedly upregulates Toll expression in the intestine (New SI Appendix, Fig. S1J). Consistently, intestinal damage induced by P.e. infection triggers the upregulation of Toll pathway components and downstream targets, and also induces the expression of Akt and PI3K21B (SI Appendix, Fig. S5A, B, and F). Importantly, ISC-specific knockdown of Dif significantly suppressed the P.e.-induced upregulation of Akt and PI3K, demonstrating that this induction depends on Toll pathway activity within intestinal stem cells. These results support the model that Toll signaling mediates the activation of PI3K/Akt in response to intestinal damage.

12. The gene expression analysis in Fig 2 is not very good. An overall summary of the CUT&Tag targets of Dorsal and Dif is not presented, and Fig 2B and 2D are not very useful. Fig 2F seems to be selected data; it would be more helpful to see a graph of all the top GO categories affected.

Response:

We thank the reviewer for the suggestion. To address this, we have added a table summarizing all CUT&Tag-identified targets of Dorsal and Dif (New SI Appendix, Table S2.). Figures 2B and 2D have been moved to the supplementary figures for reference, and we now present a plot of the top GO categories enriched among the targets to provide a more comprehensive view of the functional impact of Dorsal and Dif binding (new Fig. 2E).

13. The English grammar is quite poor in many instances, sometimes to the point of obscuring the meaning of the text.

Response:

We sincerely appreciate the reviewer's comment. We have carefully revised the manuscript to improve English grammar, clarity, and readability throughout the text. All major instances of ambiguous phrasing, incorrect tense, and grammatical errors have been corrected to ensure that the meaning is clear and precise.

14. Figures 4D and S5D present inconsistent controls. Why does the control in S5D not show a tumor phenotype?

Response:

We thank the reviewer for noticing the control images between the two figures. This difference is indeed due to the distinct experimental time points analyzed (8 days and 4 days), which was necessitated by the biological constraints of our genetic model. The control gut (*esg > GFP*; Notch RNAi) shown in Fig. 4D (now Fig. 4C) was imaged at Day 8, by which time robust tumorous overgrowth is fully established. However, as indicated by the survival curve in Fig. 5C, UAS-Dif overexpression in the same Notch-RNAi background is lethal by Day 8. To enable a direct comparison with the UAS-Dif genotype, the experiment in Fig. S5D (now SI Appendix, Fig. S6D and E) was therefore conducted at an earlier time point (Day 4).

At this 4-day time point, Notch-RNAi alone produces a much milder phenotype, likely due to residual Notch activity and compensatory mechanisms, causing the intestine to appear similar to wild-type. In contrast, UAS-Dif expression drives rapid and strong tumor growth even within this short window. To provide a clearer baseline for comparison, we have repeated the experiment and replaced the image for the Day 4 Notch-RNAi control with a sample that more clearly displays discernible—though still mild—tumor-like characteristics (New SI Appendix, Fig. S6D and E). Furthermore, the relatively low PH3⁺ cell count in the Day 4 Notch-RNAi control reflects its weaker proliferative phenotype at this early stage. This difference is accentuated in quantitative comparisons because the UAS-Dif tumors exhibit extremely high proliferation (often exceeding 1000 PH3⁺ cells/gut), which dramatically shifts the scale of the y-axis.

15. Figures 6E-6F lack sufficient methodological detail in the Results section and figure legends. In Figure 6D, most pro-mitotic genes are not upregulated during the damage stages (d1, d4), when regeneration is needed, but increase during the recovery phase (d5, d8), when mitosis should be decreasing to terminate regeneration. Please explain this.

We thank the reviewer for pointing out the need for methodological clarity regarding the selection process for *jumu* in Figures 6E-6F. Our approach was multi-faceted and systematic: We began with the list of 102 genes that were upregulated over time, as identified in the Venn diagram (Fig. 6C). From this list, we specifically focused on transcription factors. We then performed correlation analysis to identify which of these factors exhibited an expression pattern most closely resembling that of *spz* across the time series. This analysis highlighted *jumu* as a top candidate. We then integrated this finding with external evidence: 1) available chip-seq data from the ENCODE database indicated potential binding of *Jumu* to regulatory regions of genes involved in stress and proliferation; and 2) existing literature suggested a role for *Jumu* in regulating cell proliferation in other *Drosophila* contexts. The convergence of these three lines of evidence—co-expression with *spz*, prior ChIP-seq binding data, and known functional annotations—led us to prioritize *jumu* for further functional validation as a

key upstream regulator of spz in intestinal regeneration. We have now expanded the description of this selection methodology in the Results section (Page 9, Lines 251-256) and figure legends (Figures 6E-6F).

As for the temporal pattern observed in Figure 6D, it is indeed expected that most pro-mitotic genes show limited upregulation during the early damage phase (d1, d4) and become more pronounced during the recovery phase (d5, d8). This likely reflects the nature of the regenerative program: initial damage triggers stress responses and activation of upstream signaling (e.g., Jumu/Spz/Toll), but the transcriptional induction of pro-mitotic genes occurs slightly later to coordinate controlled proliferation during the recovery and tissue rebuilding phase. Thus, the observed kinetics are consistent with a tightly regulated regenerative process, where mitotic activity is temporally aligned with the recovery stage to avoid excessive proliferation.

16. In Figure 7A, the authors should use spz-GFP and Toll-GFP reporters to confirm whether the downregulation of spz and Toll is progenitor cell-autonomous. In Figure 7D, the DSS treatment appears ineffective and should be repeated.

We thank the reviewer for these critical comments and good suggestions, which have significantly strengthened our study. As the reviewer recommended, we directly tested whether the downregulation of Spz and Toll upon Jumu knockdown is cell-autonomous within progenitors. We generated “*esg^{ts}>Jumu-RNAi; spz-GFP*” and “*esg^{ts}>Jumu-RNAi; Toll-GFP*” reporter flies. Quantitative analysis confirmed that Jumu knockdown markedly attenuated both Spz-GFP and Toll-GFP signals in progenitor cells, compared to controls (New Fig. 7J, K; SI Appendix, Fig. S9C, D). This result provides direct evidence that Jumu regulates the expression of Spz and Toll in a progenitor cell-autonomous manner.

We agree with the reviewer that the DSS effect in the original Fig. 7D appeared variable. We have therefore repeated the entire DSS treatment experiment, increasing the sample size and ensuring rigorous, standardized conditions. The new data consistently show a clear and robust induction of regeneration by DSS, providing a more solid foundation for our conclusions (updated Fig. 7D).

Regarding Figure 7D, we repeated DSS treatment with 3% concentration, which showed clear efficacy (new Fig. 7F).

17. Minor point: the expression patterns of Adh-Gal4 and Myo1A-Gal4 should be clearly described in the relevant figure legends.

Response:

We appreciate the reviewer's suggestion. We have expanded the figure legends to explicitly describe the expression patterns of Adh-Gal4 (expressed in fat body) and Myo1A-Gal4 (expressed in ECs).

18. Please add information to the figures or the figure legends detailing the timing of the experiments.

Response:

We appreciate the reviewer's suggestion. We have added detailed timing of the experiments to all relevant figure legends

Reference:

Amcheslavsky A, Song W, Li Q, Nie Y, Bragatto I, Ferrandon D, Perrimon N & Ip YT (2014) Enteroendocrine Cells Support Intestinal Stem-Cell-Mediated Homeostasis in *Drosophila*. *Cell Rep* 9: 32–39

Choi NH, Lucchetta E & Ohlstein B (2011) Nonautonomous regulation of *Drosophila* midgut stem cell proliferation by the insulin-signaling pathway. *Proc Natl Acad Sci* 108: 18702–18707

Foronda D, Weng R, Verma P, Chen Y-W & Cohen SM (2014) Coordination of insulin and Notch pathway activities by microRNA miR-305 mediates adaptive homeostasis in the intestinal stem cells of the *Drosophila* gut. *Genes Dev* 28: 2421–2431

Maxton-Küchenmeister J, Handel K, Schmidt-Ott U, Roth S & Jäckle H (1999) Toll homolog expression in the beetle *Tribolium* suggests a different mode of dorsoventral patterning than in *Drosophila* embryos. *Mech Dev* 83: 107–114

Rodriguez-Fernandez IA, Qi Y & Jasper H (2019) Loss of a proteostatic checkpoint in intestinal stem cells contributes to age-related epithelial dysfunction. *Nat Commun* 10: 1050

Schneider DS, Hudson KL, Lin TY & Anderson KV (1991) Dominant and recessive mutations define functional domains of Toll, a transmembrane protein required for dorsal-ventral polarity in the *Drosophila* embryo. *Genes Dev* 5: 797–807

Valanne S, Vesala L, Maasdorp MK, Salminen TS & Rämets M (2022) The Drosophila Toll Pathway in Innate Immunity: from the Core Pathway toward Effector Functions. *J Immunol Baltim Md* 1950 209: 1817–1825

Valanne S, Vesala L, Maasdorp MK, Salminen TS & Rämets M (2022) The Drosophila Toll Pathway in Innate Immunity: from the Core Pathway toward Effector Functions. *J Immunol Baltim Md* 1950 209: 1817–1825

Zhou J, Valentini E & Boutros M (2021) Microenvironmental innate immune signaling and cell mechanical responses promote tumor growth. *Dev Cell* 56: 1884-1899.e5

Dear Prof. Zhou,

Thank you for the submission of your revised manuscript to our editorial offices. I have now received the reports from the three referees that I asked to re-evaluate the study, you will find below. As you will see, the referees now support publication of your study in EMBO reports. Referee #3 has remaining concerns and suggestions to improve the manuscript, I ask you to address in a final revised manuscript. Please also provide a final p-b-p-response to the remaining referee points and my editorial requests.

Editorial requests:

- Please provide a .docx formatted version of the final manuscript text (including legends for main figures, EV figures and tables), but without the figures included.
- Please have your final manuscript carefully proofread by a native speaker.
- Please provide a final title with not more than 100 characters (including spaces).
- Please order the manuscript sections like this, using only these names:
Title page - Abstract - Keywords - Introduction - Results - Discussion - Methods - Data availability section - Acknowledgements - Disclosure and Competing Interests Statement - References - Figure legends
- Thus, please remove "This PDF file includes:", "Keywords", "Competing Interest Statement" and "Author Contributions" from the title page.
- Please make sure that all the funding information is also entered into the online submission system and that it is complete and similar to the one in the acknowledgement section of the manuscript text file.
- We now use CRediT to specify the contributions of each author in the journal submission system. CRediT replaces the author contribution section. Please use the free text box to provide more detailed descriptions and do NOT provide your final manuscript text file with an author contributions section. See also our guide to authors:
<https://www.embopress.org/page/journal/14693178/authorguide#authorshipguidelines>
- There are name discrepancies. It is Shengeng Yi in the manuscript text file, but. Shengen Yi in the submission system, and Kang Han in the manuscript but. Han Kang in the system. Please check and make sure that author names in the submission system and the manuscript are identical.
- From the manuscript file it seems that Jun Zhou, Kang Han and Huijun Zhou are co-corresponding authors. In the submission system only Jun Zhou and Kang Han are indicated as co-corresponding authors. Author Huijun Zhou has not been entered into the submission system. Please check and add the missing author.
- Please note that corresponding authors are required to supply an ORCID ID upon submission of a revised manuscript and an institutional e-mail address. Please do this for co-corresponding authors Kang Han and Huijun Zhou. Please find instructions on how to link the ORCID ID to the account in our manuscript tracking system in our Author guidelines:
<http://www.embopress.org/page/journal/14693178/authorguide#authorshipguidelines>
- Please provide a complete author checklist, which you can download from our author guidelines (<https://www.embopress.org/page/journal/14693178/authorguide>). Please insert page numbers in the checklist to indicate where the requested information can be found in the manuscript. The completed author checklist will also be part of the RPF.
- Please provide individual production quality figure files as .eps, .tif, .jpg (one file per figure), of main figures and EV figures. Please upload these as separate, individual files upon re-submission.

The Expanded View format, which will be displayed in the main HTML of the paper in a collapsible format, has replaced the Supplementary information. You can submit up to 6 images as Expanded View. Please follow the nomenclature Figure EV1, Figure EV2 etc. The figure legend for these should be included in the main manuscript document file in a section called Expanded View Figure Legends after the main Figure Legends section. Additional Supplementary material should be supplied as a single pdf file labeled Appendix. The Appendix should have page numbers and needs to include a table of content on the first page (with page numbers) and legends for all content. Please follow the nomenclature Appendix Figure Sx, Appendix Table Sx etc. throughout the text, and also label the figures and tables according to this nomenclature.

- There is a Table S1 called out, that seems to contain primer information, and this seems to be uploaded as a Reagent Table. Please add this information directly to the reagents and tools table (see below), remove the uploaded file and change the callout (see Reagents & Tools Table).

- All Materials and Methods need to be described in the main text using our 'Structured Methods' format, which is required for all research articles. According to this format, the Methods section should include a Reagents and Tools Table (listing key reagents, experimental models, software, and relevant equipment and including their sources and relevant identifiers), uploaded as separate file, and a Methods section in which we encourage the authors to describe their methods using a step-by-step protocol format with bullet points, to facilitate the adoption of the methodologies across labs. More information on how to adhere to this format as well as downloadable templates (.doc) for the Reagents and Tools Table can be found in our author guidelines (section 'Structured Methods'):

Please add the primer information (Table S1) directly to the Reagents & Tools Table.

- There is a Table S2 called out, that seems to be uploaded as excel file as Reagent Table. This is a dataset. Please upload this file as Dataset EV1, update its callouts and add a legend for this file on the first TAB.

- Please check again that the number "n" for how many independent experiments were performed, their nature (biological versus technical replicates), the bars and error bars (e.g. SEM, SD) and the test used to calculate p-values is indicated in the respective figure legends (main, EV and Appendix figures). Please also check that all the p-values are explained in the legend, and that these fit to those shown in the figure. Please provide statistical testing where applicable. Please avoid the phrase 'independent experiment' but clearly state if these were biological or technical replicates. Please also indicate (e.g. with n.s.) if testing was performed, but the differences are not significant. In case n=2, please show the data as separate datapoints without error bars and statistics. See also:

<http://www.embopress.org/page/journal/14693178/authorguide#statisticalanalysis>

If n<5, please show single datapoints for diagrams. Moreover:

- Please note that the legend for figure 5 is not provided in the sequential manner. This needs to be rectified.

- Please note that the exact p values are not provided in the legends of figures 1D, E; 3 A, B, F, G, I; 4D, E, F, G, H; 5B, D, F, G; 7A, B, E, F, G, I, K

- Please indicate the statistical test used for data analysis in the legends of figures 1D, E; 2E, 3A, B, C, F, G, I; 4A, D, E, F, G, H; 5B, D, F, G; 6E, F; 7A, B, E, F, G, I, K

- Please note that the box plots need to be defined in terms of minima, maxima, centre, bounds of box and whiskers, and percentile in the legends of figures 1D, E; 3A, B, C, F, G, I; 4D, E, F, G, H; 5F, G; 7E, F, I, K;

- Please note that information related to n is missing in the legends of figures 1D, E; 3A, B, C, F, G, I; 4D, E, F, G, H; 5F, G; 7A, E, F, G, I, K.

As there were no legends provided for the supplemental figures, we could not check these. Please make sure that similar information as requested above is provided in the legends for EV and Appendix figures.

- Please add to each legend (main, EV and Appendix figures, where applicable) a 'Data Information' section explaining the statistics used or providing information regarding replicates and scales. See:

- Please provide the information regarding externally deposited datasets presently uploaded as excel file in the Data Availability Section. Then please remove the excel file from the manuscript files.

- Please upload the source data as one folder per main figure, grouping together all the files for this figure (and ZIPed together).

- We noted there are image reuses between Fig. 1B (panels "Toll-RNAi-2", "Myd88-RNAi" and "cactus-RNAi") and Fig. S2B (the second figure in the "Expanded View Content"), same panels. Please explain this reuse and if it is intentional, please clearly indicate the re-use in the respective figure legends.

- We also noted an image reuse between Fig. 3D (panel "UAS-Toll") and Fig. S5D (the fifth figure in the "Expanded View Content"), panel "UAS-Toll". It seems the image in 3D is an enlargement of a section of S5D. If this reuse is intentional, please

clearly indicate the re-use in the respective figure legends and mark the section in S5D with a box.

In addition, I would need from you uploaded separately (please remove this from the manuscript text file):

Please let me know if you have questions regarding the revision.

Referee #1:

In this re-submitted work, Peng and colleagues present clearly and convincingly how, in *Drosophila*, the innate immunity pathway Spz/Toll/Dif:dorsal is used to activate the transcription of Akt and PI3K to maintain the proliferation of adult ISCs. This is significant because this requirement for proliferation happens in homeostasis, regeneration after infection, and tumour growth, and it seems triggered by the activity of Jumu, which drives transcription of Spz. This suggests that multiple mechanisms to sense proliferative needs converge onto Jumu (and possibly the enzymes responsible for Spz activation) as a universal trigger for ISC proliferation.

I think that all the points I raised have been addressed with new experiments, analyses or in writing adequately, and I am satisfied that the paper is now ready for publication.

Referee #2:

There are still spelling errors and typos that need to be carefully checked and corrected by the authors or Editorial team.

Referee #3:

In this revision, the authors have addressed most of my questions by adding new data, controls, analyses, and improved interpolation in the main text. Overall, I am supportive of publishing this manuscript in EMBO R. However, a few remaining major and minor issues still need to be addressed in the second revision before final acceptance.

Major:

1. I still have concerns regarding the data from Pi3K21B (also known as Dp60) overexpression (Figures 3E-3F). The authors do not appear to know a fact that although Pi3K21B is necessary for the activation of Pi3K/Akt signaling, its overexpression often exerts a strong dominant-negative effect, ultimately reducing Akt pathway activity. This has been reported in several studies (PMID: 10508611; 11832249; 26490996). This casts doubt on whether the observed pro-mitotic effect of Pi3K21B overexpression is truly through Akt activation. At a minimum, the authors should provide data showing whether Pi3K21B overexpression in progenitor cells increases p-AKT levels, to support their current interpretation.

Minor:

1. The organization of the manuscript is still somewhat sloppy. For example, the supplementary figure legends are missing in this revision.
2. Some sections of the results are still too brief or lacking necessary context:
 - 1) The role of RelishRNAi in inhibiting the IMD pathway should be briefly explained, especially for readers unfamiliar with this.
 - 2) The expression pattern of Adh-Gal4 should be described, at least in the figure legend.
3. Some grammatical errors still need to be fixed.

EMBOR-2025-61477V3

Point by point response:

Editorial requests:

- Please provide a .docx formatted version of the final manuscript text (including legends for main figures, EV figures and tables), but without the figures included.

Response:

Done.

- Please have your final manuscript carefully proofread by a native speaker.

Response:

Thank you for this helpful suggestion. Prior to resubmission, the entire manuscript has been thoroughly proofread and edited by a professional native-English scientific editor, ensuring optimal spelling, grammar, clarity, and adherence to formal scientific tone. We believe the revised manuscript now meets the high language standards required for publication in EMBO Reports.

- Please provide a final title with not more than 100 characters (including spaces).

Response:

Revised Title: Spz/Toll signaling controls stem cell proliferation in intestinal regeneration and tumorigenesis

- Please order the manuscript sections like this, using only these names:

Title page - Abstract - Keywords - Introduction - Results - Discussion - Methods - Data availability section - Acknowledgements - Disclosure and Competing Interests Statement - References - Figure legends

Response:

Done.

- Thus, please remove "This PDF file includes:", "Keywords", "Competing Interest Statement" and "Author Contributions" from the title page.

Response:

Done.

- Please make sure that all the funding information is also entered into the online submission system and that it is complete and similar to the one in the acknowledgement section of the manuscript text file.

Response:

Done.

- We now use CRediT to specify the contributions of each author in the journal submission system. CRediT replaces the author contribution section. Please use the free text box to provide more detailed descriptions and do NOT provide your final manuscript text file with an author contributions section. See also our guide to authors:

<https://www.embopress.org/page/journal/14693178/authorguide#authorshipguidelines>

Response:

Done.

- There are name discrepancies. It is Shengeng Yi in the manuscript text file, but. Shengen Yi in the submission system, and Kang Han in the manuscript but. Han Kang in the system. Please check and make sure that author names in the submission system and the manuscript are identical.

Response:

Done.

- From the manuscript file it seems that Jun Zhou, Kang Han and Huijun Zhou are co-corresponding authors. In the submission system only Jun Zhou and Kang Han are indicated as co-corresponding authors. Author Huijun Zhou has not been entered into the submission system. Please check and add the missing author.

Response:

Done.

- Please note that corresponding authors are required to supply an ORCID ID upon submission of a revised manuscript and an institutional e-mail address. Please do this for co-corresponding authors

Kang Han and Huijun Zhou. Please find instructions on how to link the ORCID ID to the account in our manuscript tracking system in our Author guidelines:

<http://www.embopress.org/page/journal/14693178/authorguide#authorshipguidelines>

Response:

Done.

- Please provide a complete author checklist, which you can download from our author guidelines (<https://www.embopress.org/page/journal/14693178/authorguide>). Please insert page numbers in the checklist to indicate where the requested information can be found in the manuscript. The completed author checklist will also be part of the RPF.

Response:

Done.

- Please provide individual production quality figure files as .eps, .tif, .jpg (one file per figure), of main figures and EV figures. Please upload these as separate, individual files upon re-submission.

Response:

Done.

The Expanded View format, which will be displayed in the main HTML of the paper in a collapsible format, has replaced the Supplementary information. You can submit up to 6 images as Expanded View. Please follow the nomenclature Figure EV1, Figure EV2 etc. The figure legend for these should be included in the main manuscript document file in a section called Expanded View Figure Legends after the main Figure Legends section. Additional Supplementary material should be supplied as a single pdf file labeled Appendix. The Appendix should have page numbers and needs to include a table of content on the first page (with page numbers) and legends for all content. Please follow the nomenclature Appendix Figure Sx, Appendix Table Sx etc. throughout the text, and also label the figures and tables according to this nomenclature.

- There is a Table S1 called out, that seems to contain primer information, and this seems to be uploaded as a Reagent Table. Please add this information directly to the reagents and tools table (see below), remove the uploaded file and change the callout (see Reagents & Tools Table).

Response:

Done.

- All Materials and Methods need to be described in the main text using our 'Structured Methods' format, which is required for all research articles. According to this format, the Methods section should include a Reagents and Tools Table (listing key reagents, experimental models, software, and relevant equipment and including their sources and relevant identifiers), uploaded as separate file, and a Methods section in which we encourage the authors to describe their methods using a step-by-step protocol format with bullet points, to facilitate the adoption of the methodologies across labs. More information on how to adhere to this format as well as downloadable templates (.doc) for the Reagents and Tools Table can be found in our author guidelines (section 'Structured Methods'):

Response:

Done.

Please add the primer information (Table S1) directly to the Reagents & Tools Table.

Response:

Done.

- There is a Table S2 called out, that seems to be uploaded as excel file as Reagent Table. This is a dataset. Please upload this file as Dataset EV1, update its callouts and add a legend for this file on the first TAB.

Response:

Done.

- Please check again that the number "n" for how many independent experiments were performed, their nature (biological versus technical replicates), the bars and error bars (e.g. SEM, SD) and the test used to calculate p-values is indicated in the respective figure legends (main, EV and Appendix figures). Please also check that all the p-values are explained in the legend, and that these fit to those shown in the figure. Please provide statistical testing where applicable. Please avoid the phrase 'independent experiment' but clearly state if these were biological or technical replicates. Please also indicate (e.g. with n.s.) if testing was performed, but the differences are not significant. In case n=2, please show the data as separate datapoints without error bars and statistics. See also: <http://www.embopress.org/page/journal/14693178/authorguide#statisticalanalysis>

Response:

Done.

If n<5, please show single datapoints for diagrams. Moreover:

- Please note that the legend for figure 5 is not provided in the sequential manner. This needs to be rectified.

Response:

Done.

- Please note that the exact p values are not provided in the legends of figures 1D, E; 3 A, B, F, G, I; 4D, E, F, G, H; 5B, D, F, G; 7A, B, E, F, G, I, K

Response:

Done.

- Please indicate the statistical test used for data analysis in the legends of figures 1D, E; 2E, 3A, B, C, F, G, I; 4A, D, E, F, G, H; 5B, D, F, G; 6E, F; 7A, B, E, F, G, I, K

- Please note that the box plots need to be defined in terms of minima, maxima, centre, bounds of box and whiskers, and percentile in the legends of figures 1D, E; 3A, B, C, F, G, I; 4D, E, F, G, H; 5F, G; 7E, F, I, K;

Response:

Done.

- Please note that information related to n is missing in the legends of figures 1D, E; 3A, B, C, F, G, I; 4D, E, F, G, H; 5F, G; 7A, E, F, G, I, K.

Response:

Done.

As there were no legends provided for the supplemental figures, we could not check these. Please make sure that similar information as requested above is provided in the legends for EV and Appendix figures.

Response:

Done.

- Please add to each legend (main, EV and Appendix figures, where applicable) a 'Data Information' section explaining the statistics used or providing information regarding replicates and scales. See: <https://www.embopress.org/page/journal/14693178/authorguide#figureformat>

Response:

Done.

- Please provide the information regarding externally deposited datasets presently uploaded as excel file in the Data Availability Section. Then please remove the excel file from the manuscript files.

Response:

Thank you for your suggestion. We have removed Table EV1 from the manuscript files. In addition, Table EV2 is now provided as Dataset EV1.

- Please upload the source data as one folder per main figure, grouping together all the files for this figure (and ZIPed together).

Response:

Done.

- We noted there are image reuses between Fig. 1B (panels "Toll-RNAi-2", "Myd88-RNAi" and "cactus-RNAi") and Fig. S2B (the second figure in the "Expanded View Content"), same panels. Please explain this reuse and if it is intentional, please clearly indicate the re-use in the respective figure legends.

Response:

We sincerely thank you for your carefully checking the figures and spotting this error. The duplication of three panels ("Toll-RNAi-2", "MyD88-RNAi", and "Cactus-RNAi") between Figure 1B and Figure S2B (now Figure EV2C) was unintentional and resulted from an oversight during figure preparation. These panels do not represent new data in Fig. S2B (now Figure EV2C); they were mistakenly included again while assembling the Expanded View. In the revised manuscript, we have completely removed the duplicated panels from Figure EV2C. The remaining content of Figure EV2C has been reorganized and renumbered accordingly, and all figure legends have been updated to reflect the corrected composition.

- We also noted an image reuse between Fig. 3D (panel "UAS-Toll") and Fig. S5D (the fifth figure in the "Expanded View Content"), panel "UAS-Toll". It seems the image in 3D is an enlargement of a section of S5D. If this reuse is intentional, please clearly indicate the re-use in the respective figure legends and mark the section in S5D with a box.

Response:

The reuse of the image between Fig. 3D and Fig. S5D (now fig. EV3D) was intentional. Fig. 3D shows a higher-magnification view of a region within the corresponding panel of Fig. S5D in order to better visualize p-Akt staining and ISC morphology. We have now revised the figure legends to explicitly state this intentional re-use. In addition, we have added a box in Fig. S5D to indicate the specific area that is enlarged in Fig. 3D.

In addition, I would need from you uploaded separately (please remove this from the manuscript text file):

- a short, two-sentence summary of the manuscript (not more than 35 words).

Response:

Done.

- two to four short (!) bullet points highlighting the key findings of your study (two lines each).

Response:

Done.

- a schematic summary figure as separate file that provides a sketch of the major findings (not a data image) in jpeg or tiff format (with the exact width of 550 pixels and a height of not more than 400 pixels) that can be used as a visual synopsis on our website.

Response:

Done.

Please let me know if you have questions regarding the revision.

Please use this link to submit your revision: <https://embor.msubmit.net/cgi-bin/main.plex>

Referee #1:

In this re-submitted work, Peng and colleagues present clearly and convincingly how, in *Drosophila*, the innate immunity pathway Spz/Toll/Dif:dorsal is used to activate the transcription of Akt and PI3K to maintain the proliferation of adult ISCs. This is significant because this requirement for proliferation happens in homeostasis, regeneration after infection, and tumour growth, and it seems triggered by the activity of Jumu, which drives transcription of Spz. This suggests that multiple mechanisms to sense proliferative needs converge onto Jumu (and possibly the enzymes responsible for Spz activation) as a universal trigger for ISC proliferation.

I think that all the points I raised have been addressed with new experiments, analyses or in writing adequately, and I am satisfied that the paper is now ready for publication.

Response:

We sincerely thank the reviewer for their positive evaluation and insightful summary of our findings regarding the Jumu/Spz/Toll signaling in ISC proliferation. We are grateful for the constructive comments provided during the review process, which have significantly improved the quality of our

manuscript. We are delighted that the reviewer is satisfied with our revisions and supports the publication of this work.

Referee #2:

There are still spelling errors and typos that need to be carefully checked and corrected by the authors or Editorial team.

Response:

We sincerely thank the reviewer for pointing this out. The entire manuscript, including main text, figures, and legends, has now been carefully proofread and corrected by a professional native-English scientific editor. All spelling errors and typos have been eliminated. We apologize for the oversight and appreciate the reviewer's attention to detail.

Referee #3:

In this revision, the authors have addressed most of my questions by adding new data, controls, analyses, and improved interpolation in the main text. Overall, I am supportive of publishing this manuscript in EMBO R. However, a few remaining major and minor issues still need to be addressed in the second revision before final acceptance.

Major:

1. I still have concerns regarding the data from Pi3K21B (also known as Dp60) overexpression (Figures 3E-3F). The authors do not appear to know a fact that although Pi3K21B is necessary for the activation of Pi3K/Akt signaling, its overexpression often exerts a strong dominant-negative effect, ultimately reducing Akt pathway activity. This has been reported in several studies (PMID: 10508611; 11832249; 26490996). This casts doubt on whether the observed pro-mitotic effect of Pi3K21B overexpression is truly through Akt activation. At a minimum, the authors should provide data showing whether Pi3K21B overexpression in progenitor cells increases p-AKT levels, to support their current interpretation.

Response:

We thank the reviewer for raising this important point. To directly address the concern, we have now performed ISC-specific overexpression of Pi3K21B and confirmed that it strongly elevates p-Akt levels (new data added to Fig. 3B, D and Fig. EV3E, F). The corresponding text has been added on

page 7, lines 175–176: “ISC-specific overexpression of Pi3K21B also elevated p-Akt levels (Fig. 3B, D; Fig. EV3E, F).” These results demonstrate that, under our experimental conditions, Pi3K21B overexpression activates rather than inhibits the Akt pathway, thereby supporting our interpretation that the pro-mitotic effects occur through Akt activation.

Minor:

1. The organization of the manuscript is still somewhat sloppy. For example, the supplementary figure legends are missing in this revision.

Response:

We apologize for the continued lack of organization and the omission of the supplementary figure legends in the previous revision. We have meticulously checked and restructured the entire manuscript to ensure a cohesive flow and strict adherence to journal formatting guidelines. Crucially, all supplementary figure legends (now referred to as Expanded View Figure Legends) have been fully included and carefully proofread in the resubmitted manuscript file.

2. Some sections of the results are still too brief or lacking necessary context:

1) The role of *Relish* RNAi in inhibiting the IMD pathway should be briefly explained, especially for readers unfamiliar with this.

Response:

We thank the reviewer for this helpful suggestion. We have revised the relevant section to explain the role of *Relish* as the essential NF- κ B-like transcription factor of the IMD pathway, which is inhibited by the RNAi approach. The explanation is incorporated into the manuscript on page 4, lines 102-105: “Simultaneously, we inhibited the IMD pathway via RNAi against *Relish*, the NF- κ B-like transcription factor essential for this signaling cascade. Under these conditions, we observed no significant change in survival following either systemic or intestinal challenge with *Staphylococcus aureus* (*S.a*) (Fig. EV1B, D, F and G).” This addition provides the necessary context for readers unfamiliar with the *Drosophila* innate immune system.

2) The expression pattern of Adh-Gal4 should be described, at least in the figure legend.

Response:

We thank the reviewer for the suggestion. To clarify the expression pattern of Adh-Gal4, we have added the following description to the “Expanded View Figure Legends” of Figure EV1A and B: “The Adh-Gal4 driver induces gene expression in the fat body.”

3. Some grammatical errors still need to be fixed.

Response:

We sincerely thank the reviewer for pointing this out. We have conducted a final, comprehensive language review of the entire manuscript and ensured that all grammatical issues, sentence structures, and punctuation errors have been thoroughly corrected by a professional scientific editor.

Prof. Jun Zhou
Hunan University
School of biomedical science
Hubuhe Road
Changsha, Hunan 410000
China

Dear Prof. Zhou,

Thank you for the submission of your final revised manuscript to our editorial offices. I went through this and the final p-b-p-response and consider the remaining points of referee #3 and the editorial requests as adequately addressed.

I am thus very pleased to accept your manuscript for publication in the next available issue of EMBO reports. Thank you for your contribution to our journal.

You may qualify for financial assistance for your publication charges - either via a Springer Nature fully open access agreement or an EMBO initiative. Check your eligibility: <https://link.springer.com/journal/44319/how-to-publish-with-us>

Yours sincerely,

>>> Please note that it is EMBO Reports policy for the transcript of the editorial process (containing referee reports and your response letter) to be published as an online supplement to each paper. If you do NOT want this, you will need to inform the Editorial Office via email immediately. More information is available here: <https://link.springer.com/partners/embo-press/editorial-policies#Peer%20review>